# Southern Ocean controls of the vertical marine $\delta^{13}C$ gradient – a modelling study

Anne L. Morée[1], Jörg Schwinger[2], Christoph Heinze[12]

[1]Geophysical Institute, University of Bergen and Bjerknes Centre for Climate Research, 5007 Bergen, Norway
[2]Uni Research Climate, Bjerknes Centre for Climate Research, 5007 Bergen, Norway

*Correspondence to*: Anne L. Morée (anne.moree@uib.no)

**Abstract.** $\delta^{13}C$, the standardised $^{13}C/^{12}C$ ratio expressed in permil, is a widely used ocean tracer to study changes in ocean circulation, water mass ventilation, atmospheric $pCO_2$ and the biological carbon pump on timescales ranging from decades to tens of millions of years. $\delta^{13}C$ data derived from ocean sediment core analysis provide information on $\delta^{13}C$ of dissolved
inorganic carbon and the vertical $\delta^{13}C$ gradient (i.e., $\Delta\delta^{13}C$) in past oceans. In order to correctly interpret $\delta^{13}C$ and $\Delta\delta^{13}C$ variations, a good understanding is needed of the influence from ocean circulation, air-sea gas exchange and biological productivity on these variations. The Southern Ocean is a key region for these processes, and we show here that $\Delta\delta^{13}C$ in all ocean basins is sensitive to changes in the biogeochemical state of the Southern Ocean. We conduct a set of idealised sensitivity experiments with the ocean biogeochemistry general circulation model HAMOCC2s to explore the effect of biogeochemical
state changes of the Southern and Global Ocean on atmospheric $\delta^{13}C$, $pCO_2$, and marine $\delta^{13}C$ and $\Delta\delta^{13}C$. The experiments cover changes in air-sea gas exchange rates, particulate organic carbon sinking rates, sea ice cover, and nutrient uptake efficiency - in an unchanged ocean circulation field. Our experiments show that global mean $\Delta\delta^{13}C$ varies by up to about ±0.35 ‰ around the pre-industrial model reference (1.2 ‰) in response to biogeochemical change. The amplitude of this sensitivity can be larger at smaller scales, as seen from a maximum sensitivity of about -0.6 ‰ on ocean basin scale. The ocean's oldest
water (North Pacific) responds most to biological changes, the young deep water (North Atlantic) responds strongly to air-sea gas exchange changes, and the vertically well-mixed water (SO) has a low or even reversed $\Delta\delta^{13}C$ sensitivity as compared to the other basins. This local $\Delta\delta^{13}C$ sensitivity depends on the local thermodynamic disequilibrium and the $\Delta\delta^{13}C$ sensitivity to local POC export production changes. The direction of both glacial (intensification of $\Delta\delta^{13}C$) and interglacial (weakening of $\Delta\delta^{13}C$) $\Delta\delta^{13}C$ change matches the direction of the sensitivity of biogeochemical processes associated with these periods. This
supports the idea that biogeochemistry likely explains part of the reconstructed variations in $\Delta\delta^{13}C$, in addition to changes in ocean circulation.

## 1 Introduction

The vertical marine $\delta^{13}C$ gradient ($\Delta\delta^{13}C$) is the surface-to-deep difference in $\delta^{13}C$ of dissolved inorganic carbon (DIC), where the standardised $^{13}C/^{12}C$ ratio ($\delta^{13}C$) is expressed in permil (Zeebe and Wolf-Gladrow, 2001):

$$\delta^{13}C = \left( \frac{{}^{13}C/{}^{12}C}{({}^{13}C/{}^{12}C)_{standard}} - 1 \right) * 1000 ‰. \tag{1}$$

Here, ${}^{13}C/{}^{12}C_{standard}$ is the Pee Dee Belemnite standard (0.0112372) (Craig, 1957). ${}^{13}C$ is slightly heavier than the ${}^{12}C$ isotope which causes a fractionation effect during air-sea gas exchange and photosynthesis, thereby changing $\delta^{13}C$ and $\Delta\delta^{13}C$ (Laws et al., 1997; Mackenzie and Lerman, 2006; Zhang et al., 1995). Photosynthetic fractionation increases the ${}^{13}C/{}^{12}C$ ratio of surface ocean DIC (i.e., a $\delta^{13}C$ increase) due to the preferred uptake of the lighter ${}^{12}C$ into biogenic matter (which therefore has a low $\delta^{13}C$). The deep sea DIC has a relatively low $\delta^{13}C$ signature as a result of the remineralisation of low-$\delta^{13}C$ biogenic matter at depth. The resulting vertical $\delta^{13}C$ gradient is in addition influenced by air-sea gas exchange and circulation (Emerson and Hedges, 2008; Zeebe and Wolf-Gladrow, 2001; Ziegler et al., 2013). Both deep sea and surface ocean $\delta^{13}C$ signatures are archived in the calcareous shells of foraminifera in the sediments. Such records of $\delta^{13}C$ from planktic and benthic foraminiferal shell material cover tens of millions of years (Hilting et al., 2008). Using this archive, $\delta^{13}C$ and $\Delta\delta^{13}C$ have been used to reconstruct for example atmospheric $CO_2$ concentration, ocean circulation and the strength of the biological pump (Bauska et al., 2016; Broecker, 1982; Broecker and McGee, 2013; Crucifix, 2005; Curry and Oppo, 2005; Hollander and McKenzie, 1991; Hoogakker et al., 2015; Keir, 1991; Lisiecki, 2010; Oppo et al., 1990; Shackleton and Pisias, 1985; Zahn et al., 1986; Ziegler et al., 2013). $\Delta\delta^{13}C$ is independent of whole-ocean $\delta^{13}C$ shifts (due to terrestrial influences), because such influences would affect $\delta^{13}C$ equally everywhere. This makes $\Delta\delta^{13}C$ a valuable proxy to study the marine carbon cycle independent of changes in carbon storage on land. Besides the use of $\delta^{13}C$ for understanding the past ocean, contemporary measurements of $\delta^{13}C$ of DIC support the quantification of anthropogenic carbon uptake by the oceans as well as the study of the effects of biology and ocean circulation on tracer distributions (Eide et al., 2017b; Gruber and Keeling, 2001; Holden et al., 2013; Kroopnick, 1980; Kroopnick, 1985; Quay et al., 2003). However, major uncertainties remain in the interpretation of foraminiferal $\delta^{13}C$ records and $\Delta\delta^{13}C$ (Broecker and McGee, 2013; Oliver et al., 2010) as well as in the interpretation of the present day $\delta^{13}C$ data (Eide et al., 2017b).

This article addresses part of these uncertainties by exploring the pre-industrial sensitivity of $\delta^{13}C$ and $\Delta\delta^{13}C$ to biogeochemical change in idealised model experiments. By doing so we can investigate a number of biogeochemical mechanisms that could explain (part of) the observed changes in $\delta^{13}C$ and $\Delta\delta^{13}C$. We focus on the Southern Ocean (SO), the ocean south of 45° S, because the SO plays an important role in the global carbon cycle by regulating atmospheric $CO_2$ concentrations and uptake of anthropogenic $CO_2$ (Broecker and Maier-Reimer, 1992; Heinze, 2002; Marinov et al., 2006) as well as influencing the global efficiency of the biological pump, global primary production and preformed nutrients (Primeau et al., 2013).

Variations in $\Delta\delta^{13}C$ over the past few 100 000 years show that $\Delta\delta^{13}C$ is generally increased during glacial periods, due to a higher contrast of deep $\delta^{13}C$ with surface and mid-depth $\delta^{13}C$ (Broecker, 1982; Boyle, 1988; Charles et al., 2010; Oliver et al., 2010; Shackleton and Pisias, 1985). Long-term $\delta^{13}C$ and $\Delta\delta^{13}C$ variations have been explained by ocean circulation changes (Duplessy et al., 1988; Jansen, 2017; Oppo et al., 1990; Toggweiler, 1999; Menviel et al., 2016). However, $\Delta\delta^{13}C$ variability cannot be explained by ocean stratification/circulation changes alone: An interaction between biogeochemical and physical processes must be at play (Boyle, 1988; Charles et al., 2010; Keir, 1991; Mulitza et al., 1998; Schmittner and Somes, 2016;

Ziegler et al., 2013). $\Delta\delta^{13}C$ has been used in different ways over time: In earlier studies as the contrast between surface and deep water $\delta^{13}C$, derived from planktic versus benthic foraminifera (Boyle, 1988; Broecker, 1982; Duplessy et al., 1988; Shackleton et al., 1983) and now increasingly as the contrast of deep ocean (benthic) $\delta^{13}C$ with thermocline or intermediate ocean $\delta^{13}C$ (Charles et al., 2010; Lisiecki, 2010; Mulitza et al., 1998).

5 Here, we explore the sensitivity of $\delta^{13}C$ and $\Delta\delta^{13}C$ to changes in the biogeochemical state of the Global Ocean and Southern Ocean under a constant circulation field, to support the paleo-oceanographic interpretation of $\delta^{13}C$ and $\Delta\delta^{13}C$ as well as to improve the understanding of the SO role in global carbon cycling and its variability and sensitivity. In order to study biogeochemical mechanisms that could influence $\delta^{13}C$ and $\Delta\delta^{13}C$, a set of sensitivity experiments is conducted with the ocean biogeochemistry general circulation model HAMOCC2s (Heinze et al., 2016). We first estimate the contribution of biology

10 versus air-sea gas exchange to marine $\delta^{13}C$ of DIC (Sect. 3.2). The experiments focus on one or more of the biogeochemical aspects assumed to be important for $\delta^{13}C$ and $\Delta\delta^{13}C$, e.g. the biological pump efficiency and/or equilibration at the air-sea interface (Sect. 3.3.1-3.3.4). Together these experiments provide a broad spectrum of biogeochemical changes that could influence local and global $\delta^{13}C$ and $\Delta\delta^{13}C$. The modelling results of Sect. 3.3.1-3.3.4 are discussed in context of observational data from sediment cores (Sect. 3.4). As $\delta^{13}C$ and $\Delta\delta^{13}C$ are used to study changes in atmospheric $pCO_2$ ($pCO_2^{atm}$), a final

15 section will cover the relationship between atmospheric $\delta^{13}C$, $\Delta\delta^{13}C$ and $pCO_2^{atm}$ under different marine biogeochemical states (Sect. 3.5).

## 2 Methods

In this study we employ the ocean biogeochemistry general circulation model HAMOCC2s (Heinze et al., 1999; Heinze et al., 2009; Heinze et al., 2016) which simulates the inorganic and organic carbon cycle in the water column and in the sediments.

20 The horizontal resolution of the model is $3.5° \times 3.5°$ and there are 11 depth layers in the ocean. HAMOCC2s has an annual time step and an annually averaged fixed circulation field, as well as a one-layer atmosphere component (permitting exchange of $O_2$, $^{13}CO_2$ and $CO_2$ with the ocean component) which is assumed to be longitudinally well-mixed. The model is computationally very economic and thus an ideal tool for sensitivity experiments over long integration times. Biogenic particulate matter in the model is represented as particulate organic carbon (POC), calcium carbonate ($CaCO_3$) and biogenic

25 silica (opal). These biogenic particles are only modelled as export production due to the annual time-step of the model. POC and opal export production are described by Michaelis-Menten kinetics for nutrient uptake, limited by phosphate and silicic acid respectively (SI 1A). $CaCO_3$ export production depends on the ratio between opal and POC production. POC is carried as a tracer as well as transported downwards according to a set of mass balance equations that describe POC gain through surface layer POC production and POC losses through constant sinking and remineralisation rates (SI 1A). This is done

30 similarly for opal and $CaCO_3$ sinking and dissolution. As the model has an annual time step, sea ice is always present south of ~60° S and north of ~70° N in the control run (Fig. S1). A more detailed model description is provided in previous studies using a similar configuration of HAMOCC2s (Heinze, 2002; Heinze et al., 2016), as well as in SI 1A.

Fractionation between [13]C and [12]C during photosynthesis is set to a constant value of -20 ‰ (Lynch-Stieglitz et al., 1995; Tagliabue and Bopp, 2008), as model results are little influenced by the chosen parameterisation (Jahn et al., 2015; Schmittner et al., 2013). The fractionation during air-sea gas exchange depends on temperature according to $\varepsilon = -9.483 * 10^3 / T \text{ [K]} + 23.89$ ‰ (Mook, 1986), causing stronger fractionation at lower temperatures (i.e. at high latitudes). Fractionation during $CaCO_3$ formation is omitted from the model as done in previous studies (Lynch-Stieglitz et al., 1995; Marchal et al., 1998; Schmittner et al., 2013), as its size is uncertain but likely minor (~1 ‰) and effects on $\delta^{13}C$ and $\Delta\delta^{13}C$ are small (Shackleton and Pisias, 1985). In the version of HAMOCC2s used in this study, a fixed weathering input is used for [13]C to tune the ocean inventory to values comparable to observations. The weathering flux of [13]C into the ocean was determined by an iterative procedure: The model was run over 100,000 years replacing the weathering rate with the diagnosed burial rate for [13]C continuously. After this, the model [13]C inventory was recalibrated such that the atmospheric value for $\delta^{13}C$ arrived at -6.5 ‰. This procedure was repeated over three iterations. Afterwards, the weathering rate of [13]C was fixed to the last diagnosed value (0.36 Tmol [13]C yr$^{-1}$) – which results in a weathering flux $\delta^{13}C$ of DIC of -11 ‰. Another 100,000 yr run was carried out with this constant input rate in order to check whether the global [13]C distribution was stable in all reservoirs. The sensitivity experiments were then re-started from the result of that run. Weathering fluxes are added homogeneously over the first ocean layer as dissolved matter and in a fixed stoichiometric ratio for $CaCO_3$, organic carbon, $PO_4^{3-}$, Alkalinity, and Si. Annual weathering fluxes (Tmol) are 27 for $CaCO_3$, 5 for organic carbon, $5/r_{C:P}$ for $PO_4^{3-}$, $2*CaCO_3 - r_{N:P}*PO_4^{3-}$ for Alkalinity, and 4.5 for Si (with $r_{C:P} = 122$ and $r_{N:P} = 16$). These values are within the uncertainties of observational estimates for Si (5.6 Tmol yr$^{-1}$ (Tréguer, 2002)), $CaCO_3$ (~32 Tmol yr$^{-1}$ (Milliman et al., 1996)), and organic carbon (4 Tmol yr$^{-1}$ (Broecker and Peng, 1987)), and have been adjusted to improve the fit of the respective modelled marine tracer distributions as well as burial fluxes to observational estimates. The spinup procedure described here created a model setup with close-to-observed marine and atmospheric $\delta^{13}C$ ($\delta^{13}C^{atm}$) values and freely evolving atmospheric $pCO_2$ and $\delta^{13}C$. This equilibrated model version is referred to as the 'control run' in the remainder of this article. We define the vertical $\delta^{13}C$ gradient ($\Delta\delta^{13}C$) as:

$$\Delta\delta^{13}C = \delta^{13}C_{surface} - \delta^{13}C_{deep} , \tag{2}$$

where $\delta^{13}C_{surface}$ and $\delta^{13}C_{deep}$ are the volume-weighted mean $\delta^{13}C$ of DIC in the surface ocean (< 50 m depth, i.e. the model photic zone) and the deep ocean (lowermost wet layer in the model, if top of layer > 3 km depth), respectively. By doing so, we can compare the $\Delta\delta^{13}C$ summarised as one number between the different sensitivity experiments.

We conducted a set of sensitivity experiments to explore changes in air-sea gas exchange rate, sea ice extent (influencing both biological production and the air-sea gas exchange of carbon) and the efficiency of the biological pump through the POC sinking rate and nutrient uptake rate (Table 1). We employ the term 'efficiency of the biological pump' as a measure of the success of phytoplankton to maintain low nutrient concentrations in the surface ocean. All experiments are run for 2000 model years starting from the end of the spinup. These runtimes allowed for atmospheric quasi-equilibrium to establish (Fig S5), with an exception for the long-term effects caused by POC sinking rate changes (as studied in more detail by Roth et al. (2014)).

The equilibration timescale of $\Delta\delta^{13}C$ is much shorter than that of atmospheric $\delta^{13}C$: This is the case because 1) the long-term weathering-burial equilibration of $\delta^{13}C$ affects the whole ocean reservoir simultaneously – thus keeping $\Delta\delta^{13}C$ constant and 2) the processes that potentially influence $\Delta\delta^{13}C$ (changes in biological production and air-sea gas exchange) affect $\Delta\delta^{13}C$ on shorter (centennial to millennial) timescales. In order to compare effects of SO change and Global change, the gas exchange

rate and POC sinking rate experiments are done twice - once changing the respective model parameter for the Global Ocean and once for the Southern Ocean only (SO-only). The model parameters where changed in a way that marine biogeochemical tracer distributions (e.g. $PO_4^{3-}$, $\delta^{13}C$) remained reasonable but did provide an estimate of the sensitivity of the respective tracer to biogeochemical change. The model has a constant sea ice cover (Fig. S1), which permits gas transfer through the ice depending on ice cover thickness (the transfer rate is divided by ice thickness in cm) while light transfer is inhibited at ice

thicknesses over 0.01 cm. The maximum and minimum sea ice cover experiments (Ice large and Ice small, Table 1) approximate the Last Glacial Maximum winter extent and the modern summer extent of SO sea ice, respectively (Crosta (2009) and Fig. A22 therein) and assume full inhibition of gas and light transfer through ice for simplicity. The experiment on nutrient drawdown ($V_{max}$) alters the Michaelis-Menten kinetics of POC production by changing the maximum nutrient (i.e. $PO_4^{3-}$) uptake rate ($V_{max}^{POC}$ in SI 1A). The gas exchange experiments alter the specific gas exchange rate $k_w$ as described in more detail

in SI 1B. The POC sinking rate experiments change the sinking velocity constant $w_{POC}$ in the POC mass balance equations (SI 1A).

The contribution of biological processes versus air-sea gas exchange to $\delta^{13}C$ is calculated using the method of Broecker and Maier-Reimer (1992) as done for observations by Eide et al. (2017b) and in a modelling context by Sonnerup and Quay (2012):

$$\delta^{13}C_{bio}[‰] = \frac{\varepsilon_{photo}}{\overline{DIC}} * r_{c:p} * (PO_4 - \overline{PO_4}) + \overline{\delta^{13}C} \ , \tag{3}$$

where $\varepsilon_{photo} = -20$ ‰, $r_{c:p} = 122$ and the following model control run mean values are used: $\overline{DIC} = 2332.284 \ \mu mol/kg$, $\overline{PO_4} = 2.409 \ \mu mol/kg$ and $\overline{\delta^{13}C} = 0.656$ ‰. These values result in the modelled $\delta^{13}C_{bio}$:$PO_4^{3-}$ relationship $\delta^{13}C_{bio}$=3.18-1.05* $PO_4^{3-}$. The constant 3.18 is somewhat higher than estimated for observed $\delta^{13}C$ for which a constant of 2.8 was found by Eide et al. (2017b). This higher constant originates from the over-prediction of the model mean $\delta^{13}C$ and $PO_4^{3-}$ at depth, as seen in other models (Sonnerup and Quay, 2012). Eq. (3) assumes a constant biological fractionation as well as a constant $r_{c:p}$

ratio, and these assumptions will introduce some error in the partition of biological and air-sea gas exchange signatures derived from observed $\delta^{13}C$ to $PO_4^{3-}$ ratios (e.g., Eide et al. 2017b). For the purpose of determining $\delta^{13}C_{bio}$ in our model, these assumptions are unproblematic, since $r_{c:p}$ and $\varepsilon_{photo}$ actually are taken to be constant in the model formulation. The air-sea gas signature $\delta^{13}C_{AS}$ is approximated as the residual ($\delta^{13}C_{AS} = \delta^{13}C_{model} - \delta^{13}C_{bio}$). $\delta^{13}C_{AS}$ is 0 ‰ when $\delta^{13}C_{model}$=$\delta^{13}C_{bio}$, i.e. when the $\delta^{13}C$ can be explained by biology only. We express $\delta^{13}C_{bio}$ as a percentage to aid interpretation of the results (denoted

$\delta^{13}C_{bio}^{perc}$), because the values of $\delta^{13}C_{bio}$ in ‰ depend strongly on the chosen 'reference' values, i.e. mean DIC, $PO_4^{3-}$, and $\delta^{13}C$ (compare Schmittner et al., 2013; Sonnerup and Quay, 2012; Broecker and Maier-Reimer, 1992; Lynch-Stieglitz et al., 1995; Eide et al., 2017b). The conversion from $\delta^{13}C_{bio}$ to a percentage is calculated as follows:

$$\delta^{13}C_{bio}^{perc}[\%] = \frac{|\delta^{13}c_{bio}|}{|\delta^{13}c_{bio}| + |\delta^{13}C_{AS}|} * 100 \%$$ (4)

In our analysis, we define the total amount of air-sea carbon exchange as $F_{u+d}=F_{up}+F_{down}$, with $F_{up}$ as the upward annual carbon flux from the ocean into the atmosphere and $F_{down}$ its downward counterpart (SI 1B and Heinze et al. (1999)). $F_{u+d}$ is relevant for understanding the sensitivity of $\delta^{13}C$. The net carbon exchange is defined as $F_{net}=F_{up}-F_{down}$. The sign of $F_{net}$ indicates

whether a region is a source or a sink for carbon and is relevant for understanding changes in $pCO_2^{atm}$.

## 3 Results and discussion

### 3.1 Model control run

The model reproduces the main features of observed marine $\delta^{13}C$, as shown in Fig. 1 and Fig. S2. The modelled global mean surface ocean $\delta^{13}C_{surface}$ of DIC is higher (1.88 ‰) than deep ocean $\delta^{13}C_{deep}$ (0.67 ‰), creating a mean ocean $\Delta\delta^{13}C$ of 1.21 ‰.

In the North Atlantic and SO, $\Delta\delta^{13}C$ is least pronounced (0.9 and 0.8 ‰ respectively) due to vertical mixing between surface and deep water during deep water formation and upwelling (Duplessy et al., 1988). $\Delta\delta^{13}C$ increases with water mass age as expected from the increased imprint of remineralisation on $\delta^{13}C$. The mean modelled ocean $\delta^{13}C$ is higher by 0.16 ‰ relative to observations (Eide et al., 2017b), which is especially pronounced in the oldest water masses (Fig. S2). This is observed in other models as well and attributed to the model's relative contribution of deep water production in the North Atlantic and

Southern Ocean (Sonnerup and Quay, 2012). The modelled global export POC production is 9.6 Gt C yr$^{-1}$ of which 18 % is produced in the SO, which is within the uncertainty of observational estimates (MacCready and Quay, 2001; Nevison et al., 2012; Dunne et al., 2007; Lutz et al., 2007; Schlitzer, 2002). The atmosphere has a modelled equilibrium $pCO_2^{atm}$ of 279 ppm and a $\delta^{13}C^{atm}$ of -6.50 ‰, which developed in the model from the 'best-fit' weathering value $F_{eq}^{w}$ as described above in Sect. 2. Net air-sea gas exchange is close to zero (ventilating ~2×10$^{-7}$ Gt of carbon to the atmosphere annually). The resulting drift of

the model control over 2000 years is 7×10$^{-7}$ ‰ both $\delta^{13}C^{atm}$, and +2×10$^{-4}$ ppm for $pCO_2^{atm}$.

### 3.2 Air-sea gas exchange versus biology

The contribution of biology based on equations (2) and (3) to the $\delta^{13}C$ distribution is presented in Fig. 2, broadly in agreement with previous studies (Kroopnick, 1985; Schmittner et al., 2013). The contribution of biology to the modelled $\delta^{13}C$ distribution is generally below 45 % and has a steep gradient from the surface to the deep ocean. The (thermodynamic) fractionation effect

of air-sea gas exchange on $\delta^{13}C$ is strongly impeded by the long equilibration time of $^{13}C$, which leaves room for biological processes to contribute significantly to $\delta^{13}C$ and $\Delta\delta^{13}C$ (Eide et al., 2017a; Lynch-Stieglitz et al., 1995; Murnane and Sarmiento, 2000; Schmittner et al., 2013). In the ocean below 250m, the influence of biology increases to 35-45 % due to the remineralisation of POC, with the exception of the Arctic Ocean where no POC production is modelled due to the sea ice cover (Fig. 2b and Fig. S1). $\delta^{13}C_{bio}^{perc}$ is close to 50 % around 1000 m depth in the northern Pacific and Indian oceans, due to the old

water masses located there, which have accumulated a large fraction of remineralised DIC. At the surface, air-sea gas exchange dominates the $\delta^{13}C$ signature of DIC as visible from the low $\delta^{13}C_{bio}^{perc}$ (Fig. 2a). The only exception at the surface is in upwelling regions, where a relatively high $\delta^{13}C_{bio}^{perc}$ is expected due to high POC production and upwelled remineralised carbon. High $\delta^{13}C_{bio}^{perc}$ generally corresponds to a low-$\delta^{13}C$ water mass (compare Fig. 1 and Fig. 2), as expected from the

upwelling of $^{13}$C-depleted DIC and modelled and observed close to the Antarctic continent (Fig. 1a and observations by Eide et al. (2017a)). The results presented in Fig. 2 appear to be quite robust as $\delta^{13}C_{bio}^{perc}$ typically does not change by more than 5-10 % for the wide range of biogeochemical states as explored in the sensitivity experiments presented below.

## 3.3 Sensitivity of $\Delta\delta^{13}C$ and $\delta^{13}C$

### 3.3.1 Air-sea gas exchange of carbon

Atmospheric and marine carbon isotopic ratios are generally in thermodynamic disequilibrium due to their relatively long equilibration timescales as compared to the time of contact of a water parcel with the atmosphere. $CO_2$ equilibration through the air-sea interface takes ~4 months (Jones et al., 2014) and is inversely related to the Revelle Buffer factor, and slowed down by a factor ~20 as compared to inert gases due to carbon speciation (i.e. the adjustment of the bicarbonate pool). Again ~10 times slower than $CO_2$, the air-sea equilibration of the atmospheric isotope ratio $^{13}CO_2/^{12}CO_2$ (i.e. $\delta^{13}C^{atm}$) with marine

$DI^{13}C/DI^{12}C$ (i.e. $\delta^{13}C$) takes ~4 years (Jones et al., 2014). The equilibration timescale of the carbon isotopes is namely not facilitated by the buffering reaction of $CO_2$ with $H_2O$, but instead depends on the $DIC:CO_2$ ratio of seawater (Jones et al., 2014; Galbraith et al., 2015; Broecker and Peng, 1974). Over 90 % of the surface ocean $\delta^{13}C$ signature is set by air-sea gas exchange outside of upwelling regions across the world's oceans (Fig. 2), making the equilibration across the air-sea interface important for surface ocean $\delta^{13}C$. Understanding the effects of equilibration across the air-sea interface is thus key to understanding

global surface ocean $\delta^{13}C$ signatures. Here we explore two extreme cases, very slow but non-zero gas exchange ('Gas slow', gas exchange rate divided by 4) and very fast gas exchange to bring the air-sea equilibration close to equilibrium ('Gas fast', gas exchange rate multiplied by 4).

Our results show that the effects of changes in air-sea gas exchange on $\delta^{13}C$ mainly depend on the prior disequilibrium $\delta^{13}C_{diseq}$ ($\delta^{13}C_{diseq} = \delta^{13}C - \delta^{13}C_{eq}$, where $\delta^{13}C_{eq}$ represents the $\delta^{13}C$ value a water parcel would have had if it would have fully

equilibrated with the atmosphere, see also Gruber et al. (1999)). Full isotopic equilibrium with the atmosphere results in a $\delta^{13}C_{surface}$ of ~0.5 (low latitudes) to ~4 (high latitudes) ‰ (Menviel et al., 2015), where the range is caused by the temperature dependent fractionation (Mook et al., 1986; Zhang et al., 1995). In this study, modelled ε is between 7.7 and 11 ‰. This thermodynamic effect increases the difference between $\delta^{13}C_{surface}$ and $\delta^{13}C_{eq}$ at the poles (Menviel et al., 2015), thus increasing the potential of high latitude surface water to be affected by air-sea gas exchange fluxes.

Our gas exchange experiments (Table 1) show a transient phase where the net global air-sea gas exchange flux $F_{net}$ is non-zero, which affects $pCO_2^{atm}$ until a new quasi-equilibrium is established (Fig. S3). We find an increase of $pCO_2^{atm}$ by 10 ppm (slow gas exchange) and a decrease by 4 ppm (fast gas exchange) after 2000 years, respectively. If gas exchange is only

changed in the SO (i.e. for 22 % of the global ice-free ocean area), an effect of 3.7 and -0.7 ppm is found after 2000 years (Table 2). Gas exchange in the SO can thus cause a disproportionate response (~30 % of the sensitivity) in $p\mathrm{CO_2^{atm}}$. These changes occur in the first ~600 years of the sensitivity experiments (Fig. S3), with the strongest changes occurring after ~100 years. The air-sea $p\mathrm{CO_2}$ difference is smaller at increased gas exchange rates and larger at reduced gas exchange rates (Fig.

S4). $\delta^{13}\mathrm{C^{atm}}$ decreases by 0.3 ‰ during fast gas exchange and increases by 0.2 ‰ when the gas exchange rate is reduced. This is explained by the increase in the global amount of air-sea gas exchange $F_{u+d}$ in the fast gas exchange experiment (4 times more, at 542 Gt C $yr^{-1}$). Such an increase leads to a smaller thermodynamic disequilibrium, which increases the mean marine $\delta^{13}\mathrm{C}$ and lowers $\delta^{13}\mathrm{C^{atm}}$. Slow gas exchange reduces $F_{u+d}$ (by 73 % to 36 Gt C $yr^{-1}$), thus decreasing the role of air-sea gas exchange on surface ocean $\delta^{13}\mathrm{C}$. This results in an increased contrast between atmospheric and surface ocean $\delta^{13}\mathrm{C}$, which

explains the increase of $\delta^{13}\mathrm{C^{atm}}$. Moreover, our SO-only experiments show that these effects on $\delta^{13}\mathrm{C^{atm}}$ are more pronounced if gas exchange only changes in the SO. This indicates that the remainder of the ocean offsets part of the atmospheric sensitivity to SO change.

$\delta^{13}\mathrm{C}$ shows a different response in high latitudes as compared to the lower latitudes in the surface ocean (Fig. 3a and S5): An increased air-sea gas exchange rate lowers the surface ocean $\delta^{13}\mathrm{C}$ of DIC by 0.2 to 0.9 ‰ at the lower latitudes and increases

surface ocean $\delta^{13}\mathrm{C}$ at high latitudes by 0.2-0.5 ‰ (Fig. 3 and 4). The direction of the response indicates whether $\delta^{13}\mathrm{C_{diseq}}$ is positive or negative, and is in line with previous studies (Schmittner et al., 2013; Galbraith et al., 2015) that show that the disequilibrium is negative ($\delta^{13}\mathrm{C} < \delta^{13}\mathrm{C_{eq}}$) at high latitudes and in low latitude upwelling regions, and positive elsewhere. The sign of $\delta^{13}\mathrm{C_{diseq}}$, and thus the direction of the $\delta^{13}\mathrm{C}$ response, is understood from the difference between the natural $\delta^{13}\mathrm{C}$ distribution (Fig. 1) and the $\delta^{13}\mathrm{C_{eq}}$ which depends on thermodynamic fractionation (Sect. 2). At increased gas exchange rates

(i.e. closer to equilibrium), $\delta^{13}\mathrm{C}$ has to increase in cool high latitude surface waters and has to decreases in warm low latitude surface waters in order to get closer to equilibrium (Menviel et al., 2015; Murnane and Sarmiento, 2000). The net effect of a slower gas exchange rate on surface ocean $\delta^{13}\mathrm{C}$ is less pronounced than the effect of an increased gas exchange rate (Fig. S5, Fig. 3). The smaller effects seen for slow gas exchange indicate that the control model ocean is a 'slow ocean', i.e. closer to (very) slow gas exchange than to thermodynamic equilibrium (infinitely fast gas exchange).

The effect of the gas exchange rate on marine $\delta^{13}\mathrm{C}$ is mostly established in the top 250 to 1000 m of the water column (Fig. 3c, d, Fig. 4). Recording this air-sea gas exchange signal thus strongly depends on the reliability of planktic $\delta^{13}\mathrm{C}$-based $\delta^{13}\mathrm{C_{surface}}$ reconstructions and knowledge of the living depth represented by the planktic foraminifera. The signal penetrates deepest (to ~2000 m depth) into the North Atlantic (Fig. 4), where $\delta^{13}\mathrm{C}$ is strongly influenced by air-sea gas exchange (Fig. 2a). However, the interpretation of variations in North Atlantic benthic $\delta^{13}\mathrm{C}$ as coming partly from air-sea gas exchange (Lear

et al., 2016) is not supported by our experiment. Due to the limited export of the $\delta^{13}\mathrm{C}$ signal to depth, the sensitivity of $\Delta\delta^{13}\mathrm{C}$ to the gas exchange rate mainly comes from surface ocean $\delta^{13}\mathrm{C}$. Globally, the $\Delta\delta^{13}\mathrm{C}$ weakens to 0.87 ‰ when the thermodynamic disequilibrium is decreased (i.e. 'Gas fast', Fig. 5) and $\Delta\delta^{13}\mathrm{C}$ strengthens to 1.32 ‰ when the thermodynamic disequilibrium is increased ('Gas slow', Fig. 5). The extent to which thermodynamic equilibrium can develop is thus an efficient way to change the biologically-induced $\Delta\delta^{13}\mathrm{C}$ (Murnane and Sarmiento, 2000), however this is only true in lower latitudes

where $\delta^{13}C_{diseq}$ is positive: The direction of the high-latitude SO $\Delta\delta^{13}C$ sensitivity mirrors the sensitivity of the low latitude regions (Fig. 4) as well as the global mean due to its negative $\delta^{13}C_{diseq}$.

### 3.3.2 The biological pump: POC sinking rate

The net effect of a regionally changed biological pump efficiency depends on the sequestration efficiency, which depends on the interplay between the biological pump and ocean circulation (DeVries et al., 2012). A more efficient biological pump (here, a higher POC sinking rate) leads to a loss of carbon to the sediments, which affects $p$CO$_2^{atm}$ and $\delta^{13}C^{atm}$ on millennial timescales. Here we present results from a 2000-year simulation (as for the other experiments), which are thus transient results. A full equilibrium of the system could take as long as 200 000 years (Roth et al., 2014). On these long timescales other processes and feedbacks would occur (Tschumi et al., 2011), which complicates the attribution of changes to a primary trigger. A fast POC sinking rate leads to a $p$CO$_2^{atm}$ decrease of 23 ppm and higher (+0.2 ‰) atmospheric $\delta^{13}C$ after 2000 years (Table 2, Fig. S3) as well as an increase of mean ocean $\delta^{13}C$ of 0.15 ‰, caused by the transient sediment burial of low-$\delta^{13}C$ POC. The transient imbalance between weathering and burial fluxes can thus cause profound changes in both marine and atmospheric $\delta^{13}C$, and moreover provides an important feedback for the long-term marine carbon cycle (Roth et al., 2014; Tschumi et al., 2011). In our experiment, an efficient biological pump leads to a global ~6 % decrease in the amount of air-sea gas exchange $F_{u+d}$ because of efficient export of carbon to depth, thereby lowering the net upward advection of carbon. A mirrored but weaker response is modelled for a decrease in biological pump efficiency: Halving the POC sinking rate leads to a 13 ppm increase in $p$CO$_2^{atm}$ (of which 23 % can be explained by the SO) and a more negative atmospheric $\delta^{13}C$ (-6.7 ‰) and 7 % increased $F_{u+d}$ (Table 2, Fig. S3).

Surface ocean $\delta^{13}C$ is mostly influenced by the changes in productivity and the vertical displacement of the POC remineralisation depth. The deepening of the remineralisation depth has been extensively discussed in the literature (Boyle, 1988; Keir, 1991; Mulitza et al., 1998; Roth et al., 2014), and likely explains lowered mid-depth glacial $\delta^{13}C$ together with changes in ocean circulation (for example, Toggweiler, 1999). POC sinking removes nutrients and preferentially light $^{12}C$ carbon from the surface ocean, while exporting them to the deep ocean. If POC sinking rates are high, this increases the mean surface ocean $\delta^{13}C$ (contributing to the $\delta^{13}C^{atm}$ increase) by 0.12 ‰ and lowers mean deep ocean $\delta^{13}C$ by 0.01 ‰ (Fig. 6) – with values corrected for the overall 0.15 ‰ increase in marine $\delta^{13}C$ which occurs due to transient imbalance between weathering and sediment burial. Therefore, even though the absolute export production is globally reduced by 26 %, the biological pump is more efficient as any new nutrients in the surface ocean will immediately be used and exported. With a halving of the POC sinking rate, the remineralisation is confined closer to the surface ocean (Fig. 4). The net effect is that surface ocean $\delta^{13}C$ is reduced (in the mean by 0.21 ‰ – corrected for the mean ocean change of 0.04 ‰) throughout the ocean (Fig. 4), because the fractionation effect during photosynthesis is counteracted by the remineralisation of POC (which would normally have occurred at depth). The SO plays a relatively minor role in the sensitivity to the POC experiments (Fig. 6b). Changes in deep ocean $\delta^{13}C$ depend on the water mass age (Fig. 6c): Old water (North Pacific) has a larger remineralisation signal when the biological pump is efficient. Independent of the biological pump efficiency, the relatively young waters of the

deep North Atlantic generally adopt about the same $\delta^{13}$C signal as the surface ocean $\delta^{13}$C, which is set by air-sea gas exchange. This is in agreement with a relatively low $\delta^{13}C_{bio}^{perc}$ estimate for the deep North Atlantic (~30 %).

The sensitivity of $\Delta\delta^{13}$C to changes in POC sinking rate depends strongly on location (Fig. 4 and 6). In general, the $\Delta\delta^{13}$C strengthens for an increased biological pump efficiency (Fig. 5), and this effect is stronger with water mass age (Fig. 6c, Fig. 4). The downward shift of the remineralisation depth of low-$\delta^{13}$C POC drives this increase in $\Delta\delta^{13}$C, a mechanism discussed among others by Boyle (1988) and Mulitza et al. (1998) to understand glacial $\Delta\delta^{13}$C increase. Our results show that the vertical displacement of the $\delta^{13}$C profile is most pronounced in the North and South Pacific (Fig. 4). The North Atlantic $\Delta\delta^{13}$C is much less affected as these waters are mostly influenced by air-sea gas exchange. Instead, the entire North Atlantic profile is shifted more than in the other ocean basins (Fig. 4). $\Delta\delta^{13}$C weakens for a reduced biological pump efficiency (Fig. 4 and 5), especially in older water where $\delta^{13}C_{bio}^{perc}$ is higher (Fig. 2a). It is worth noting, however, that the changes in $\Delta\delta^{13}$C in the SO are comparably small because the vertical mixing in the SO of the low-$\delta^{13}$C deep water mostly causes shifts in the entire $\delta^{13}$C profile, not a change in the gradient (Fig. 4).

### 3.3.3 The biological pump: SO nutrient depletion

Consistent with previous studies (Primeau et al., 2013; Marinov et al., 2006; Sarmiento et al., 2004), we find a large atmospheric impact of our SO nutrient depletion experiment. The high SO nutrient uptake efficiency (i.e. an efficient biological pump) causes a 44 ppm reduction in $p$CO$_2^{atm}$ after 2000 years. The V$_{max}$ experiment reaches quasi-equilibrium after ~800 years, as seen from the time evolution of $p$CO$_2^{atm}$ and $\delta^{13}$C$^{atm}$ (Fig. S3). $\delta^{13}$C$^{atm}$ increases to -6.1 ‰ due to the increased surface ocean $\delta^{13}$C (Fig. 7a). This 0.4 ‰ increase is high compared to the results of Menviel et al. (2015), who found a $\delta^{13}$C$^{atm}$ sensitivity of 0.1-0.2 ‰ in response to changes in SO nutrient utilization. The different development time as compared to the fast POC sinking rate experiment is explained by the absence of long-term loss of carbon to the sediments in the V$_{max}$ experiment, because transport and water-column remineralisation within the SO limits an increase in POC burial there. In the SO, net carbon uptake ($F_{net}$) increases fourfold to 1.5 Gt C yr$^{-1}$ (Fig. S6) because the high nutrient and carbon consumption transport carbon into the ocean interior and do not permit CO$_2$ to escape to the atmosphere.

SO export production of POC is increased (Fig. S7) by a factor 2.4, causing global POC export production to increase by 15 % albeit reducing lower-latitude productivity (non-SO up to ~35 °N) by 13 %. This relocation of global POC export production in response to SO increased nutrient uptake efficiency is in agreement with earlier studies (Primeau et al., 2013; Marinov et al., 2006). The increased surface ocean $\delta^{13}$C signature everywhere north of the SO sea ice edge (Fig. 7a) is in the SO attributed to increased POC export production counteracted by a decreased $F_{u+d}$ (which would reduce $\delta^{13}$C$_{surface}$ in the SO because of the negative $\delta^{13}$C$_{diseq}$, Fig. 3 and S5). In lower latitudes, the decreased $F_{u+d}$ (which increases $\delta^{13}$C$_{surface}$ in lower latitudes because of the positive $\delta^{13}$C$_{diseq}$, Fig. 3 and S5) dominates the effect of the 13 % lower POC export production on $\delta^{13}$C$_{surface}$. At depth and under the sea ice in the Antarctic where deep water upwells, the imprint of additional POC remineralisation at depth decreases $\delta^{13}$C of DIC (Fig. 7). This decrease in $\delta^{13}$C is only visible in water masses downstream of the SO (Fig.7b and c) and most pronounced in the deep North Pacific (Fig. 7c). The increased nutrient uptake rate in the SO increases global mean $\Delta\delta^{13}$C

by 0.4 ‰ (Fig. 5) as well as increasing $\Delta\delta^{13}C$ in all ocean basins (Fig. 4), as seen for the fast POC sinking rate experiment. $\Delta\delta^{13}C$ is affected more in older waters, where a more pronounced remineralisation imprint has developed (Fig. 4). Besides effects on the $\delta^{13}C$ distribution (Fig. 7), the $O_2$ and $PO_4^{3-}$ distributions change as well: The $O_2$ distribution is reorganised such that surface ocean $O_2$ is increased (by up to 20 µmol $kg^{-1}$, with largest changes in the SO), while deep ocean $O_2$ is reduced

downstream of the SO (by up to 40 µmol $kg^{-1}$). Surface ocean $PO_4^{3-}$ is reduced mostly in the SO (by up to 0.8 µmol $kg^{-1}$). This signal is however too small to significantly increase mean deep ocean $PO_4^{3-}$. This implies a reduction in global preformed phosphate governed by the efficient nutrient uptake in the SO, see also Primeau et al. (2013). SO nutrient drawdown can thus cause negligible mean (deep) ocean $PO_4^{3-}$ and $\delta^{13}C$ changes, despite causing large changes in local $\delta^{13}C$ and $\Delta\delta^{13}C$ through the interplay between biology and air-sea gas exchange. Interesting in light of glacial proxy interpretation, the fit to the

$\delta^{13}C:PO_4^{3-}$ relationship is improved throughout the ocean for the $V_{max}$ experiment, similar to the effects of the 'Gas slow' experiment (Fig. 8). Changes in the goodness-of-fit of $\delta^{13}C$ and $PO_4^{3-}$ data to the $\delta^{13}C:PO_4^{3-}$ relationship (i.e. $\delta^{13}C_{bio}=3.18-1.05* PO_4^{3-}$, Sect. 2) are usually interpreted as changes in ventilation or air-sea gas exchange (Eide et al., 2017b; Lear et al., 2016). However, here we show that changes in the fit represent the relative importance of biology and air-sea gas exchange in determining $\delta^{13}C$, as both changes in $\delta^{13}C_{diseq}$ (i.e. gas exchange rate experiments) as well as changes in the biological pump

can affect the goodness-of-fit to the $\delta^{13}C:PO_4^{3-}$ relationship (Fig. 8).

### 3.3.4 Southern Ocean sea ice cover

The sea ice cover of the SO changes considerably over glacial-interglacial cycles, as well as on seasonal timescales (Crosta (2009) and Fig. A22 therein). In general, the model sea ice cover will inhibit light penetration into the surface ocean and limit air-sea gas exchange based on its thickness (Sect. 2). In the sensitivity experiments we assume complete inhibition of both

light and air-sea carbon exchange by the sea ice. In this section we thus explore the effect of both biological production and air-sea gas exchange in two extreme cases, i) the largest realistic sea ice cover based on the glacial maximum winter extreme (50° S) and ii) the smallest sea ice cover based on the contemporary summer minimum sea ice extent (70° S). Note that there is a constant sea ice cover about north of 70°N and south of 60° S in the control run of the model. Therefore, the strongest marine $\delta^{13}C$ change is expected south of 60° S for a decreased sea ice cover and between 50-60° S for an increased sea ice

cover, as this is the area where ice cover is altered relative to the control run. Ocean circulation changes that could result from a changed sea ice cover are not taken into account, as we want to study the potential isolated effect of sea ice on $\delta^{13}C$ through biological and air-sea gas exchange changes.

Both local and global air-sea carbon fluxes are influenced by a change of the SO sea ice cover, which results in changes in $pCO_2^{atm}$ and $\delta^{13}C^{atm}$. In our experiment, $pCO_2^{atm}$ increases by 9 ppm for an increased sea ice cover and decreases by 6 ppm for

a decreased sea ice cover (Table 2, Fig. S3). As noted in Sect. 3.3.1, a change in $pCO_2^{atm}$ is governed by a transient change in the net air-sea gas exchange flux $F_{net}$ until a new equilibrium is established. An extended ice cover causes more $CO_2$ to remain in the atmosphere because the additional ice covers a part of the SO that is a sink for $CO_2$ (Fig. S4 - Control). As the net global air-sea gas exchange $F_{net}$ approaches equilibrium, the non-SO ocean therefore becomes a smaller source for carbon. This

reduces the net gas exchange $F_{net}$ inside and outside of the SO by ~40-50 %. Our results show that the effects of a changed sea ice cover on $p$CO$_2^{atm}$ are yet to be fully understood: Stephens and Keeling (2000) for example modelled a strong decrease of $p$CO$_2^{atm}$ in response to an increased sea ice cover south of the Antarctic Polar Front, because they mostly cover a part of the SO that is a source of carbon to the atmosphere. In our study, the reduction in $p$CO$_2^{atm}$ by 6 ppm due to a reduced sea ice cover

is attributable to the POC production in the previously ice-covered area between ~60° S and 70° S. In a sensitivity experiment where the ice cover influences air-sea gas exchange only, the sea ice retreat leads to an increase in $p$CO$_2^{atm}$ because the region below the ice is strongly supersaturated in carbon with respect to the atmosphere. The increased sea ice cover leads to a complete suppression of air-sea gas exchange south of 50° S. Since this region is in negative carbon isotopic disequilibrium with the atmosphere ($\delta^{13}$C < $\delta^{13}$C$_{eq}$), the ice cover inhibits a $\delta^{13}$C flux into the ocean. As a result, $\delta^{13}$C$^{atm}$ increases to -6.2 ‰,

while the opposite happens for a reduced sea ice cover, leading to a lowered $\delta^{13}$C$^{atm}$ (-6.6 ‰).

The increased sea ice cover over the SO results in a surface ocean $\delta^{13}$C reduction relative to the control of -0.5 ‰ to -0.1 ‰ in the SO (Fig. 4, Fig. 9), while $\delta^{13}$C increases outside of the SO by 0-0.2 ‰ (Fig. 9a). The reduction is especially pronounced between 40-60° S. The ~40 % reduced POC export production in the SO due to light inhibition by the sea ice cover causes a major part of the SO surface $\delta^{13}$C reduction, as the absence of photosynthesis will cause lower surface ocean $\delta^{13}$C. Next to

that, the reduced air-sea gas exchange $F_{u+d}$ in the SO also leads to a lowered surface ocean $\delta^{13}$C signature. About the opposite happens when we simulate a strongly decreased sea ice cover (only ice south of 70° S): A small reduction of $\delta^{13}$C is modelled outside the SO, but the SO $\delta^{13}$C$_{surface}$ locally becomes up to ~0.8 ‰ higher relative to the control (Fig. 9b) as the increased amount of air-sea gas exchange $F_{u+d}$ decreases the carbon isotopic disequilibrium and increases POC production in the newly exposed area, both acting to increase $\delta^{13}$C of DIC.

The effect of a changed ice cover on deep ocean $\delta^{13}$C is less than ~0.1 ‰ (Fig. 9c, d) as the surface signal is diluted while it follows the general ocean circulation. As for air-sea gas exchange (Sect. 3.3.1), no pronounced deep ocean $\delta^{13}$C signal is found outside of the SO due to sea ice cover changes (this opposed to interpretations by Lear et al., 2016). Global mean $\Delta\delta^{13}$C is not significantly affected by changes in the SO sea ice cover (Fig. 5) because the low and high latitude effects on $\delta^{13}$C$_{surface}$ cancel each other out. The SO $\Delta\delta^{13}$C however weakens considerably to 0.4 ‰ when the 50-60° S region becomes covered with sea

ice and strengthens to 1 ‰ if the sea ice is removed between 60-70° S (Fig. 4). The presence or absence of a sea ice cover should thus be clearly visible in especially planktic SO $\delta^{13}$C sediment records. The effect on $\Delta\delta^{13}$C spreads downstream of the SO, where $\Delta\delta^{13}$C is increased by up to 0.2 ‰ in the Pacific Ocean for an increased SO sea ice cover (Fig. 4).

### 3.4 Modelled versus observed $\Delta\delta^{13}$C variations

The variations in $\Delta\delta^{13}$C on glacial-interglacial timescales provide researchers with a tracer to study the biogeochemical state

of the past global ocean, under the condition that we can interpret (variations in) $\Delta\delta^{13}$C. The idealised perturbations made to the (Southern) Ocean in this study show that global mean $\Delta\delta^{13}$C is particularly sensitive to an increased gas exchange rate and changes in the efficiency of the biological pump. Global mean $\Delta\delta^{13}$C varies by up to about ±0.35 ‰ around the pre-industrial model reference (1.2 ‰) in response to biogeochemical change (Fig. 5) - under the assumption of a constant ocean circulation.

However, the sensitivity of $\Delta\delta^{13}C$ to biogeochemical changes depends strongly on location for all sensitivity experiments (Fig. 4), possibly explaining part of the incoherency of reconstructed planktic and benthic foraminiferal $\delta^{13}C$ from sediment cores (Oliver et al., 2010). In general, such $\Delta\delta^{13}C$ reconstructions show $\Delta\delta^{13}C$ variations of ~1 ‰ over the past 350 000 years (Boyle, 1988; Shackleton et al., 1983; Shackleton and Pisias, 1985; Ziegler et al., 2013; Charles et al., 2010; Oliver et al., 2010). Ocean

circulation changes explain at least part of these variations in $\Delta\delta^{13}C$ (Charles et al., 2010; Heinze et al., 1991; Jansen, 2017; Heinze and Hasselmann, 1993; Oppo et al., 1990; Toggweiler 1999). However, the changes in the biogeochemical state of the ocean imposed in our experiments show that variations in $\Delta\delta^{13}C$ could be strongly influenced by (SO) biogeochemistry as well. $\Delta\delta^{13}C$ is increased during glacials and reduced during interglacials across a large set of sediment cores (Boyle, 1988; Charles et al., 2010; Oliver et al., 2010; Ziegler et al., 2013). Rapid and large changes have been documented for SO $\Delta\delta^{13}C$

records (Ziegler et al., 2013), and here we show that biogeochemical changes in the SO affect $\Delta\delta^{13}C$ globally. Our results show that an increase in mean $\Delta\delta^{13}C$ could biogeochemically result from slower gas exchange, increased POC sinking rates, or an increased nutrient uptake rate in the SO (Fig. 4 and 5). Such biogeochemical changes have been associated with glacial periods (for example, Ziegler et al. (2013)), and are potential candidates to explain part of the $\Delta\delta^{13}C$ increase in interplay with stronger ocean stratification. Sediment-core reconstructions of $\Delta\delta^{13}C$ show that an increased $\Delta\delta^{13}C$ can originate from a downward

shift of the metabolic imprint of low-$\delta^{13}C$ POC which would increase shallow $\delta^{13}C$ (Boyle, 1988; Charles et al., 2010; Mulitza et al., 1998; Toggweiler, 1999), and/or a deep ocean $\delta^{13}C$ decrease (Broecker, 1982; Boyle, 1988; Oliver et al., 2010) with little variation recorded for surface ocean $\delta^{13}C$. The absence of a clear surface $\delta^{13}C$ signal could in the SO be the net effect of an increased sea ice cover (locally decreasing $\delta^{13}C$, Fig. 4 and 9a) and an increased biological pump efficiency (locally increasing $\delta^{13}C_{surface}$, Fig. 6a and b, Fig. 7a) or increased SO thermodynamic equilibrium (Fig. 3a and b) – if these opposing

signals get mixed. A pronounced deep ocean $\delta^{13}C$ decrease is associated with an efficient biological pump and older water masses in our study (Fig. 4). Interestingly, large local changes in deep ocean $\delta^{13}C$ and $\Delta\delta^{13}C$ do not necessarily imply changes in mean deep ocean $PO_4^{3-}$ (Sect. 3.3.3).

The local character of the $\Delta\delta^{13}C$ sensitivity (Fig. 4) implies that correlations between sediment core $\Delta\delta^{13}C/\delta^{13}C$ variations and global parameters (e.g. $p$CO$_2$) are not easily extrapolated to other sediment cores over large distances. Analysis of SO $\Delta\delta^{13}C$

reconstructions from sediment cores at 42° S and 46° S (Charles et al., 2010) for example shows that there is a strong correlation between these cores and Northern Hemisphere climate fluctuations. We expect that this strong correlation does not exist for cores further south in the SO because our results indicate that the SO south of ~50-60° S often has a different $\Delta\delta^{13}C$ response to biogeochemical change than the rest of the ocean.

Interglacial periods are generally thought to be associated with a decrease in the efficiency of the biological pump and increased

deep-ocean ventilation via southern-sourced water masses (Gottschalk et al., 2016). Increased deep-ocean ventilation might be driven by increased winds (Tschumi et al., 2011), which would (apart from changing the SO circulation pattern) also increase gas exchange rates. Each of these processes indeed reduces $\Delta\delta^{13}C$ in the mean in our experiments (Fig. 5), although less pronounced so in the SO (Fig. 4). However, the interglacial reduction of $\Delta\delta^{13}C$ seems to originate from a deep ocean $\delta^{13}C$ increase as compared to the glacial deep ocean $\delta^{13}C$ (Broecker, 1982; Charles et al., 2010; Oliver et al., 2010). Our results

show that neither an inefficient biological pump nor fast gas exchange can be associated with a pronounced deep sea $\delta^{13}C$ increase relative to our control, because their effects are restricted to the surface ocean. On the other hand, the interglacial decrease of $\Delta\delta^{13}C$ is a decrease as compared to the glacial state: If glacial SO nutrient uptake was higher ($V_{max}$), a return to the 'normal' state (i.e. the model control run) would result in a major decrease of $\Delta\delta^{13}C$ (Fig. 4 and 5).

**3.5 The relationship between $\Delta\delta^{13}C$, $\delta^{13}C^{atm}$ and $p\mathrm{CO}_2^{atm}$**

One would expect variations of $\delta^{13}C^{atm}$ as well as $\Delta\delta^{13}C$ to correlate with variations in $p\mathrm{CO}_2^{atm}$, because similar processes (biology and air-sea gas exchange) steer their distribution/concentrations (Shackleton and Pisias, 1985; this article). $\Delta\delta^{13}C$ is considered a promising proxy for reconstructions of $p\mathrm{CO}_2^{atm}$ for times predating ice-core records (Lisiecki, 2010). Here we show that a positive linear relationship between $\delta^{13}C^{atm}$ and global mean $\Delta\delta^{13}C$ (Fig. 10a) holds over a wide range of

biogeochemical states as simulated in the sensitivity experiments. However, the negative linear relationship between $p\mathrm{CO}_2^{atm}$ and global mean $\Delta\delta^{13}C$ (Fig. 10b) is weak ($R^2=0.39$). However, previous studies do show the existence of a correlation between local $\Delta\delta^{13}C$ and $p\mathrm{CO}_2^{atm}$ (such as found by for example Dickson et al. (2008)), and correlation of modified $\Delta\delta^{13}C$ between ocean basins with $p\mathrm{CO}_2^{atm}$ (Lisiecki, 2010). The effects of ocean circulation on glacial-interglacial $\delta^{13}C^{atm}$ changes, not studied here, are most pronounced in response to Antarctic Bottom Water formation rate variations and are of the order of 0-0.15 ‰

(Menviel et al., 2015). Our results show that $\delta^{13}C^{atm}$ varies by up to about ±0.5 ‰ in response to biogeochemical changes (Table 2). Changes in the POC sinking rate lie approximately along a line in $\delta^{13}C^{atm}$:$\Delta\delta^{13}C$ space (Fig. 10a), suggesting that changes in the biological pump efficiency are important for the $\delta^{13}C^{atm}$:$\Delta\delta^{13}C$ relationship. Likewise, both the gas exchange rate and biological pump experiments lie along an approximate lines in $p\mathrm{CO}_2^{atm}$:$\Delta\delta^{13}C$ space (Fig. 10b, albeit a different one - leading to a low total correlation). Changes in air-sea gas exchange (as simulated in the gas exchange and sea ice cover

experiments) affect $\delta^{13}C^{atm}$ more than $\Delta\delta^{13}C$. This confirms the idea that $\Delta\delta^{13}C$ is governed by biological processes and will also set $\delta^{13}C^{atm}$, unless air-sea gas exchange gets the chance to dominate $\delta^{13}C^{atm}$. The high potential of SO air-sea gas exchange to change $\delta^{13}C^{atm}$ (Table 2: Sea ice and gas exchange rate experiments) complements recent studies showing that increased SO ventilation of deep ocean carbon is a likely candidate for glacial-interglacial $\delta^{13}C^{atm}$ excursions – which are of the order of 0.5 ‰ (Bauska et al., 2016; Eggleston et al., 2016; Lourantou et al., 2010; Menviel et al., 2015).

**4 Summary and conclusions**

This study addresses the sensitivity of modelled marine and atmospheric $\delta^{13}C$ and $\Delta\delta^{13}C$ to changes in biogeochemical parameters under constant ocean circulation, focusing on the contribution of the SO (the ocean south of 45° S, 22 % of the global ice-free ocean area). Variations of $\Delta\delta^{13}C$ recorded in sediment records are sensitive to ocean circulation changes as shown in previous studies, but here we show that the biogeochemical state of the (Southern) Ocean also can have large effects

on $\Delta\delta^{13}C$ across all ocean basins. Using the ocean biogeochemistry general circulation model HAMOCC2s, a set of sensitivity experiments was carried out, which focuses on the biogeochemical aspects known to be important for $\delta^{13}C$ and $\Delta\delta^{13}C$.

Specifically, the experiments explore changes in air-sea gas exchange rate, sea ice extent (influencing both biological production and the air-sea gas exchange of carbon) and the efficiency of the biological pump through the POC sinking rate and nutrient uptake rate.

The results show the important role of the SO in determining global $\delta^{13}C$ and $\Delta\delta^{13}C$ sensitivities, as well as the strong spatial differences in these. A new quasi-equilibrium state developed mostly within the first 100-800 years of the sensitivity experiments, except for the POC sinking experiment where an imbalance between weathering and burial causes a long-term drift. The $\delta^{13}C$ signature is governed by different processes depending on location: Air-sea gas exchange sets surface ocean $\delta^{13}C$ in all ocean basins, contributing 60-100 % to the $\delta^{13}C$ signature. At depth and with increasing water mass age, the influence of biology increases to 50 % in the oldest water masses (North Pacific) due to POC remineralisation. This spatial pattern behind the $\delta^{13}C$ signature of a water parcel results in a non-uniform sensitivity of $\delta^{13}C$ to biogeochemical change. Global mean $\Delta\delta^{13}C$ varies by up to about ±0.35 ‰ due to biogeochemical state changes in our experiments (at a constant ocean circulation) (Fig. 5). This amplitude is almost half of the reconstructed variation in $\Delta\delta^{13}C$ on glacial-interglacial timescales (1 ‰), and could thus contribute to variations in $\Delta\delta^{13}C$ together with ocean circulation changes. However, $\Delta\delta^{13}C$ can have a different response on a basin scale: The ocean's oldest water (North Pacific) responds most to biological changes, the young deep water (North Atlantic) responds strongly to air-sea gas exchange changes, and the vertically well-mixed water (SO) has a low or even reversed $\Delta\delta^{13}C$ sensitivity as compared to the other basins. The amplitude of the $\Delta\delta^{13}C$ sensitivity can be higher at decreasing scale, as seen from a maximum sensitivity of about -0.6 ‰ on ocean basin scale (Fig. 4). Interestingly, the direction of both glacial (intensification of $\Delta\delta^{13}C$) and interglacial (weakening of $\Delta\delta^{13}C$) $\Delta\delta^{13}C$ change matches changes in biogeochemical processes thought to be associated with these periods. This supports the idea that biogeochemistry explains part of the reconstructed variations in $\Delta\delta^{13}C$, in addition to changes in ocean circulation.

An increased gas exchange rate has the potential to reduce the biologically-induced $\Delta\delta^{13}C$ through the reduction of surface ocean and atmospheric $\delta^{13}C$. Increased gas exchange however only reduces $\Delta\delta^{13}C$ in the low latitudes: In high latitudes, increased gas exchange will increase $\Delta\delta^{13}C$ (by increasing $\delta^{13}C_{surface}$) because of the negative disequilibrium $\delta^{13}C_{diseq}$ (i.e. $\delta^{13}C < \delta^{13}C_{eq}$) in this region, and thus a potential to increase $\delta^{13}C_{surface}$ (section 3.3.1). Notably, $pCO_2^{atm}$, $\delta^{13}C^{atm}$ and marine $\delta^{13}C$ are shown to be disproportionally sensitive to SO gas exchange rate changes.

Changes in the efficiency of the biological pump also have a major potential to alter $\Delta\delta^{13}C$ as well as $pCO_2^{atm}$ and $\delta^{13}C^{atm}$. The globally increased POC sinking rate experiment shows that $\Delta\delta^{13}C$ strengthens in low latitudes (and more so in older waters) by deepening the low-$\delta^{13}C$ signature of remineralised POC, while SO $\Delta\delta^{13}C$ is not very sensitive to POC sinking rates. The SO effects are comparably small because the vertical mixing in the SO of the low-$\delta^{13}C$ deep water only causes shifts in the entire $\delta^{13}C$ profile, not a change in the gradient (Fig. 4). Increased POC sinking causes a long-term imbalance between weathering and sediment burial which leads to an increase in mean $\delta^{13}C$ and $\delta^{13}C^{atm}$ (of about 0.15 ‰) after 2000 years. Increased nutrient uptake in the SO ($V_{max}$ experiment) results in 13 % lower non-SO POC export production up to ~35 °N, in agreement with previous studies on the role of the SO biological pump in lower latitude productivity. Interestingly, the increase of $\Delta\delta^{13}C$ in all ocean basins occurs without significantly changing mean (deep) ocean $PO_4^{3-}$, which advocates for increased

SO nutrient uptake to explain (part of) glacial-interglacial $\Delta\delta^{13}C$ variations. Furthermore, our results show that improved goodness-of-fit of the model data to the $\delta^{13}C:PO_4^{3-}$ relationship can be driven by reduced gas exchange as well as biological uptake efficiency in the SO, since both increase the importance of biology relative to air-sea gas exchange for $\delta^{13}C$. Caution should thus be exercised when interpreting changes in the fit of observations to the $\delta^{13}C:PO_4^{3-}$ relationship as changes in ocean

ventilation or air-sea gas exchange alone.

A significant linear relationship was found across the sensitivity experiments between $\delta^{13}C^{atm}$ and $\Delta\delta^{13}C$ ($R^2=0.71$), and a weaker one ($R^2=0.39$) for $p\text{CO}_2^{atm}$ and $\Delta\delta^{13}C$. This result shows that paleo-reconstructions of $\delta^{13}C^{atm}$ based on $\Delta\delta^{13}C$ could be valid for a wide range of biogeochemical states. Previous studies have shown good correlation between $p\text{CO}_2^{atm}$ and local $\Delta\delta^{13}C$, but our results suggest that the relationship may not be valid if both biological and gas exchange rate changes occur.

The maximum response of $\delta^{13}C^{atm}$ to the biogeochemical changes imposed in our experiments (up to 0.5 ‰) is larger than the idealised maximum effect of ocean circulation changes on $\delta^{13}C^{atm}$ (0-0.15 ‰ (Menviel et al., 2015)), which underlines the potential importance of biogeochemical processes for variations in $\delta^{13}C^{atm}$. The high potential of SO air-sea gas exchange to steer $\delta^{13}C^{atm}$ (Table 2: Sea ice and gas exchange rate experiments) complements recent studies showing that increased SO ventilation of deep ocean carbon is a likely candidate for glacial-interglacial $\delta^{13}C^{atm}$ excursions.

As an outlook, the use of a more complex model with a higher horizontal and vertical resolution and a shorter time-step (resolving seasonal variations) could provide valuable additional information. For example, the role of different regions within the SO on the global $\delta^{13}C$ distribution could be better studied with a more complex model. Sediment core-based reconstructions of the global carbon cycle could possibly be aided by a more complex model with a finer grid and higher time resolution, by providing more detailed information on the contribution of biogeochemical processes to local ocean tracers. Next to that,

exploring the effect on $\Delta\delta^{13}C$ of a glacial model circulation field could provide a way to quantify the maximum combined effect of circulation and biogeochemical change on $\Delta\delta^{13}C$.

*Code and data availability.* The model and model output can be made available upon request to the main author.

*Author contribution.* CH provided the HAMOCC2s model. AM and CH designed the model experiments and AM carried them out. AM analysed the results and prepared the manuscript, with feedback and supervision from JS and CH.

*Competing interests.* The authors declare that they have no conflict of interest.

*Acknowledgements.* The authors would like to thank two anonymous reviewers for their constructive and helpful comments, which improved this manuscript. This study is a contribution to the project "Earth system modelling of climate variations in the Anthropocene" (EVA; grant no. 229771) as well as the project "Overturning circulation and its implications for global carbon cycle in coupled models" (ORGANIC; grant no. 239965) which are both funded by the Research Council of Norway. This is a contribution to the Bjerknes Centre for Climate Research (Bergen, Norway). Storage resources were provided by the

Norwegian storage infrastructure of Sigma2 (NorStore project ns2980k). Anne Morée is grateful for PhD funding through the Faculty for Mathematics and Natural Sciences of the University of Bergen and the Meltzer Foundation. Christoph Heinze acknowledges sabbatical support from the Faculty for Mathematics and Natural Sciences of the University of Bergen.

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

| Experiment | Experiment setup |
| --- | --- |
| Gas fast | $CO_2$ gas exchange rate * 4 |
| Gas slow | $CO_2$ gas exchange rate / 4 |
| Efficient biological pump | POC sinking rate doubled to 6m/d |
| Inefficient biological pump | POC sinking rate halved to 1.5m/d |
| $V_{max}$ | High nutrient uptake rate (control*5) in the Southern Ocean |
| Ice large | Southern Ocean sea ice cover south of 50° S |
| Ice small | Southern Ocean sea ice cover south of 70° S |

**Table 1 Description of the sensitivity experiments. The sensitivity experiments on the $CO_2$ gas exchange rate and the biological pump have been done twice, once for the Global Ocean and once only making changes in the Southern Ocean (south of 45° S).**

| | Global experiments | | SO-only experiments | |
|---|---|---|---|---|
| | $p\mathrm{CO_2}^{atm}$ | $\delta^{13}\mathrm{C}^{atm}$ | $p\mathrm{CO_2}^{atm}$ | $\delta^{13}\mathrm{C}^{atm}$ |
| *Control* | 279 | -6,5 | - | |
| *Gas exchange* | | | | |
| *Fast* | 275 | -6,8 | 278 | -7,0 |
| *Slow* | 289 | -6,3 | 283 | -6,2 |
| *Biological pump* | | | | |
| *POC: Efficient* | 256 | -6,3 | 275 | -6,5 |
| *POC: Inefficient* | 292 | -6,7 | 282 | -6,5 |
| *$V_{max}$* | - | | 235 | -6,1 |
| *Ice* | | | | |
| *Large* | - | | 287 | -6,2 |
| *Small* | - | | 272 | -6,6 |

**Table 2 Results of $p\mathrm{CO_2}^{atm}$ [ppm] and $\delta^{13}\mathrm{C}^{atm}$ [‰] for all sensitivity experiments.**

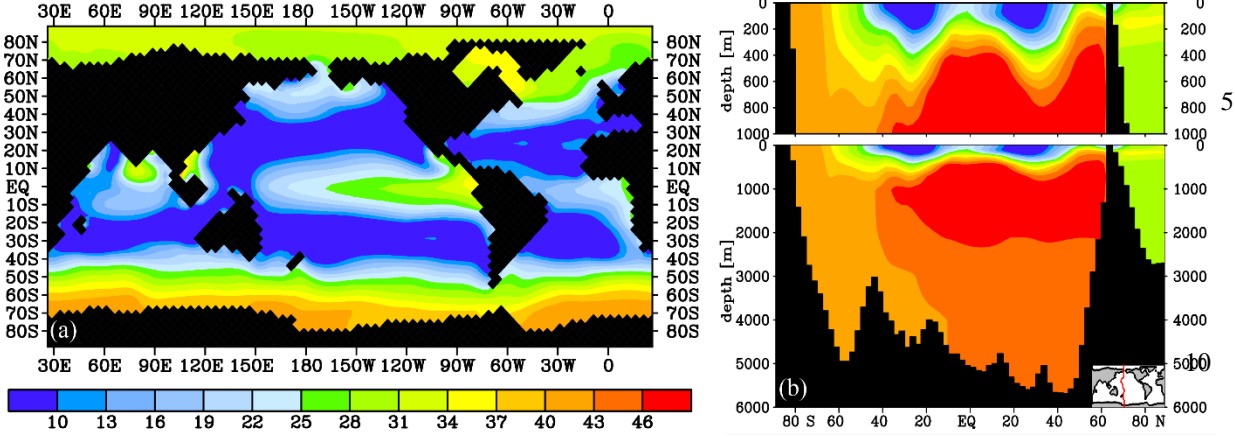

**Figure 1 Modelled $\delta^{13}C$ of DIC [‰] distribution for the model control run: (a) $\delta^{13}C$ at 25 m depth, (b) Pacific transect of $\delta^{13}C$, (c) zonal transect of $\delta^{13}C$ at 26° S, and (d) Atlantic transect of $\delta^{13}C$.**

**Figure 2 $\delta^{13}C_{bio}^{perc}$, the contribution of biology to the local $\delta^{13}C$ signal [%], as calculated using Eq. (4) at (a) 25 m depth and (b) a Pacific transect. The remainder of 100 % is attributed to air-sea gas exchange. The $\delta^{13}C_{bio}$ and $\delta^{13}C_{AS}$ values in ‰ are very similar to the values found by Schmittner et al. (2013).**

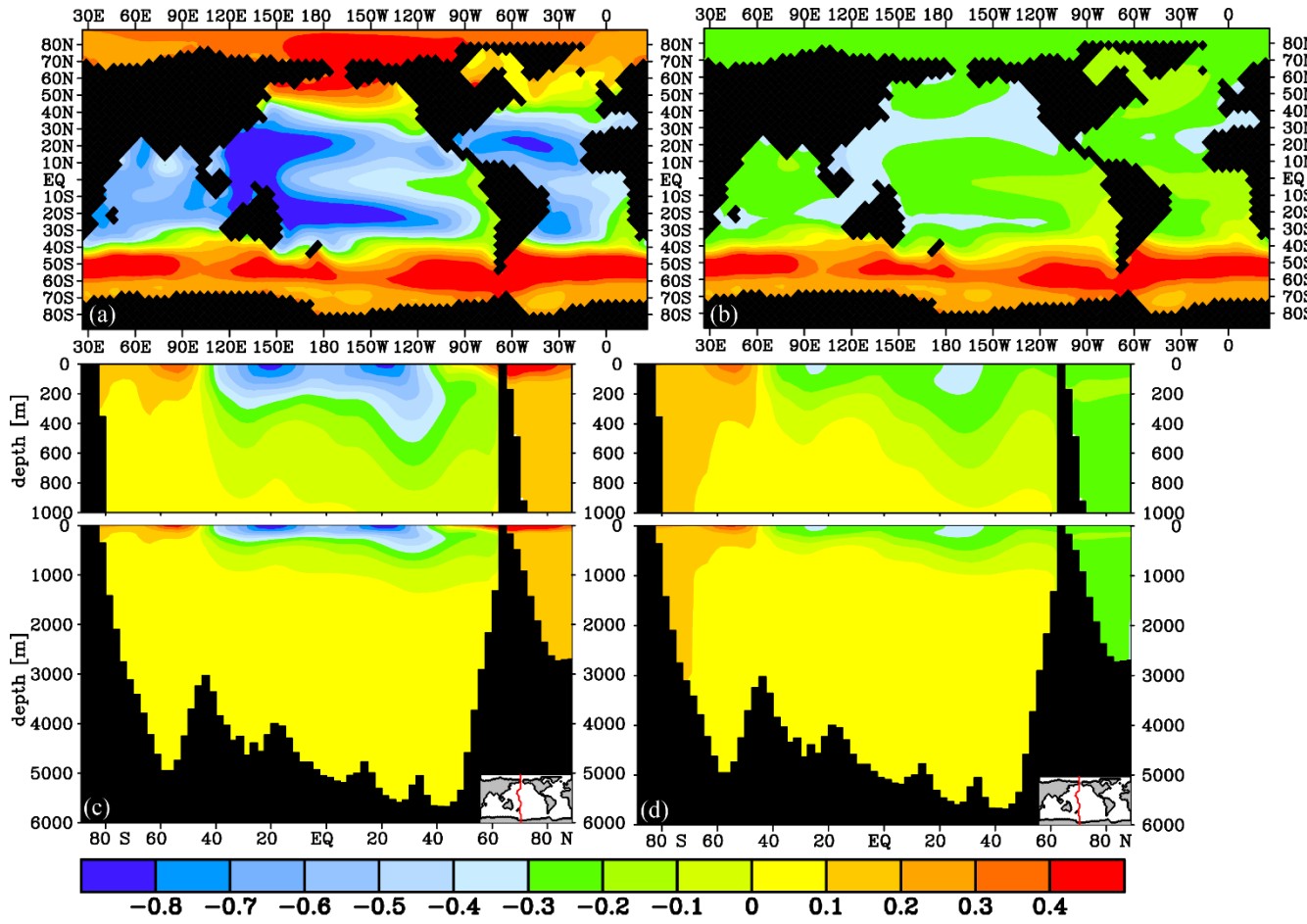

**Figure 3** δ¹³C of DIC [‰] difference after 2000 years for the fast gas exchange experiments (experiment – control): global experiments (a) and (c) and SO-only experiments (b) and (d), at 25 m depth (a) and (b) and as a Pacific transect of δ¹³C difference (c) and (d).

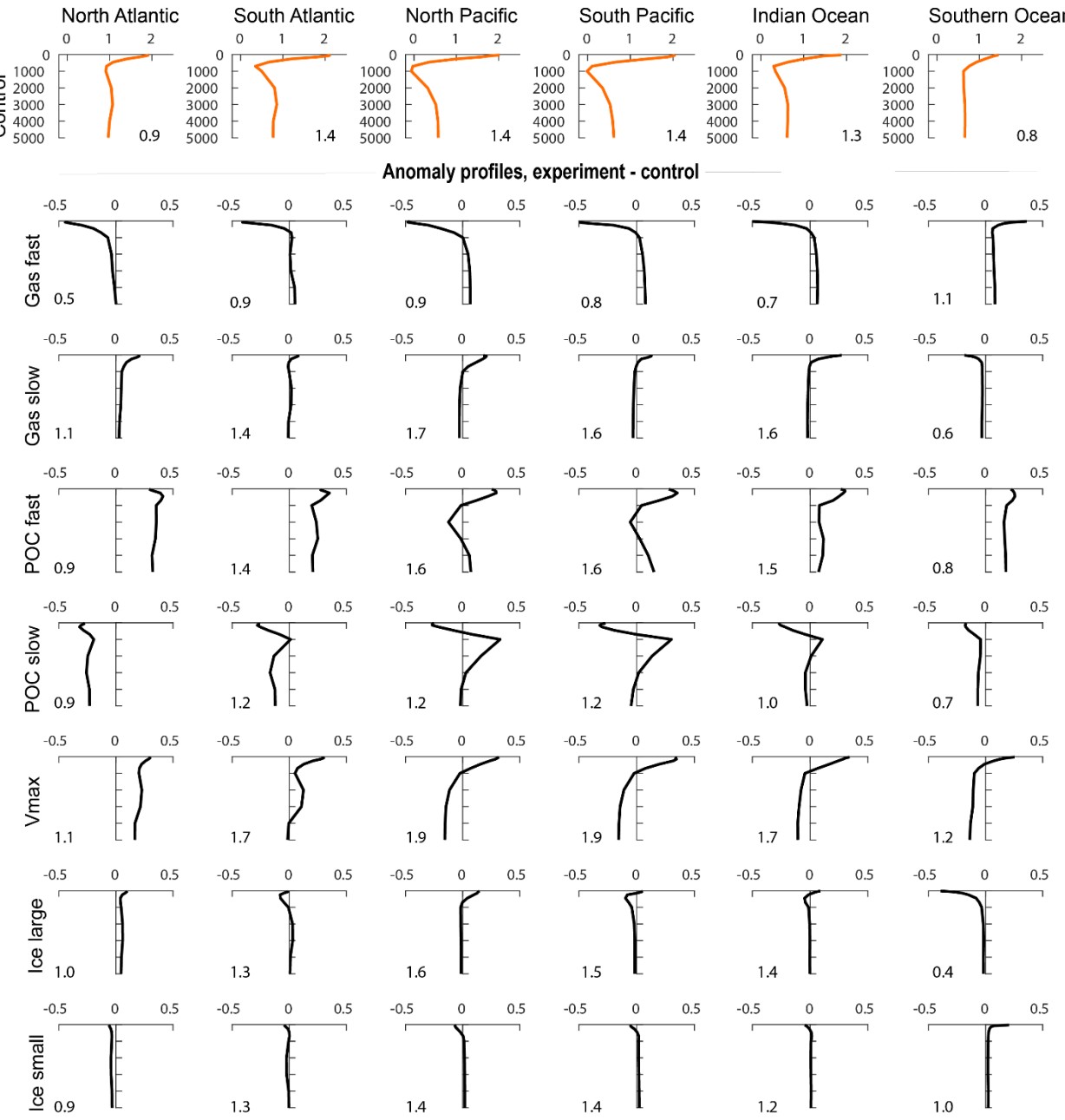

**Figure 4 Volume-weighted basin mean anomaly profiles of δ¹³C after 2000 years, with respect to the control profiles (upper row). Δδ¹³C denoted per basin in the lower right (control) and lower left (sensitivity experiments) corner of each subgraph. Results are presented for the global gas exchange and POC sinking experiments. Basin extent is visualised in Fig. S8.**

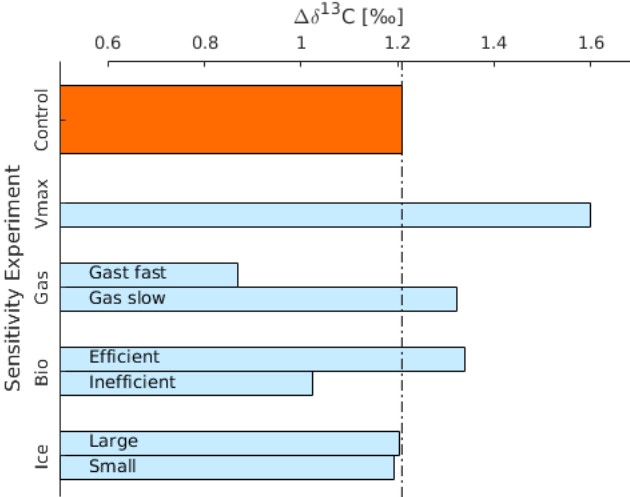

**Figure 5 Global mean Δδ¹³C after 2000 years for the different sensitivity experiments (Table 1). 'Bio Efficient' represents the high POC sinking rate experiment, 'Bio Inefficient' the slow POC sinking rate experiment. The results for the Southern Ocean-only experiments (Sect. 2) are described in the text.**

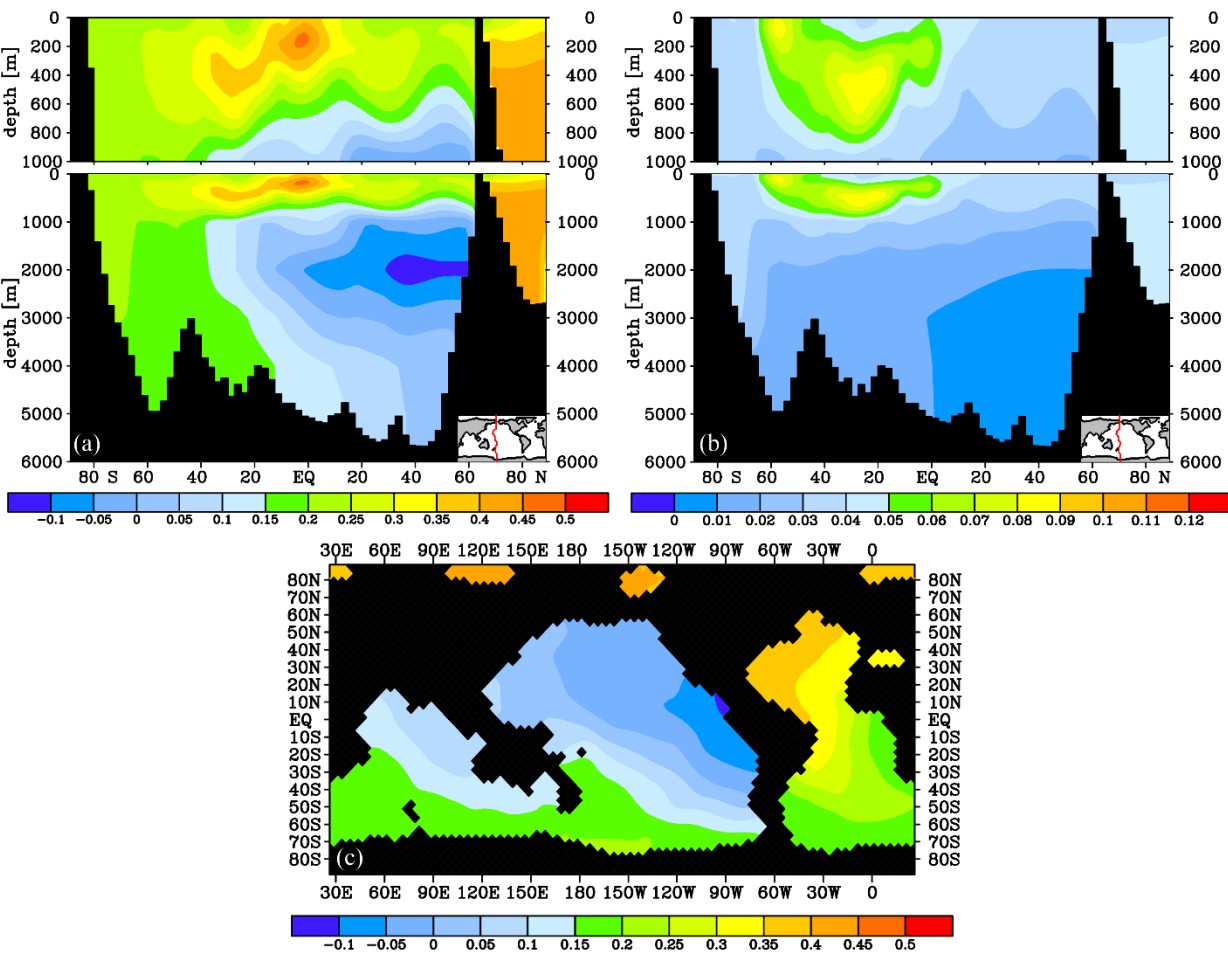

**Figure 6** $\delta^{13}$C of DIC [‰] difference after 2000 years for the POC sinking rate experiments (experiment – control): (a) the global efficient biological pump (high POC sinking rate) experiment for a Pacific transect and (b) the SO-only efficient biological pump experiment for a Pacific transect and (c) the global efficient biological pump experiment at 3000m depth. Note the different scales.

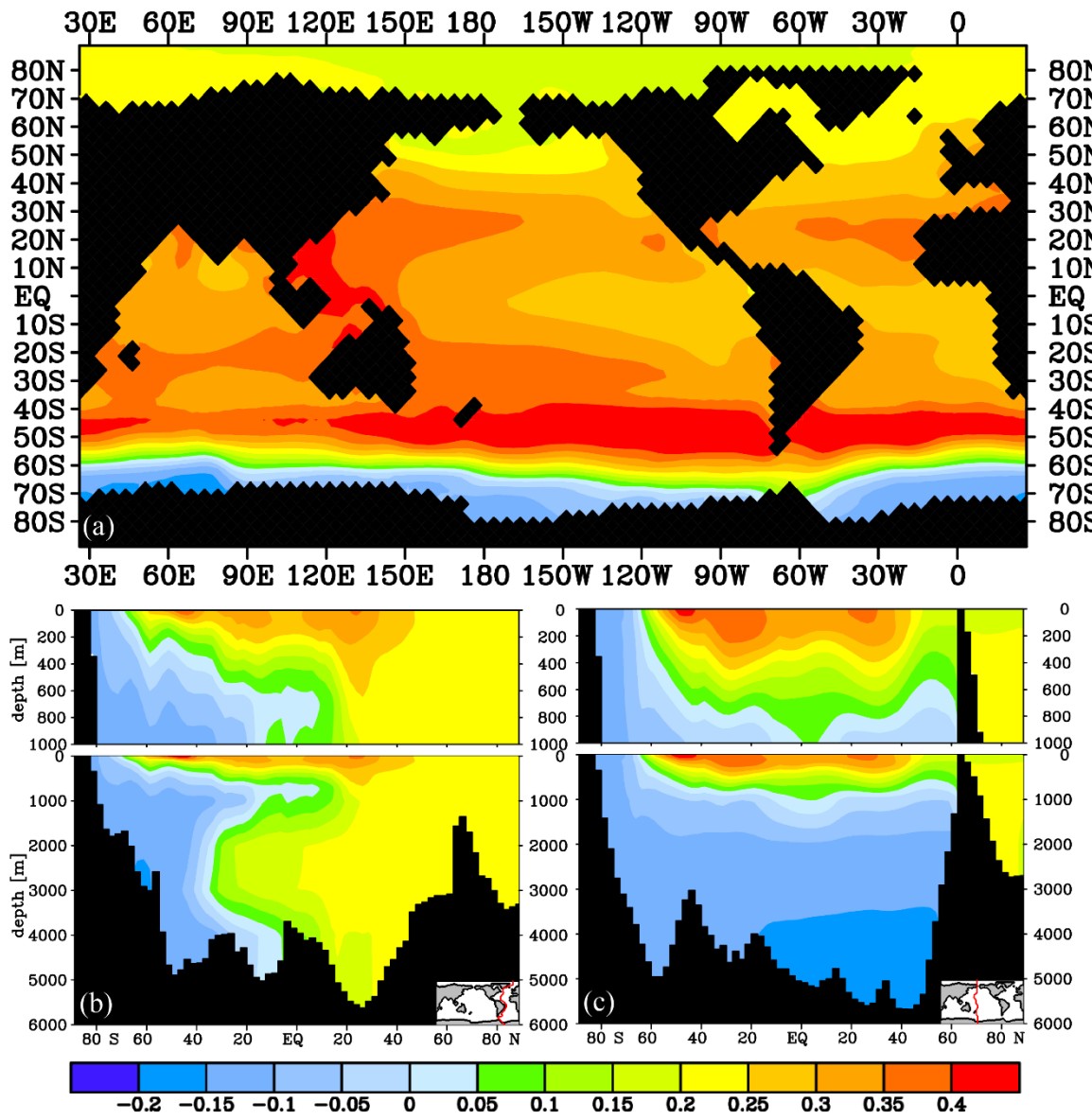

**Figure 7** δ13C of DIC [‰] difference after 2000 years for the Vmax nutrient depletion experiment (experiment – control): (a) at 25m depth and for (b) an Atlantic transect and (c) a Pacific transect.

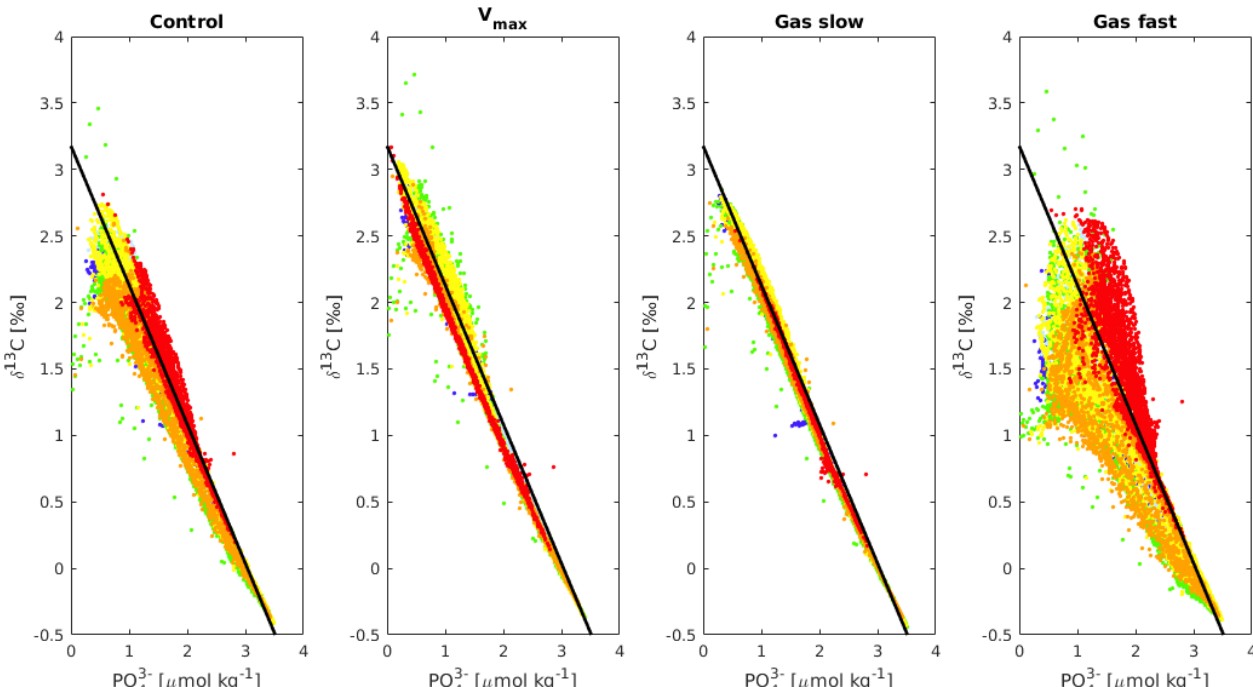

**Figure 8** $\delta^{13}C$ versus $PO_4^{3-}$ for the control, $V_{max}$, 'Gas slow' and 'Gas fast' experiments for all ocean basins except the Nordic Seas (i.e. basins A to F in Fig. S8). Dark blue = A / North Atlantic, Light blue = B / South Atlantic, Red = C / Southern Ocean, Yellow = D / South Pacific, Green = E / North Pacific, Orange = F / Indian Ocean. The black line is the $\delta^{13}C_{bio}=3.18-1.05* PO_4^{3-}$ relationship, i.e. the relationship between $\delta^{13}C$ and $PO_4^{3-}$ expected if only biology affected $\delta^{13}C$ (Sect. 2). Deviations from the black line represent the relative importance of air-sea gas exchange as compared to biology for $\delta^{13}C$.

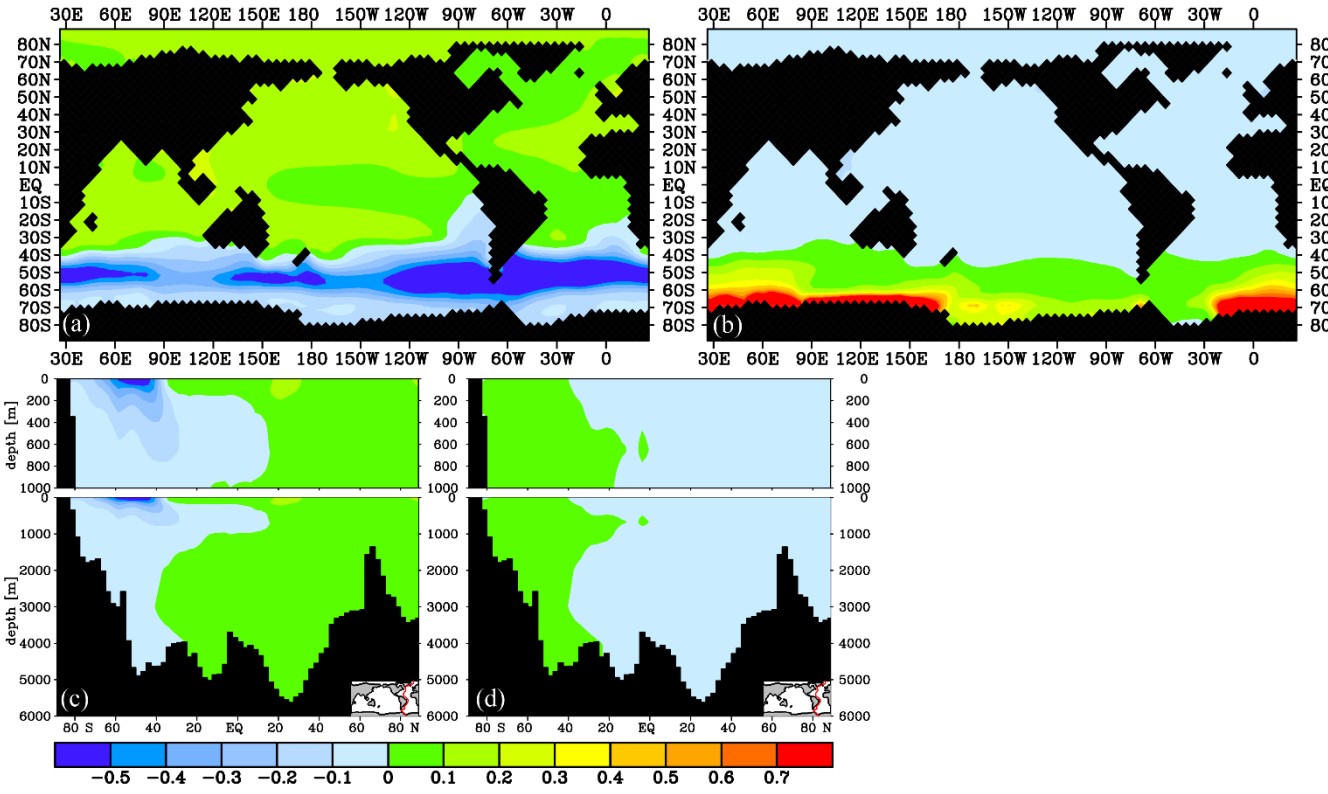

**Figure 9** δ13C of DIC [‰] difference after 2000 years for the Antarctic sea ice cover experiments (experiment – control): The effect of a large (a, c) and small (b, d) Antarctic sea ice cover, for 25 m depth (a, b) and an Atlantic transect (c, d).

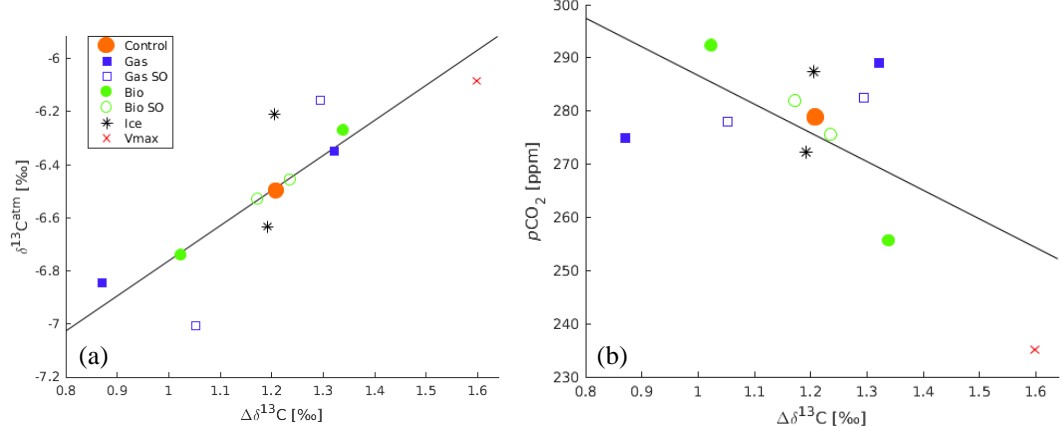

**Figure 10 Relationships between global mean Δδ¹³C, δ¹³Cᵃᵗᵐ and *p*CO₂ᵃᵗᵐ. (a) Global mean Δδ¹³C versus δ¹³Cᵃᵗᵐ of the different sensitivity experiments. R² of the best-fit line is 0.71, and the line is described by y=1.3x-8.1 (b) Global mean Δδ¹³C versus *p*CO₂ᵃᵗᵐ of the different sensitivity experiments. R² of the best-fit line is 0.39, and the line is described by y=-54x+341.**