# Peer review of "Southern Ocean controls of the vertical marine $\delta^{13}C$ gradient – a modelling study"

_Biogeosciences, 2018_

## Referee Comment (RC1) · Anonymous Referee #1 · 19 Mar 2018

Review of Morée et al.,

Morée et al., perform a suite of sensitivity experiments with the ocean biogeochemistry model HAMOCC2s to test the impact of changes in biogeochemical and physical parameters on atmospheric and oceanic d13C. The processes assessed are changes in air-sea gas exchange, POC sinking, Southern Ocean nutrient utilization and sea-ice extent. The end goal is to gain a better understanding of the processes leading to past changes in the vertical gradient of oceanic d13C. It is an interesting process study worth publishing in Biogeosciences. My main comment is that a few processes are studied, with the minimum information given on each of the experiments. I feel that some additional information is needed to fully assess and understand the results. Please find below some specific comments.

1) Modelling study in the context of paleoproxy data:

The motivation behind the study is to better understand variations in oceanic d13C as measured in foraminiferas. This is discussed in the context of the two site-specific studies: Charles et al., (2010) and Ziegler et al., (2013), comparing mid-depth (400m and 1500m) to deep d13C in the Southern Ocean as well as the global study of Oliver et al., (2010). But all the discussion stays very vague and qualitative with "increased/decreased" vertical gradients over "glacial/interglacial" timescales and mostly "globally averaged" for the numerical experiments. This induces some relatively vague conclusions such as in the abstract L. 17-18, or p12 L. 20-25.

This is also true in section 3.4. In addition, in that section results of Charles et al. (2010) and Ziegler et al., (2013) are discussed in a bit more detailed but they are compared to the simulated mean vertical d13C gradient, which is defined as d13Csurface-d13Cdeep, where d13Csurface and d13Cdeep respectively represent mean d13C for depths above and below 250m (please note that the "deep" ocean cannot be defined as the area below 250m depth). This is however different to Ziegler et al., who compare ~400m depth to the deep ocean (~3000m), and Charles et al., (2010) who compare cores at ~1200m and ~4600m.

In general, wouldn't it make sense to show vertical profiles of globally average or basin average Dd13C (d13C at depth compared to d13C averaged over the first 250m)? Such a figure could replace Figure 4 and add a bit more information about the processes at play.

2) Air sea gas exchange experiments: I find the results quite surprising. A pCO2 increase and d13CO2 decrease for fast gas exchange make sense, but a pCO2 increase for a slow gas exchange is surprising. There are no graphs shown for the slow gas exchange case, so it is hard to judge.

3) POC sinking rate: P7, L.20-21: As POC sinking rate increases, the decrease in air-sea gas exchange is most likely due to a reduced advection/mixing of carbon rich

waters into the mixed layer. P7, L.28 it is stated that marine d13C increases overall when POC sinking rates are high. Since d13Catm increases under high POC sinking rates, it seems surprising that marine d13C would also overall increase. . .

In fact, the limited negative d13C anomalies shown in Figure 5 are surprising. Is there a strong increase in organic carbon burial? Would it then make sense to show the transient changes? I am not sure about L. 33-34 p7: the difference in between the global change in POC and SO only change in POC export could only be due to difference in the area to which the forcing is applied, but might not be specific to SO. When applied globally, there is a significant impact on global export production as well as marine and atmospheric d13C. The SO is a relatively small area of the ocean, so changes applied to that region only can be easily compensated.

Results could be discussed with respect to previous experiments performed with the Bern3D and looking at the influence of the remineralization depth on atmospheric CO2 and d13C (e.g. Roth et al., 2014 Earth system dynamics and Menviel et al., 2012, Quaternary Science Reviews).

4) Vmax: It is quite surprising that d13Catm decreases when nutrient utilization increases. P8, L. 27: I doubt the correct reason for the surface negative d13C anomaly is put forward. Maps of changes in export production and nutrients could be added to better understand the model response. If the nutrient advection to the surface of regions outside of SO is reduced, then so should be the advection of carbon rich - 13C depleted waters. This is also consistent with the significant atmospheric CO2 reduction, but the d13CO2 is more surprising. The change in nutrient utilization in the Southern Ocean should be given, as well as control and perturbed surface nutrient content.

5) Sea-ice: Legend of Figure S4 needs additional information

6) Hasted conclusions: The vertical gradient of d13C is stated to vary by no more than 0.5 permil. But it should be noted that this includes the full range of anomalies

obtained: from much lower to much higher than the control state . For example, the maximum changes in vertical d13C gradient are obtained for Vmax (∼+0.2 permil) and a fast gas exchange (∼-0.25 permil), thus leading to ∼0.5 permil change. It would be more appropriate to say that the maximum variation of each parameter leads to a ∼0.25 permil change in vertical d13C gradient, as the pre-industrial control state is an interglacial state.

Section 3.4., p10: very broad statements are made with respect to the impact of changes in ocean circulation on d13C L. 17-18 and L. 20-27. These statements do not rely on any quantitative work on the impact of changes in ocean circulation on oceanic d13C. The authors could for example consider looking at Menviel et al., 2015 (Global Biogeochemical Cycles) to have a better estimate of the impact of ocean circulation changes on d13C. L. 21 to 23 are particularly unjustified because the rate of change of d13C resulting from both biogeochemical changes and oceanic circulation are not studied here.

L. 14-15, p 12: I don't think that the results shown here indicate that the changes in pCO2 and d13Catm are dependent on the location of the sea-ice edge, nor that sea-ice has a strong impact on atmospheric or oceanic d13C.

Minor points and typos:

Throughout the text, please write "biogeochemical" without parentheses.

P2, L. 3: "Air-sea" P6- L.2, please rephrase P6, L. 29: Please remove "In the ocean," Figure 8: y axis of second plot should read "pCO2 (ppm)"

---

## Referee Comment (RC2) · Anonymous Referee #2 · 9 Apr 2018

**1   General comments**

**1.1   Appreciation of the manuscript**

Anne L. Morée and co-authors use the HAMburg Ocean Carbon Cycle Model in its configuration for long-term simulations, HAMOCC2s (Heinze and Maier-Reimer, 1999). The authors report the results of four sensitivity experiments (actually four plus two, as two out of the four are run in duplicate, once for the global ocean and once for the Southern Ocean) to analyse (1) the effect of variations of the air-sea exchange parameters, (2) the sequestration efficiency of the organic pump via changed particulate organic carbon (POC) sinking rates, (3) the sequestration efficiency of the organic pump

via increased nutrient utilisation efficiency, (4) changing sea-ice cover on atmospheric $pCO_2$, $\delta^{13}C$ of atmospheric $CO_2$ and $\delta^{13}C$ in the ocean, and more specifically on the global mean vertical gradient of $\delta^{13}C$ in dissolved inorganic carbon (DIC), quantified as the difference between DIC $\delta^{13}C$ in the surface and the deep ocean, denoted $\Delta\delta^{13}C$.

Upon reading the abstract of this paper I got really excited. Carbon isotopes are a particularly useful tool for studying carbon cycling between the different spheres that make it up. They have been used for a long time for this purpose, but over the past years a wealth of new data have been published and more and more comprehensive global data compilations have become available. The time is thus ready to re-assess the different mechanisms with a model that offers an excellent balance between the comprehensiveness of the processes taken into account and their complexity and execution time, so that meaningful simulation experiments can be carried out for time scales of tens to hundreds of millennia.

The manuscript itself, however, did unfortunately not meet my expectations, far from.

The language used, albeit generally fluent, seriously lacks precision and is rather colloquial. As an example, we repeatedly read that $\delta^{13}C$ is depleted or enriched. It is of course DIC that is depleted or enriched in $^{13}C$. A relative deviation — such as $\delta^{13}C$ — cannot be enriched or depleted; it can be high(er), greater or low(er), decreased or increased.

The literature review is very poor; the same holds for comparison of the results obtained here to those of previous studies. Many important previous studies that called upon carbon isotopes for the study of glacial-interglacial carbon cycle changes are not cited (see below for details). $\Delta\delta^{13}C$, the proxy that is central to the paper really ought to be introduced with a more solid background. It was probably first used by Broecker (1982), at the very beginning of the "gold rush" time of the glacial-interglacial atmospheric $CO_2$

problem studies (1980s). It was then used as a proxy for glacial-interglacial $pCO_2$ variations, later fell out of favour, but has resurfaced over the past few years. One thing that would be important to emphasize here is, that it evolved in time: during those early stages, $\Delta\delta^{13}C$ stood for the difference between $\delta^{13}C$ in the deep and the surface ocean DIC. In the recent studies (e.g., Ziegler et al. (2013)), it now most often stands for the difference between $\delta^{13}C$ of DIC in the deep sea and intermediate-depth (typically 400 m).

The model description is incomplete. The processes that are relevant for the study are not described at all, only a reference to a previous paper is given.

The experimental design leaves quite a number of questions open: the duration of the simulation experiments is only 2000 years. The separation between surface and deep ocean waters is questionable and as it obviously has an important influence on the results, the side-effects of this choice should have been discussed.

Not all of the figures are reader-friendly: on Figure 5, e.g., readers are expected to visually extract $\Delta\delta^{13}C$ from latitude-depth transects of $\delta^{13}C$ by first averaging the topmost 250 m, then the depths below and to subtract both averages from each other.

As a consequence, I cannot recommend this manuscript for publication in *Biogeosciences* at this stage. It should nevertheless be possible to reconsider it after a major revision and I strongly encourage the authors to prepare a version that remedies to all the shortcomings mentioned here. Please provide us with a better description of what is done, how it is done and why it is done that way. The study deals with an interesting and timely subject. The biogeochemical model at hand perfectly fits the needs. Please take full advantage of the possibilities it offers!

[Figure]

**2 Specific comments**

**2.1 Literature**

Since the $\Delta\delta^{13}$C proxy has been in usage for more than 35 years, there is a wealth of studies that are available. They range from data-oriented studies to model-based studies, covering very similar approaches as done here. Only very few of them are cited in the manuscript and it is not entirely clear for what reasons they are included and others are excluded. There are more than 20 papers that come to my mind right away in this framework and that have not been considered in the literature review and the discussion of this paper

1. Broecker (1982)

2. Boyle (1988)

3. Keir (1988)

4. Oppo and Fairbanks (1990)

5. Oppo et al. (1990)

6. Keir (1991)

7. Mulitza et al. (1998)

8. Toggweiler (1999)

9. Flower et al. (2000)

10. Murnane and Sarmiento (2000)

[Figure]

11. Brovkin et al. (2002)

12. Hodell et al. (2003)

13. Köhler et al. (2005)

14. Köhler et al. (2010)

15. Lisiecki (2010)

16. Lourantou et al. (2010)

17. Tschumi et al. (2011)

18. Brovkin et al. (2012)

19. Menviel et al. (2012)

20. Peterson et al. (2014)

21. Menviel et al. (2015)

22. Eggleston et al. (2016)

23. Lear et al. (2016)

24. Menviel et al. (2017)

Please do not get me wrong: I do not expect all of these papers to be cited. However, even this "out-of-the-mind" list is simply so long (and still far from exhaustive) that it is incomprehensible that none of these studies has been cited or taken into account for the purpose of discussing the results.
**2.2  Model description and experimental design**

**2.2.1  Model description is insufficient**

The model description given in the paper neither allows to reproduce the results reported here without a lot of guesswork, nor does it allow to fully understand the results. The provided description is in some instances even confusing: on page 3 (lines 13–14), we read that "POC is carried as a tracer as well as transported downwards according to an exponential penetration depth and constant settling velocity, [...]". The exponential penetration profile and the constant settling velocity are of course not independent of each other. In the original technical reference to HAMOCC2s (Heinze and Maier-Reimer, 1999) – not cited in the manuscript – we read that "The vertical flux of biogenic particulate matter is parametrised through exponential redistribution profiles which implicitly include both sinking velocity and re-dissolution rate." This is not the same! It is quite easy to establish that the characteristic length scale in the exponential profile is equal to $\omega/k$, if the (constant) settling velocity is denoted $\omega$ and POC respiration is assumed to follow first order kinetics with a rate constant $k$. Since one of the experiments involves changes of the settling velocity, the adopted parametrisations must be correctly described.

**2.2.2  Model spin-up procedure**

The description of the model spin-up procedure lacks important details. We only read that "[...]  a fixed weathering input is used to tune the ocean inventories to values comparable to the observations." (page 3, line 24). On the basis of what quantitative constraints is this weathering flux determined? Are there separate fluxes

- for nutrients (phosphate)? – which would be necessary if organic matter is buried

in the sediment together with the nutrients they lock up

- for DIC and alkalinity? – which would have to be separated if organic matter and carbonate are buried in the sediment

- for dissolved silica? – opal is also included in the model

- for $^{13}$C? – what is the $\delta^{13}$C signature of the DIC flux?

A decent model description would have answered half of the questions already... To what extent are the mismatches in the deep-ocean $\delta^{13}$C and PO$_4$ concentration resulting from this spin-up? I would expect that they go together with global $^{13}$C and PO$_4$ inventory mismatches as well, which, according to the description given here, are constraints.

**2.2.3 Sensitivity experiment duration**

The quality of the spin-up experiment is well quantified (residual drifts etc.). Unfortunately, nothing similar is reported for the sensitivity experiments. Readers are only told that these have been run for 2000 yr with the steady-state control run as initial condition. The strength of the model design for allowing long-term simulation experiments is initially emphasized (page 3, lines 10–11), a 110 000 yr spin-up run is carried out, and then the core experiments for the paper are run over a comparatively short duration of 2000 years only. For some of the perturbations (e.g., POC penetration depth changes...), the model carbon cycle is still in the transient phase 2000 years after the onset of the perturbation. The choice of such short simulation experiments is thus rather incomprehensible.

**2.3 Analysis of the results**

**2.3.1 Up- and downward fluxes, equilibrium $\delta^{13}$C**

Analysis of the results involves up- and downward fluxes $F_{\text{up}}$ and $F_{\text{down}}$: how are these obtained? To my best knowledge, it is only the net exchange flux $F_{\text{net}}$ which is proportional to the $p_{CO_2}$ difference between the surface water and the overlying atmosphere that can be calculated.

The equilibrium $\delta^{13}$C ($\delta^{13}$C$_{\text{eq}}$, first mentioned on page 7 at line 1) is not defined and an explanation how this is calculated is missing as well.

**2.3.2 Separation between surface and deep realms**

In this study, the ocean is simply partitioned into a surface part, which encompasses the water masses above a 250 m depth horizon, and a deep part for the rest. No justification or explanation regarding this choice are given.

First of all, it is a choice that leads to complications. Information gathered from previous publications based up HAMOCC2s (Heinze et al., 1999, 2016) indicates that the eleven-layer configuration has no layer interface at 250 m depth, but a layer centred on 250 m depth. A more natural separation would be located at layer boundaries.

Secondly, this choice is critical as it controls the results of the study to a large extent extent. At 250 m depth, the depth profile of DIC $\delta^{13}$C is generally rapidly decreasing (see e.g., Kroopnick (1985), but this should also be visible from the model results). Accordingly, the average surface ocean $\delta^{13}$C will be strongly biased towards lower values and the deep ocean slightly towards higher values. As a consequence, the amplitude of the vertical gradient, $|\Delta\delta^{13}C|$, is thus systematically underestimated. I think that surface ocean $\delta^{13}$C would more conveniently be taken from the surface layer down to 50 or 112.5 m depth (these are layer boundary depths in the 11-layer HAMOCC2s configuration, or even regionally variable in case information on the local mixed-layer depth would be available), and the deep ocean from the 1500 or the 2500 m depth horizons down to the sea floor.

In any case care must be taken in the model-data comparison to make sure that surface-to-deep model gradients are compared to surface-to-deep data gradients and not to intermediate-to-deep data gradients.

**2.3.3  Regionalization**

$\Delta\delta^{13}$C results are only shown in the global mean. The three-dimensional HAMOCC2s should allow for a finer analysis than that. In the text, regional $\Delta\delta^{13}$C outcomes are sometimes mentioned, but it would be useful to have these results reported graphically as well, at least for basins or sub-basins (e.g., North Atlantic, South Atlantic, North Pacific, South Pacific, Southern Ocean). Figure 4 could be easily adapted to show such more regionalized values in a useful and expressive way.

**2.4  Discussion shortcomings**

Parts of the discussion are rather confusing. Section 3.4 is one of them. On one hand, we read that "The idealised and large perturbations [. . . ] show that mean $\Delta\delta^{13}$C varies no more than 0.5‰" on the other hand that "[the] reconstructed intra-millennial variability in $\Delta\delta^{13}$C could be driven more by changes in the biogeochemical state than by changes in ocean circulation because (bio)geochemical changes might occur more rapidly than whole-ocean circulation changes." Are *large* and *whole-ocean* changes in the biogeochemical state of the ocean really that more realistic on the time scales of a

few millennia than circulation changes?

At the latest from page 10, lines 29–30 on it is not clear any more which conclusions to draw from this study. Readers that have come this far will have seen the discussion revolve around SO $\Delta\delta^{13}$C in several instances, to learn now that, except for the North Atlantic, "data are too sparse to get a coherent picture of $\Delta\delta^{13}$C variations". Previously we have been shown that in the North Atlantic the deep-sea $\delta^{13}$C is mainly controlled by the air-sea exchange $\delta^{13}$C.

**3  Technical comments**

Throughout the paper: please check precision of the language. . .

**Page 1, lines 3 and 27–28**: "The standardised $^{13}$C isotope, $\delta^{13}$C, [. . .]": I have never seen this denomination in the peer-reviewed literature before. $\delta^{13}$C expresses the molar $^{13}$C/$^{12}$C ratio of a sample in terms of its relative deviation from the ratio in a standard (initially PDB, now VPDB), generally expressed in permille. The references provided for this "standardised $^{13}$C isotope" are incomprehensible: Stenström et al. (2011) is a non peer-reviewed internal university report, Stuiver and Polach (1977) deals with radiocarbon. It should be straightforward to find an appropriate textbook reference for $\delta^{13}$C.

**Page 2, line 3**: "air-se" should read "air-sea"

**Page 2, line 6**: "10s" should be written out as "tens"

**Page 2, line 24**: the study by Shackleton and Pisias (1985) absolutely needs to be cited here alongside Charles et al. (2010) and Oliver et al. (2010).

**Page 3, line 17**: should "HAMOCC2" not read "HAMOCC2s"?

**Page 3, line 23**: should "HAMOCC2" not read "HAMOCC2s" again?

**Page 4, line 23**: "Eide (2017)": 2017a or 2017b?

**Page 4, line 23**: on the basis of the provided mean values, the intercept of the $\delta^{13}$C:PO$_4$ relationship is 3.27733, which would normally be rounded to 3.3, not to 3.4. Please check the numbers.

**Page 5, line 12**: "The modelled global POC production is [. . . ]": I guess this is the *new* or the *export* production – please clarify!

**Page 5, lines 26–27**: "[. . . ] with the exception of the Arctic Ocean where no POC production is modelled due to the sea ice cover [. . . ]": elsewhere in the paper we read that the sea-ice cover also isolates the surface ocean with respect to air-sea exchange. Does the partitioning into $\delta^{13}$C$_{bio}^{perc}$ and $\delta^{13}$C$_{atm}^{perc}$ make sense in ice-covered regions?

**Page 6, line 4**: "change more than" should read "change by more than"

**Page 6, line 9**: "[. . . ] due to the fact that $^{12}$C needs to speciate [. . . ]": this does not make sense. $^{12}$C can only equilibrate at the same time as $^{13}$C – there are only the two of them. $^{12}$C should probably be corrected to DIC or CO$_{2\,(aq)}$.

**Page 6, line 19**: "[. . . ] 22% of the global ocean area [. . . ]": does this include the ice-covered parts of the SO? – please specify

**Page 6, line 21**: "$F_u$" should read "$F_{up}$"

**Page 6, line 22**: "$F_d$" should read "$F_{down}$"

**Page 6, line 30**: "[. . . ] lowers the surface ocean $\delta^{13}$C $-0.2$ to $-0.9\,‰$ in the lower latitudes [. . . ]" should read "[. . . ] lowers the surface ocean $\delta^{13}$C by $-0.2$ to $-0.9\,‰$ at the lower latitudes [. . . ]"

**Page 6, line 31**: "in high latitudes" should read "at high latitudes"

**Page 6, line 31–32**: "These results indicate the sign of the thermodynamic $\delta^{13}$C disequilibrium between surface ocean and atmosphere." – this sentence does not make sense, please reformulate.

**Page 7, line 7**: please add the ‰ sign to the 0.65 and the 1.00

**Page 7, lines 16–17**: "A more efficient biological pump [. . . ] leads to a loss of carbon to the sediments, which dominates the effects on $p\mathrm{CO_2^{atm}}$ and $\delta^{13}\mathrm{C^{atm}}$.": after 2000 years of simulation these effects have certainly not yet developed to their full strength.

**Page 7, lines 24–25**: "remineralisation horizon": a horizon depicts, in my understanding, a surface or a narrow zone, such as the calcite saturation horizon. I am not aware of the existence of a POC remineralisation horizon (and not even a carbonate remineralization horizon). Please rewrite.

**Page 7, lines 29–30**: "When reducing the biological pump efficiency both remineralisation and POC production are confined to the surface ocean.": as far as I know

HAMOCC2s, the POC production is always confined to the surface and the remineral-isation is taking place in subsurface intermediate and greater depths. Would "With a lower POC sinking rate, the remineralisation is confined to shallower depths." not be more correct?

**Page 8, lines 5–9**: Figure 5 which is referred to here, depicts $\delta^{13}C$ and DIC anomalies with respect to the control run. Having readers derive information about $\Delta\delta^{13}C$ from that figure is really asking too much. Why not provide the latitudinal evolution of the $\Delta\delta^{13}C$ alongside? This would be a straightforward line plot.

**Page 9, section 3.3.4**: I would expect that such large ice-cover changes would also lead to circulation changes. A comment on this would be of order, wouldn't it?

**Page 9, line 25**: $\delta^{13}C_{eq}$: see above

**Page 10, lines 23–24**: "Analysis of SO $\Delta\delta^{13}C$ reconstructions from sediment cores at 42°S and 46°S (Charles et al., 2010) shows that there is a strong correlation between these cores and Northern Hemisphere $\Delta\delta^{13}C$ variations." This is not correct. Charles et al. (2010) show that there is a tight correlation between SO $\Delta\delta^{13}C$ and "Northern Hemisphere climate fluctuations"; their paper does not even mention any $\Delta\delta^{13}C$ record outside the SO.

**Figures**: if $\Delta\delta^{13}C$ informations are to be read from a figure, this latter should then also show $\Delta\delta^{13}C$.

**Page 24, Figure 8b**: units for $pCO_2$ on the vertical axis should be ppm or $\mu$atm on the vertical axis, not ‰.

**References**

Boyle, E. A.: The role of vertical chemical fractionation in controlling Late Quaternary atmospheric carbon dioxide, J. Geophys. Res., 93, 15 701–15 714, doi:10.1029/JC093iC12p15701, 1988.

Broecker, W. S.: Ocean chemistry during glacial times, Geochim. Cosmochim. Ac., 46, 1689–1705, doi:10.1016/0016-7037(82)90110-7, 1982.

Brovkin, V., Hofmann, M., Bendtsen, J., and Ganopolski, A.: Ocean biology could control atmospheric $\delta^{13}$C during glacial-interglacial cycle, Geochem., Geophy., Geosy., 3, 1027, doi:10.1029/2001GC000270, 2002.

Brovkin, V., Ganopolski, A., Archer, D., and Munhoven, G.: Glacial $CO_2$ cycle as a succession of key physical and biogeochemical processes, Clim. Past, 8, 251–264, doi:10.5194/cp-8-251-2012, 2012.

Eggleston, S., Schmitt, J., Bereiter, B., Schneider, R., and Fischer, H.: Evolution of the stable carbon isotope composition of atmospheric $CO_2$ over the last glacial cycle, Paleoceanography, 31, 434–452, doi:10.1002/2015PA002874, 2016.

Flower, B. P., Oppo, D. W., McManus, J. F., Venz, K. A., Hodell, D. A., and Cullen, J. L.: North Atlantic Intermediate to Deep Water circulation and chemical stratification during the past 1 Myr, Paleoceanography, 15, 388–403, doi:10.1029/1999PA000430, 2000.

Heinze, C. and Maier-Reimer, E.: The Hamburg Oceanic Carbon Cycle Circulation Model version "HAMOCC2s" for long time integrations, Technical Report 20, Deutsches Klimarechenzentrum, Hamburg (DE), available at https://www.dkrz.de/mms/pdf/reports/ReportNo.20.pdf, 1999.

Heinze, C., Maier-Reimer, E., Winguth, A. M. E., and Archer, D.: A global ocean sediment model for long-term climate studies, Global Biogeochem. Cy., 13, 221–250, doi:10.1029/98GB02812, 1999.

Heinze, C., Hoogakker, B. A. A., and Winguth, A.: Ocean carbon cycling during the past 130 000 years – a pilot study on inverse palaeoclimate record modelling, Clim. Past, 12, 1949–1978, doi:10.5194/cp-12-1949-2016, 2016.

Hodell, D. A., Venz, K. A., Charles, C. D., and Ninnemann, U. S.: Pleistocene vertical carbon isotope and carbonate gradients in the south atlantic sector of the southern ocean, Geochem., Geophy., Geosy., 4, 1004, doi:10.1029/2002GC000367, 2003.

Keir, R. S.: On the Late Pleistocene ocean geochemistry and circulation, Paleoceanography, 3,
413–445, doi:10.1029/PA003i004p00413, 1988.

Keir, R. S.: The effect of vertical nutrient redistribution on surface ocean $\delta^{13}$C, Global Biogeochem. Cy., 5, 351–358, doi:10.1029/91GB01913, 1991.

Köhler, P., Fischer, H., Munhoven, G., and Zeebe, R. E.: Quantitative interpretation of atmospheric carbon records over the last glacial termination, Global Biogeochem. Cy., 19, GB4020, doi:10.1029/2004GB002345, 2005.

Köhler, P., Fischer, H., and Schmitt, J.: Atmospheric $\delta^{13}CO_2$ and its relation to $pCO_2$ and deep ocean $\delta^{13}$C during the late Pleistocene, Paleoceanography, 25, PA1213, doi:10.1029/2008PA001703, 2010.

Kroopnick, P. M.: The distribution of $^{13}$C of $\Sigma CO_2$ in the world oceans, Deep-Sea Res. A, 32, 57–84, doi:10.1016/0198-0149(85)90017-2, 1985.

Lear, C. H., Billups, K., Rickaby, R. E. M., Diester-Haass, L., Mawbey, E. M., and Sosdian, S. M.: Breathing more deeply : Deep ocean carbon storage during the mid-Pleistocene climate transition, Geology, 44, 1035, doi:10.1130/G38636.1, 2016.

Lisiecki, L. E.: A benthic $\delta^{13}$C-based proxy for atmospheric pCO$_2$ over the last 1.5 Myr, Geophys. Res. Lett., 37, L21708, doi:10.1029/2010GL045109, 2010.

Lourantou, A., Lavrič, J. V., Köhler, P., Barnola, J.-M., Paillard, D., Michel, E., Raynaud, D., and Chappellaz, J.: Constraint of the CO$_2$ rise by new atmospheric carbon isotopic measurements during the last deglaciation, Global Biogeochem. Cy., 24, GB2015, doi:10.1029/2009GB003545, 2010.

Menviel, L., Joos, F., and Ritz, S. P.: Simulating atmospheric CO2, $^{13}$C and the marine carbon cycle during the last glacial-interglacial cycle : possible role for a deepening of the mean remineralization depth and an increase in the oceanic nutrient inventory, Quaternary Sci. Rev., 56, 46–68, doi:10.1016/j.quascirev.2012.09.012, 2012.

Menviel, L., Mouchet, A., Meissner, K. J., Joos, F., and England, M. H.: Impact of oceanic circulation changes on atmospheric $\delta^{13}CO_2$, Global Biogeochem. Cy., 29, 1944–1961, doi:10.1002/2015GB005207, 2015.

Menviel, L., Yu, J., Joos, F., Mouchet, A., Meissner, K. J., and England, M. H.: Poorly ventilated deep ocean at the Last Glacial Maximum inferred from carbon isotopes : A data-model comparison study, Paleoceanography, 32, 2–17, doi:10.1002/2016PA003024, 2017.

Mulitza, S., Rühlemann, C., Bickert, T., Hale, W., Pätzold, J., and Wefer, G.: Late Quaternary $\delta^{13}$C gradients and carbonate accumulation in the western equatorial Atlantic, Earth Planet. Sc. Lett., 155, 237–249, doi:10.1016/S0012-821X(98)00012-0, 1998.

Murnane, R. J. and Sarmiento, J. L.: Roles of biology and gas exchange in determining the $\delta^{13}$C distribution in the ocean and the preindustrial gradient in atmospheric $\delta^{13}$C, Global Biogeochem. Cy., 14, 389–405, doi:10.1029/1998GB001071, 2000.

Oppo, D. W. and Fairbanks, R. G.: Atlantic Ocean thermohaline circulation of the last 150,000 years : Relationship to climate and atmospheric $CO_2$, Paleoceanography, 5, 277–288, doi:10.1029/PA005i003p00277, 1990.

Oppo, D. W., Fairbanks, R. G., Gordon, A. L., and Shackleton, N. J.: Late Pleistocene Southern Ocean $\delta^{13}$C variability, Paleoceanography, 5, 43–54, doi:10.1029/PA005i001p00043, 1990.

Peterson, C. D., Lisiecki, L. E., and Stern, J. V.: Deglacial whole-ocean $\delta^{13}$C change estimated from 480 benthic foraminiferal records, Paleoceanography, 29, 549–563, doi:10.1002/2013PA002552, 2014.

Shackleton, N. J. and Pisias, N. G.: Atmospheric carbon dioxide, orbital forcing, and climate, in: The Carbon Cycle and Atmospheric $CO_2$ : Natural Variations Archean to Present, edited by Sundquist, E. T. and Broecker, W. S., vol. 32 of *Geophys. Monogr. Ser.*, pp. 303–317, AGU, Washington, DC, 1985.

Toggweiler, J. R.: Variation of atmospheric $CO_2$ by ventilation of the ocean's deepest water, Paleoceanography, 14, 571–588, doi:10.1029/1999PA900033, 1999.

Tschumi, T., Joos, F., Gehlen, M., and Heinze, C.: Deep ocean ventilation, carbon isotopes, marine sedimentation and the deglacial $CO_2$ rise, Clim. Past, 7, 771–800, doi:10.5194/cp-7-771-2011, 2011.

Ziegler, M., Diz, P., Hall, I. R., and Zahn, R.: Millennial-scale changes in atmospheric $CO_2$ levels linked to the Southern Ocean carbon isotope gradient and dust flux, Nat. Geosci., 6, 457–461, doi:10.1038/NGEO1782, 2013.

---

## Author Comment (AC1) · 1 May 2018

**Author responses to Referee #1**

**Comment 1)**

Modelling study in the context of paleoproxy data: The motivation behind the study is to better understand variations in oceanic δ13C as measured in foraminiferas. This is discussed in the context of the two site-specific studies: Charles et al., (2010) and Ziegler et al., (2013), comparing mid-depth (400m and 1500m) to deep δ13C in the Southern Ocean as well as the global study of Oliver et al., (2010). But all the discussion stays very vague and qualitative with "increased/decreased" vertical gradients over "glacial/interglacial" timescales and mostly "globally averaged" for the numerical experiments. This induces some relatively vague conclusions such as in the abstract L. 17-18, or p12 L. 20-25. This is also true in section 3.4. In addition, in that section results of Charles et al. (2010) and Ziegler et al., (2013) are discussed in a bit more detailed but they are compared to the simulated mean vertical δ13C gradient, which is defined as δ13Csurfaceδ13Cdeep, where δ13Csurface and δ13Cdeep respectively represent mean δ13C for depths above and below 250m (please note that the "deep" ocean cannot be defined as the area below 250m depth). This is however different to Ziegler et al., who compare ∼400m depth to the deep ocean (∼3000m), and Charles et al., (2010) who compare cores at ∼1200m and ∼4600m. In general, wouldn't it make sense to show vertical profiles of globally average or basin average Δδ13C (δ13C at depth compared to δ13C averaged over the first 250m)? Such a figure could replace Figure 4 and add a bit more information about the processes at play.

**Author response to Comment 1)**

Referee #1 kindly makes us aware of the too generalized and qualitative discussion throughout the manuscript. We will address this issue by 1) providing basin-averaged δ13C profiles (as a revision of Figure 4) and 2) redefining deep ocean δ13C as the volume-averaged δ13C below 2500m depth, and 3) adjusting the discussion and conclusions to be more specific and quantitative on a basin scale. In doing so, comparison with the sediment core studies will become more meaningful and the spatial differences in the sensitivity of Δδ13C will be more visible.

**Author's changes in the manuscript in response to Comment 1)**

- Revision of Figure 4 by basin-mean vertical profiles of δ13C for the Southern Ocean, North Pacific, South Pacific, North Atlantic, South Atlantic and Indian ocean for the global sensitivity experiments.
- Redefining deep ocean δ13C and updating the results accordingly
- Updating discussion/abstract/conclusion to be more quantitative and specific and using the revised Figure 4 to discuss the sensitivity of Δδ13C on a basin scale

**Comment 2)**

Air sea gas exchange experiments: I find the results quite surprising. A pCO2 increase and δ13CO2 decrease for fast gas exchange make sense, but a pCO2 increase for a slow gas exchange is surprising. There are no graphs shown for the slow gas exchange case, so it is hard to judge

**Author response to Comment 2)**

The authors agree with Referee #1 that the discussion of *p*CO2 sensitivity to slow gas exchange rates is not explained enough in the current manuscript. As stated in the manuscript, 'pCO2 atm is governed by the transient change in the net air-sea gas exchange flux Fnet, which occurs until a new equilibrium is established'.
In order to explain the slow gas exchange experiment, we will add carbon flux figures at 100 years, when the transient response determines the new equilibrium atmospheric pCO2 (which can be seen from the atmospheric development Figure, see last page of this document). Here one can see that gas exchange is reduced as compared to the model control run for the Gas Slow experiment, and increased relative to the control for the Gas Fast experiment. Integrated globally, the net air-sea C flux is into the atmosphere during this transient phase for both experiments. Last, effects of slow gas exchange on marine δ13C will be visualised by providing Figure 3 for slow gas exchange in the SI.

**Author's changes in the manuscript in response to Comment 2)**

- Add a figure in the SI on the slow gas exchange experiment similar to the one in the main text for the fast gas exchange experiment (Figure 3):

[Figure]

- Improve text by explaining the cause for the pCO2 increase in the slow gas exchange experiment
- Provide plots of atmospheric development of δ13C and pCO2 during all experiments (see last page of this document)

- Add carbon flux figures at 100 years into the sensitivity experiments (Gas Fast and Gas Slow), in order to show the transient response that sets atmospheric pCO2:

[Figure]

Figure of carbon flux through the air-sea interface for the model control (upper left), and global Gas Fast (upper right) and Gas Slow (lower left) sensitivity experiments at 100 years. The corresponding globally integrated flux after 100 years is 0.037 Gt C/yr flux into the atmosphere for the Gas Fast experiment and 0.026 Gt C/yr flux into the atmosphere for Gas Slow.

**Comment 3)**

POC sinking rate: P7, L.20-21: As POC sinking rate increases, the decrease in air-sea gas exchange is most likely due to a reduced advection/mixing of carbon rich waters into the mixed layer. P7, L.28 it is stated that marine δ13C increases overall when POC sinking rates are high. Since δ13Catm increases under high POC sinking rates, it seems surprising that marine δ13C would also overall increase. . . In fact, the limited negative δ13C anomalies shown in Figure 5 are surprising. Is there a strong increase in organic carbon burial? Would it then make sense to show the transient changes? I am not sure about L. 33-34 p7: the difference in between the global change in POC and SO only change in POC export could only be due to difference in the area to which the forcing is applied, but might not be specific to SO. When applied globally, there is a significant impact on global export production as well as marine and atmospheric δ13C. The SO is a relatively small area of the ocean, so changes applied to that region only can be easily compensated. Results could be discussed with respect to previous experiments performed with the Bern3D and looking at the influence of the remineralization depth on atmospheric CO2 and δ13C (e.g. Roth et al., 2014 Earth system dynamics and Menviel et al., 2012, Quaternary Science Reviews).

**Author response to Comment 3)**

We thank Referee #1 for several detailed comments on our analysis of the POC sinking rate experiments. Here we try to respond to each point:

P7, L.20-21: The reduced air-sea gas exchange rate in response to high POC sinking rates is due to the almost complete export of surface ocean carbon to depth – thus not permitting escape to the atmosphere. Net upward advection/mixing of carbon and nutrients is thus reduced.

P7, L.28: Both marine (+0.15 permil) and atmospheric δ13C (+0.23 permil) increase because there is indeed a relatively higher loss of 12C than 13C (in POC) to the sediments in our experiment. The results presented are thus a transient response (as can be seen as well in the new SI Figure on atmospheric development during the experiments, see last page of this document). We have added an additional 10 000 years to the POC fast experiment to show the effects of a continued experiment, but argue not to go beyond that due a.o. extremely long equilibration times (as stated in Roth (2014), δ13C changes for over 200 000 years): See response to Referee #2, comment 2.2.3.

P7, L. 33-34: The relatively minor effect of the SO-only POC experiment is indeed compensated for outside of the SO, thank you for this improved explanation of our results. Discussion on context of previous studies: We agree with Referee #1 (and Referee #2) that a comparison to more previous studies are valuable for the reader and will improve the manuscript.

**Author's changes in the manuscript in response to Comment 3)**

-       Updating the discussed sentences on Page 7
-       Putting our results into the perspective of additional previous studies, as suggested by both Referee #1 and #2.

**Comment 4)**

Vmax: It is quite surprising that $\Delta\delta13Catm$ decreases when nutrient utilization increases. P8, L. 27: I doubt the correct reason for the surface negative $\delta13C$ anomaly is put forward. Maps of changes in export production and nutrients could be added to better understand the model response. If the nutrient advection to the surface of regions outside of SO is reduced, then so should be the advection of carbon rich - 13C depleted waters. This is also consistent with the significant atmospheric $CO_2$ reduction, but the $\delta13CO_2$ is more surprising. The change in nutrient utilization in the Southern Ocean should be given, as well as control and perturbed surface nutrient content.

**Author response to Comment 4)**

It is indeed surprising that $\delta13Catm$ decreases, and we see an explanation should be in place. We will provide the reader with a selection of maps of export production, nutrients, air-sea $pCO_2$ difference and carbon flux through the air-sea interface, such that the marine and atmospheric changes in response to the experiment are visualized. We also consider exploring alternative formulations of the experiment with a smaller perturbation of Vmax to check whether qualitatively the same changes occur then.

**Author's changes in the manuscript in response to Comment 4)**

-       Explain $\delta13Catm$ decrease, while providing the reader with more maps/information on the effects of the experiment on air-sea gas exchange, $pCO_2$ difference and nutrients.

**Comment 5)**

Sea-ice: Legend of Figure S4 needs additional information

**Author response to Comment 5)**

The caption of Figure S4 is indeed incomplete. We hope adding units and additional text on how the figure should be understood should provide the reader with enough information to interpret Figure S4.

**Author's changes in the manuscript in response to Comment 5)**

- Caption of Figure S4 adjusted to 'The pCO2 difference [ppm] between the surface ocean and the atmosphere for the model control run, based on an atmospheric value of 279 ppm. Negative values indicate a potential carbon flux into the ocean.'

**Comment 6)**

Hasted conclusions: The vertical gradient of δ13C is stated to vary by no more than 0.5 permil. But it should be noted that this includes the full range of anomalies obtained: from much lower to much higher than the control state. For example, the maximum changes in vertical δ13C gradient are obtained for Vmax (~+0.2 permil) and a fast gas exchange (~-0.25 permil), thus leading to ~0.5 permil change. It would be more appropriate to say that the maximum variation of each parameter leads to a ~0.25 permil change in vertical δ13C gradient, as the pre-industrial control state is an interglacial state.

Section 3.4., p10: very broad statements are made with respect to the impact of changes in ocean circulation on δ13C L. 17-18 and L. 20-27. These statements do not rely on any quantitative work on the impact of changes in ocean circulation on oceanic δ13C. The authors could for example consider looking at Menviel et al., 2015 (Global Biogeochemical Cycles) to have a better estimate of the impact of ocean circulation changes on δ13C. L. 21 to 23 are particularly unjustified because the rate of change of δ13C resulting from both biogeochemical changes and oceanic circulation are not studied here.

L. 14-15, p 12: I don't think that the results shown here indicate that the changes in pCO2 and δ13Catm are dependent on the location of the sea-ice edge, nor that sea-ice has a strong impact on atmospheric or oceanic δ13C.

**Author response to Comment 6)**

The generalization of 0.5 permil is something we want to make more specific. We will do this in two ways: 1) make the discussion and conclusion region-specific (revising Figure 4 with basin vertical gradients of δ13C) and 2) follow the reviewers advice to describe the effect of the individual sensitivity studies, instead of generalizing to a global average and a total effect.

Section 3.4: Regarding our statements on the effects of ocean circulation, we will extend our literature study in order to make more specific statements on the potential effects of ocean circulation on δ13C.

L. 14-15, p 12: We would like to rephrase this sentence to 'The effect of sea ice cover on pCO2atm and δ13Catm as well as marine δ13C depends on whether the sea ice covers a source or sink region for carbon.' We will also update P9, L20 accordingly, as it describes the same idea.

**Author's changes in the manuscript in response to Comment 6)**

-	Replace the general statement on 0.5 permil throughout the manuscript with a basin region-specific description of the results and describing the isolated effect of the sensitivity experiments
-	Update P12 L14-15 and P9, L20 to describe that the effect of sea ice cover depends on whether the ice covers a source or sink region for carbon.
-	Extend our literature study on the effects of ocean circulation on δ13C to make more specific statements on this topic.

**Comment minor points and typos)**

Throughout the text, please write "biogeochemical" without parentheses. P2, L. 3: "Air-sea" P6- L.2, please rephrase P6, L. 29: Please remove "In the ocean," Figure 8: y axis of second plot should read "pCO2 (ppm)"

**Author response to minor points and typos)**

Our apologies for these mistakes, and thank you for pointing them out.

**Author's changes in the manuscript in response to minor points and typos)**

- We will replace (bio)geochemical with biogeochemical
- We will replace air-se with air-sea on P2, L. 3
- Removing P6, L. 29 'In the ocean'
- We will replace the Figure 8 y-axis units with [ppm]

[Figure]

Draft version of New SI Figure showing atmospheric development of pCO2 and δ13C during the sensitivity experiments (global experiments for POC sinking and gas exchange)

---

## Author Comment (AC2) · 1 May 2018

**Author responses to Referee #2**

**Comment 1 General comments**

Anne L. Morée and co-authors use the HAMburg Ocean Carbon Cycle Model in its configuration for long-term simulations, HAMOCC2s (Heinze and Maier-Reimer, 1999). The authors report the results of four sensitivity experiments (actually four plus two, as two out of the four are run in duplicate, once for the global ocean and once for the Southern Ocean) to analyse (1) the effect of variations of the air-sea exchange parameters, (2) the sequestration efficiency of the organic pump via changed particulate organic carbon (POC) sinking rates, (3) the sequestration efficiency of the organic pump via increased nutrient utilisation efficiency, (4) changing sea-ice cover on atmospheric pCO2, $\delta^{13}C$ of atmospheric CO2 and δ13C in the ocean, and more specifically on the global mean vertical gradient of δ13C in dissolved inorganic carbon (DIC), quantified as the difference between DIC $\delta^{13}C$ in the surface and the deep ocean, denoted $\Delta$δ13C. Upon reading the abstract of this paper I got really excited. Carbon isotopes are a particularly useful tool for studying carbon cycling between the different spheres that make it up. They have been used for a long time for this purpose, but over the past years a wealth of new data have been published and more and more comprehensive global data compilations have become available. The time is thus ready to re-assess the different mechanisms with a model that offers an excellent balance between the comprehensiveness of the processes taken into account and their complexity and execution time, so that meaningful simulation experiments can be carried out for time scales of tens to hundreds of millennia.

The manuscript itself, however, did unfortunately not meet my expectations, far from. The language used, albeit generally fluent, seriously lacks precision and is rather colloquial. As an example, we repeatedly read that δ13C is depleted or enriched. It is of course DIC that is depleted or enriched in 13C. A relative deviation — such as δ13C — cannot be enriched or depleted; it can be high(er), greater or low(er), decreased or increased. The literature review is very poor; the same holds for comparison of the results obtained here to those of previous studies. Many important previous studies that called upon carbon isotopes for the study of glacial-interglacial carbon cycle changes are not cited (see below for details). $\Delta$δ13C, the proxy that is central to the paper really ought to be introduced with a more solid background. It was probably first used by Broecker (1982), at the very beginning of the "gold rush" time of the glacial-interglacial atmospheric CO2 problem studies (1980s). It was then used as a proxy for glacial-interglacial pCO2 variations, later fell out of favour, but has resurfaced over the past few years. One thing that would be important to emphasize here is, that it evolved in time: during those early stages, $\Delta$δ13C stood for the difference between δ13C in the deep and the surface ocean DIC. In the recent studies (e.g., Ziegler et al. (2013)), it now most often stands for the difference between δ13C of DIC in the deep sea and intermediate-depth (typically 400 m). The model description is incomplete. The processes that are relevant for the study are not described at all, only a reference to a previous paper is given. The experimental design leaves quite a number of questions open: the duration of the simulation experiments is only 2000 years. The separation between surface and deep ocean waters is questionable and as it obviously has an important influence on the results, the side-effects of this choice should have been discussed. Not all of the figures are reader-friendly: on Figure

5, e.g., readers are expected to visually extract $\Delta\delta13C$ from latitude-depth transects of $\delta13C$ by first averaging the topmost 250 m, then the depths below and to subtract both averages from each other. As a consequence, I cannot recommend this manuscript for publication in Biogeosciences at this stage. It should nevertheless be possible to reconsider it after a major revision and I strongly encourage the authors to prepare a version that remedies to all the shortcomings mentioned here. Please provide us with a better description of what is done, how it is done and why it is done that way. The study deals with an interesting and timely subject. The biogeochemical model at hand perfectly fits the needs. Please take full advantage of the possibilities it offers!

**Author's response to Comment 1**

Thank you for your detailed and thorough review of our manuscript. We appreciate the effort you have made to improve it: See below for a detailed reply to your comments. Regarding the points you only make in Comment 1, we will improve the precision of the language in general, and when discussing $\delta13C$ depletion/enrichment/increasing/decreasing. We will also extend the introduction to include a  paragraph on the development of $\Delta\delta13C$ research.

**Author's changes in the manuscript in response to Comment 1**

- Adjust the mention of 'four sensitivity experiments' to 'a set of sensitivity experiments'
- Replacing enriched/depleted when referring to $\delta13C$ with increased/decreased or higher/lower throughout the manuscript. Check language in the manuscript and improve where not precise.
- Add information on the development of $\Delta\delta13C$ research in the introduction, based on a selection of the papers mentioned in Comment 2.1.

**Comment 2.1 Literature**

Since the $\Delta\delta13C$ proxy has been in usage for more than 35 years, there is a wealth of studies that are available. They range from data-oriented studies to model-based studies, covering very similar approaches as done here. Only very few of them are cited in the manuscript and it is not entirely clear for what reasons they are included and others are excluded. There are more than 20 papers that come to my mind right away in this framework and that have not been considered in the literature review and the discussion of this paper [... literature list provided by Referee #2 ...]
Please do not get me wrong: I do not expect all of these papers to be cited. However, even this "out-of-the-mind" list is simply so long (and still far from exhaustive) that it is incomprehensible that none of these studies has been cited or taken into account for the purpose of discussing the results.

**Author's response to Comment 2.1**

Thank you for providing us with this literature list. We understand your wish for a stronger literature review and discussion, and made a selection of the literature you provided for our discussion/introduction.

**Author's changes in the manuscript in response to Comment 2.1**

- Where relevant, we will refer to and discuss additional previous studies. Specifically, we will to focus on
  Broecker (1982)
  Boyle (1988)
  Shackleton et al. (1983)
  Oppo et al (1990)
  Keir (1991)
  Mulitza et al. (1998)
  Toggweiler (1999)
  Murnane and Sarmiento (2000)
  Köhler et al. (2010)
  Lisiecki (2010)
  Lourantou et al (2010)
  Tschumi et al (2011)
  Menviel et al (2015)
  Eggleston et al (2016)
  Lear et al. (2016)
  Menviel et al. (2017)
  Duplessy et al. (1988)

**Comment 2.2.1 Model description is insufficient**

The model description given in the paper neither allows to reproduce the results reported here without a lot of guesswork, nor does it allow to fully understand the results. The provided description is in some instances even confusing: on page 3 (lines 13–14), we read that "POC is carried as a tracer as well as transported downwards according to an exponential penetration depth and constant settling velocity, [. . . ]". The exponential penetration profile and the constant settling velocity are of course not independent of each other. In the original technical reference to HAMOCC2s (Heinze and MaierReimer, 1999) – not cited in the manuscript – we read that "The vertical flux of biogenic particulate matter is parametrised through exponential redistribution profiles which implicitly include both sinking velocity and re-dissolution rate." This is not the same! It is quite easy to establish that the characteristic length scale in the exponential profile is equal to $\omega/k$, if the (constant) settling velocity is denoted $\omega$ and POC respiration is assumed to follow first order kinetics with a rate constant k. Since one of the experiments involves changes of the settling velocity, the adopted parametrisations must be correctly described.

**Author's response to Comment 2.2.1**

We will correct the description of the POC sinking (Page 3, lines 13-14) to agree with the technical report (Heinze and MaierReimer, 1999). Furthermore, we will provide additional information on how sea ice, nutrient uptake and air-sea gas exchange are parameterised in the model, so that the reader can better understand the changes we made for the sensitivity experiments.

**Author's changes in the manuscript in response to Comment 2.2.1**

- Add reference to the technical report on HAMOCC2 (Heinze and Maier-Reimer, 1999) in addition to the currently used reference to Heinze et al. (2016) - which contains the model version closest to the current one
- Provide information on parameterisation POC remineralisation/sinking
- Extend the paragraph on Page 4, lines 4-13 to complement Table 1 in how the experiments were set up. Thereby explain how changing Vmax relates to the Michaelis Menten kinetics and how the air-sea gas exchange experiments change the gas transfer coefficient. We will also add that the model is set up to make sea ice limit gas exchange and light penetration based on ice fraction.

**Comment 2.2.2 Model spin-up procedure**

The description of the model spin-up procedure lacks important details. We only read that "[. . . ] a fixed weathering input is used to tune the ocean inventories to values comparable to the observations." (page 3, line 24). On the basis of what quantitative constraints is this weathering flux determined? Are there separate fluxes
• for nutrients (phosphate)? – which would be necessary if organic matter is buried in the sediment together with the nutrients they lock up
• for DIC and alkalinity? – which would have to be separated if organic matter and carbonate are buried in the sediment
• for dissolved silica? – opal is also included in the model
• for 13C? – what is the δ13C signature of the DIC flux?
A decent model description would have answered half of the questions already. . . To what extent are the mismatches in the deep-ocean δ13C and PO4 concentration resulting from this spin-up? I would expect that they go together with global 13C and PO4 inventory mismatches as well, which, according to the description given here, are constraints.

**Author's response to Comment 2.2.2**

The weather flux is determined as described on P.3 lines 24-26: 'The 'best-fit' weathering value was found by running the model with a restored (to a value of -6.5 ‰) atmospheric δ13C until the prognostic burial rate reached equilibrium with weathering (after ~110000 model years).' So initially, the weathering flux was set to equal the prognostic sediment burial, and atmospheric carbon was restored to -6.5 ‰ (δ13C) and 278 ppm (pCO2atm). After ~110000 model years the burial (and thus weathering) flux equilibrated to a constant value. This also led to an ocean δ13C distribution that was closer to observed values. In consecutive model runs, we removed the restoring of the atmosphere and fixed the weathering rate to the value we obtained at the end of the 110000 year run, while keeping the burial rate prognostic. In this way, the ocean inventories remained close to observed, while permitting free atmospheric change.
Weathering fluxes are added homogeneously over the first ocean layer as dissolved matter. They are added in a fixed stoichiometric ratio for 12C, 13C, 14C, Alk, PO4 and Si. The 13C/12C ratio in the weathering flux would be equivalent to a δ13C of DIC of 14 ‰.

Author's changes in the manuscript in response to Comment 2.2.2

- Extend the explanation of the spinup procedure regarding burial/weathering to include tracers, stoichiometry, more details about the procedure and quantitative measures used
- The model description will be improved in general in the manuscript, see comment 2.2.1

**Comment 2.2.3 Sensitivity experiment duration**

The quality of the spin-up experiment is well quantified (residual drifts etc.). Unfortunately, nothing similar is reported for the sensitivity experiments. Readers are only told that these have been run for 2000 yr with the steady-state control run as initial condition. The strength of the model design for allowing long-term simulation experiments is initially emphasized (page 3, lines 10–11), a 110 000 yr spin-up run is carried out, and then the core experiments for the paper are run over a comparatively short duration of 2000 years only. For some of the perturbations (e.g., POC penetration depth changes. . . ), the model carbon cycle is still in the transient phase 2000 years after the onset of the perturbation. The choice of such short simulation experiments is thus rather incomprehensible.

**Author's response to Comment 2.2.3**

The length of the sensitivity experiments is chosen to be 2000 years, as we observed the air-sea gas exchange rate and sea ice cover experiments to show very little change in atmospheric carbon signature after this time (new SI Figure on atmospheric development, see last page of response to Referee #1). We agree that the effects of changing the biological pump (i.e., the POC and Vmax experiments) are still ongoing after 2000 years. To reach full isotopic equilibrium in the ocean however, over 200 000 model years of runtime could be needed (Roth (2014), see adjustments made to Page 7, lines 16-17). In an open system, the sediment loss of nutrients and carbon over time will empty the whole ocean of nutrients, which would not give very meaningful results. Besides that, over 200 000 years other feedback processes would happen as well in reality. To show the continued effect of a change in the biological pump efficiency, we will provide the reader with atmospheric development results of an extra 10 000 years for the fast POC sinking rate experiment in the SI.

**Author's changes in the manuscript in response to Comment 2.2.3**

- Provide the reader with the atmospheric development per sensitivity experiment of pCO2 and δ13C in the SI, see last page of response to Referee #1 document
- Clarify in sections 3.3.2 and 3.3.3 that there are still ongoing changes in the model
- Add δ13C and pCO2 development figure in SI for the fast POC sinking rate experiment for an additional 10 000 years

**Comment 2.3.1 Up- and downward fluxes, equilibrium δ13C**

Up- and downward fluxes, equilibrium δ13C Analysis of the results involves up- and downward fluxes Fup and Fdown: how are these obtained? To my best knowledge, it is only the net exchange flux Fnet which is proportional to the pCO2 difference between the surface water and the overlying atmosphere that can be calculated. The equilibrium δ13C (δ 13Ceq, first mentioned on page 7 at line 1) is not defined and an explanation how this is calculated is missing as well.

**Author's response to Comment 2.3.1**

In the model, separate fluxes Fup and Fdown are calculated by splitting the gas transfer formulation into to parts.
The gas transfer formulation as for example described in for example Orr et al. (2017):
'FA = kw ([A]sat − [A]), where for gas A, kw is its gas transfer velocity, [A] is its simulated surface-ocean dissolved concentration, and [A]sat is its corresponding saturation concentration in equilibrium with the water-vapor-saturated atmosphere at a total atmospheric pressure Pa.'
In the model, Fup=kw*[A]sat and Fdown=kw*[A] and Fnet=Fup-Fdown. This splitting up is useful for the calculation of air-sea fractionation of the carbon isotopes, as the fractionation factor only needs to be multiplied with Fup when calculating the effects of the air-sea gas exchange. It also proved to be useful for our discussion, because the total amount of exchange Fu+d influences δ13C.
The paragraph above will be added to the SI in order to explain our calculation method.

For clarification of the use of δ13Cdiseq and δ13Ceq, we add a definition of both to Page 6. L15: '[...] depending on the prior disequilibrium δ13Cdiseq (δ13Cdiseq = δ13C - δ13Ceq, where δ13Ceq represents the δ13C signature a water parcel would have had if it would have fully equilibrated with the atmosphere).'
δ13Ceq is not calculated, but the Gas Fast experiment provides insight into where the surface ocean is over or undersaturated with respect to δ13C, as the air-sea gas exchange rate is increased. Galbraith et al. 2015 provide more information on δ13Ceq and δ13Cdiseq and the study by Schmittner et al. 2013 also explored an infinite gas exchange rate.

**Author's changes in the manuscript in response to Comment 2.3.1**

- Adding a section in the SI on the calculation of Fup and Fdown
- Add to Page 6. L15: '[...] depending on the prior disequilibrium δ13Cdiseq (δ13Cdiseq = δ13C - δ13Ceq, where δ13Ceq represents the δ13C signature a water parcel would have had if it would have fully equilibrated with the atmosphere).'

**Comment 2.3.2 Separation between surface and deep realms**

In this study, the ocean is simply partitioned into a surface part, which encompasses the water masses above a 250 m depth horizon, and a deep part for the rest. No justification or explanation regarding this choice are given. First of all, it is a choice that leads to complications. Information gathered from previous publications based up HAMOCC2s

(Heinze et al., 1999, 2016) indicates that the eleven-layer configuration has no layer interface at 250 m depth, but a layer centred on 250 m depth. A more natural separation would be located at layer boundaries. Secondly, this choice is critical as it controls the results of the study to a large extent extent. At 250 m depth, the depth profile of DIC δ13C is generally rapidly decreasing (see e.g., Kroopnick (1985), but this should also be visible from the model results). Accordingly, the average surface ocean δ13C will be strongly biased towards lower values and the deep ocean slightly towards higher values. As a consequence, the amplitude of the vertical gradient, |Δδ13C|, is thus systematically underestimated. I think that surface ocean δ13C would more conveniently be taken from the surface layer down to 50 or 112.5 m depth (these are layer boundary depths in the 11-layer HAMOCC2s configuration, or even regionally variable in case information on the local mixed-layer depth would be available), and the deep ocean from the 1500 or the 2500 m depth horizons down to the sea floor. In any case care must be taken in the model-data comparison to make sure that surface-to-deep model gradients are compared to surface-to-deep data gradients and not to intermediate-to-deep data gradients.

**Author's response to Comment 2.3.2**

We see the potential problem with choosing the surface ocean as above 250 m depth and the deep ocean as below 250 m depth. We will adjust the definitions to better fit the model design and to prevent the strong surface ocean δ13C gradient to influence the averaging too much. In order to do so, we define the model photic layer (top 50 meter) as the 'surface ocean', because this is where biological production and fractionation during air-sea gas will mostly increase δ13C. We define 'the deep ocean' as the lowest model layer above the sea floor (if this over 3 km depth), as this is were benthic foraminifera will dwell and this ocean volume will be least influenced by the strong gradient in the vertical δ13C profiles (which could influence Δδ13C). We realise that due to the different definitions used for Δδ13C over the past decades/in different studies, no definition chosen by us will make direct comparison with a previous study possible. We feel however that by providing basin-averaged vertical gradients of δ13C, the reader could deduce their gradient of interest, or directly use the Δδ13C we will report.

**Author's changes in the manuscript in response to Comment 2.3.2**
- Adjust Δδ13C definition on Page 4, lines 1-3 to include the surface ocean as the ocean above 50 m depth/the photic zone and the deep ocean to be the lowermost wet layer in the ocean, if above 3 km depth.
- All reported Δδ13C values changed to fit the new definition of Δδ13C

**Comment 2.3.3 Regionalization**

Δδ13C results are only shown in the global mean. The three-dimensional HAMOCC2s should allow for a finer analysis than that. In the text, regional Δδ13C outcomes are sometimes mentioned, but it would be useful to have these results reported graphically as well, at least for basins or sub-basins (e.g., North Atlantic, South Atlantic, North Pacific,

South Pacific, Southern Ocean). Figure 4 could be easily adapted to show such more regionalized values in a useful and expressive way.

**Author's response to Comment 2.3.3**

We see the need for a less generalized and more basin/specific discussion of the results. This would also support a better comparison with sediment core studies/observational data.

**Author's changes in the manuscript in response to Comment 2.3.3**

- We will revise Figure 4 to show basin-mean δ13C profiles per global sensitivity experiment for the North Atlantic, South Atlantic, North Pacific, South Pacific, Southern Ocean and Indian Ocean, with the value for Δδ13C stated besides the profile. See also comment 1 to Referee #1. Part of this plot might be put in the SI, in order to not overwhelm the reader. The Δδ13C value is based on the new Δδ13C definition.
- Adjust the results, discussion and conclusion sections to use and describe the adjusted Figure 4 (i.e. results on basin scale)

**Comment 2.4 Discussion shortcomings**

Parts of the discussion are rather confusing. Section 3.4 is one of them. On one hand, we read that "The idealised and large perturbations [. . . ] show that mean Δδ13C varies no more than 0.5‰" on the other hand that "[the] reconstructed intra-millennial variability in Δδ13C could be driven more by changes in the biogeochemical state than by changes in ocean circulation because (bio)geochemical changes might occur more rapidly than whole-ocean circulation changes." Are large and whole-ocean changes in the biogeochemical state of the ocean really that more realistic on the time scales of a few millennia than circulation changes? At the latest from page 10, lines 29–30 on it is not clear any more which conclusions to draw from this study. Readers that have come this far will have seen the discussion revolve around SO Δδ13C in several instances, to learn now that, except for the North Atlantic, "data are too sparse to get a coherent picture of Δδ13C variations". Previously we have been shown that in the North Atlantic the deep-sea δ13C is mainly controlled by the air-sea exchange δ13C.

**Author's response to Comment 2.4**

We will clarify the discussion by putting it in a broader context (extended literature study, see comment 2.1) and by discussing results on a basin scale.

**Author's changes in the manuscript in response to Comment 2.4**

- Incorporation in the discussion of additional literature (see comment 2.1)
- Discussing the results on a basin scale
- Restructure section 3.4, remove Page 10 lines 21-23

**Comment 3 Technical comments**

Page 1, lines 3 and 27–28: "The standardised 13C isotope, δ 13C, [. . . ]": I have never seen this denomination in the peer-reviewed literature before. δ13C expresses the molar 13C/12C ratio of a sample in terms of its relative deviation from the ratio in a standard (initially PDB, now VPDB), generally expressed in permille. The references provided for this "standardised 13C isotope" are incomprehensible: Stenström et al. (2011) is a non peer-reviewed internal university report, Stuiver and Polach (1977) deals with radiocarbon. It should be straightforward to find an appropriate textbook reference for δ 13C.

Page 2, line 3: "air-se" should read "air-sea" Page 2, line 6: "10s" should be written out as "tens"

Page 2, line 24: the study by Shackleton and Pisias (1985) absolutely needs to be cited here alongside Charles et al. (2010) and Oliver et al. (2010).

Page 3, line 17: should "HAMOCC2" not read "HAMOCC2s"?

Page 3, line 23: should "HAMOCC2" not read "HAMOCC2s" again?

Page 4, line 23: "Eide (2017)": 2017a or 2017b?

Page 4, line 23: on the basis of the provided mean values, the intercept of the δ 13C:PO4 relationship is 3.27733, which would normally be rounded to 3.3, not to 3.4. Please check the numbers.

Page 5, line 12: "The modelled global POC production is [. . . ]": I guess this is the new or the export production – please clarify!

Page 5, lines 26–27: "[. . . ] with the exception of the Arctic Ocean where no POC production is modelled due to the sea ice cover [. . . ]": elsewhere in the paper we read that the sea-ice cover also isolates the surface ocean with respect to air-sea exchange. Does the partitioning into δ13C perc bio and δ13Cpercatm make sense in ice-covered regions?

Page 6, line 4: "change more than" should read "change by more than"

Page 6, line 9: "[. . . ] due to the fact that 12C needs to speciate [. . . ]": this does not make sense. 12C can only equilibrate at the same time as 13C – there are only the two of them. 12C should probably be corrected to DIC or CO2 (aq).

Page 6, line 19: "[. . . ] 22% of the global ocean area [. . . ]": does this include the ice-covered parts of the SO? – please specify

Page 6, line 21: "Fu" should read "Fup"

Page 6, line 22: "Fd" should read "Fdown"

Page 6, line 30: "[. . . ] lowers the surface ocean δ13C −0.2 to −0.9 ‰ in the lower latitudes [. . . ]" should read "[. . . ] lowers the surface ocean δ13C by −0.2 to −0.9 ‰ at the lower latitudes [. . . ]"

Page 6, line 31: "in high latitudes" should read "at high latitudes"

Page 6, line 31–32: "These results indicate the sign of the thermodynamic δ13C disequilibrium between surface ocean and atmosphere." – this sentence does not make sense, please reformulate.

Page 7, line 7: please add the ‰ sign to the 0.65 and the 1.00

Page 7, lines 16–17: "A more efficient biological pump [. . . ] leads to a loss of carbon to the sediments, which dominates the effects on pCOatm 2 and δ13C atm.": after 2000 years of simulation these effects have certainly not yet developed to their full strength.

Page 7, lines 24–25: "remineralisation horizon": a horizon depicts, in my understanding, a surface or a narrow zone, such as the calcite saturation horizon. I am not aware of the existence of a POC remineralisation horizon (and not even a carbonate remineralization horizon). Please rewrite.

Page 7, lines 29–30: "When reducing the biological pump efficiency both remineralisation and POC production are confined to the surface ocean.": as far as I know HAMOCC2s, the POC production is always confined to the surface and the remineralisation is taking place in subsurface intermediate and greater depths. Would "With a lower POC sinking rate, the remineralisation is confined to shallower depths." not be more correct?

Page 8, lines 5–9: Figure 5 which is referred to here, depicts δ13C and DIC anomalies with respect to the control run. Having readers derive information about Δδ13C from that figure is really asking too much. Why not provide the latitudinal evolution of the Δδ13C alongside? This would be a straightforward line plot.

Page 9, section 3.3.4: I would expect that such large ice-cover changes would also lead to circulation changes. A comment on this would be of order, wouldn't it?

Page 9, line 25: δ13Ceq: see above

Page 10, lines 23–24: "Analysis of SO Δδ13C reconstructions from sediment cores at 42◦S and 46◦S (Charles et al., 2010) shows that there is a strong correlation between these cores and Northern Hemisphere Δδ13C variations." This is not correct. Charles et al. (2010) show that there is a tight correlation between SO Δδ13C and "Northern Hemisphere climate fluctuations"; their paper does not even mention any Δδ13C record outside the SO.

Figures: if Δδ13C informations are to be read from a figure, this latter should then also show Δδ 13C.

Page 24, Figure 8b: units for pCO2 on the vertical axis should be ppm or µatm on the vertical axis, not ‰.

**Author response to Comment 3**

We apologise for the mistakes/lacking information at the points you have listed. We will clarify and correct the manuscript accordingly. See below for details.

**Author's changes in the manuscript in response to Comment 3**

Page 1, lines 3 and 27–28: Replaced
'The vertical marine δ13C gradient is the surface-to-deep difference in δ13C, the standardised 13C isotope (Stenström et al., 2011; Stuiver and Pollack, 1977). 13C is slightly heavier than the 12C isotope, which causes a fractionation effect during air-sea gas exchange and biogenic carbon uptake during photosynthesis (Laws et al., 1997; Mackenzie and Lerman, 2006; Zhang et al., 1995).'
By
'The vertical marine δ13C gradient is the surface-to-deep difference in δ13C, the standardised 13C/12C ratio expressed in permil (Eq. 1) (Zeebe, 2001). 13C is slightly heavier than the 12C isotope, which causes a fractionation effect during air-sea gas exchange and biogenic carbon uptake during photosynthesis thereby changing the 13C/12C ratio (Laws et al., 1997; Mackenzie and Lerman, 2006; Zhang et al., 1995).'

$$\delta^{13}\text{C} = \left( \frac{\left(\frac{^{13}\text{C}}{^{12}\text{C}}\right)_{\text{sample}}}{\left(\frac{^{13}\text{C}}{^{12}\text{C}}\right)_{\text{standard}}} - 1 \right) * 1000\ ^o/_{oo} \qquad \text{, (Eq. 1)}$$

where we used the PDB (13C/12C)standard (0.0112372).

Zeebe, R., & Wolf-Gladrow, D. (2001). CO2 in Seawater: Equilibrium, Kinetics,
Isotopes (Vol. 65). Amsterdam, The Netherlands: Elsevier Science B.V.

Page 2, line 3: Corrected as suggested

Page 2, line 6: Corrected as suggested

Page 2, line 24: Added reference to Shackleton and Pisias (1985)

Page 3, line 17: Corrected to HAMOCC2s

Page 3, line 23: Corrected to HAMOCC2s

Page 4, line 23: This should be 2017b, corrected accordingly

Page 4, line 23: Lines 18-19 are meant here, this should indeed be 3.3 - corrected

Page 5, line 12: Corrected to "The modelled global export POC production is [. . . ]"

Page 5, lines 26–27: Partitioning in air-sea gas exchange and biological components does mean something in ice-covered regions, as the upstream signal will be visible in such regions, and if the water mass transports POC, the biological-remineralization signal can increase with water mass age under the ice as well.

Page 6, line 4: Corrected as suggested

Page 6, line 9: Corrected as suggested

'This difference in equilibration time is due to the fact that 12C needs to speciate into all marine carbon species to reach equilibrium (~20x slower than O2), after which 13 10 C needs to go through full isotopic exchange between all carbon species to reach equilibrium (~10x slower than 12C) (Jones et al., 2014; Galbraith et al., 2015).'
to
'This difference in equilibration time is due to the fact that DIC needs to speciate into all marine carbon species to reach equilibrium (~20x slower than O2), while 13C needs to go through full isotopic exchange between all carbon species to reach equilibrium (~10x slower than DIC) (Jones et al., 2014; Galbraith et al., 2015).'

Page 6, line 19: "[. . . ] 22% of the global ocean area [. . . ]" corrected to "[. . . ] 22% of the global ice-free ocean area [. . . ]"

Page 6, line 21: Corrected as suggested

Page 6, line 22: Corrected as suggested

Page 6, line 30: Corrected as suggested

Page 6, line 31: Corrected as suggested

Page 6, line 31–32: "These results indicate the sign of the thermodynamic $\delta13C$ disequilibrium between surface ocean and atmosphere." adjusted to "These results show whether the thermodynamic $\delta13C$ disequilibrium $\delta13C_{diseq}$ is positive or negative."

For clarification of the use of $\delta13C_{diseq}$ and $\delta13C_{eq}$, we add to Page 6. L15: '[...] depending on the prior disequilibrium $\delta13C_{diseq}$ ($\delta13C_{diseq} = \delta13C - \delta13C_{eq}$, where $\delta13C_{eq}$ represents the $\delta13C$ signature a water parcel would have had if it would have fully equilibrated with the atmosphere).

Page 7, line 7: Corrected as suggested

Page 7, lines 16–17: The authors agree that these effects have not yet developed to their full strength, and will adjust the sentence to 'A more efficient biological pump (here, a higher POC sinking rate) leads to a loss of carbon to the sediments, which affects $pCO2_{atm}$ and $\delta13C_{atm}$ long-term (reference to new SI Figure on atmospheric development during the experiments, see last page of response to Referee #1 document), as found in a model study by Roth (2014).'

Page 7, lines 24–25: "remineralisation horizon" replaced by 'POC remineralisation'

Page 7, lines 29–30: We will rephrase this sentence to "With a lower POC sinking rate, the remineralisation is confined to the surface ocean."

Page 8, lines 5–9: We will address the issue with the visualisation of $\Delta\delta13C$ by presenting basin-specific δ13C profiles in an adjusted Figure 4 (see also comment 1 to Referee #1), with a basin-average $\Delta\delta13C$ noted next to each profile. Referral to that new figure instead of Figure 5 should provide the reader with enough information to understand the effects of the sensitivity experiment on $\Delta\delta13C$.

Page 9, section 3.3.4: We will add a sentence on Page 9, line 12 to state 'Ocean circulation changes that could result from a changed sea ice cover are not taken into account, as we want to study the potential isolated effect of sea ice on biological production and air-sea gas exchange.'

Page 9, line 25: See response to Page 6, line 31–32

Page 10, lines 23–24: In discussing our results in view of more literature, we will also correct the comparison and discussion with Charles et al. (2010) their results.

Figures: When referring to a figure when discussing or presenting $\Delta\delta13C$, we will in the adjusted manuscript refer to the basin-specific δ13C profiles that is presented in an adjusted Figure 4, and includes a value for $\Delta\delta13C$ for each basin. The $\Delta\delta13C$ value is based on the new $\Delta\delta13C$ definition.

Page 24, Figure 8b: Corrected to ppm

**References**

Duplessy, J. C., Shackleton, N. J., Fairbanks, R. G., Labeyrie, L., Oppo, D., and Kallel, N.: Deepwater source variations during the last climatic cycle and their impact on the global deepwater circulation, Paleoceanography, 3, 343-360, 10.1029/PA003i003p00343, 1988.

Heinze, C., and Maier-Reimer, E.: The Hamburg Oceanic Carbon Cycle Circulation Model Version "HAMOCC2s" for long time integrations, Max-Planck-Institut für Meteorologie, Hamburg REPORT 20, 1999.

Orr, J. C., Najjar, R. G., Aumont, O., Bopp, L., Bullister, J. L., Danabasoglu, G., Doney, S. C., Dunne, J. P., Dutay, J.-C., Graven, H., Griffies, S. M., John, J. G., Joos, F., Levin, I., Lindsay, K., Matear, R. J., McKinley, G. A., Mouchet, A., Oschlies, A., Romanou, A., Schlitzer, R., Tagliabue, A., Tanhua, T., and Yool, A.: Biogeochemical protocols and diagnostics for the CMIP6 Ocean Model Intercomparison Project (OMIP), Geosci. Model Dev., 10, 2169-2199, https://doi.org/10.5194/gmd-10-2169-2017, 2017.

Roth, R., Ritz, S. P., and Joos, F.: Burial-nutrient feedbacks amplify the sensitivity of atmospheric carbon dioxide to changes in organic matter remineralisation, Earth Syst. Dynam., 5, 321-343, https://doi.org/10.5194/esd-5-321-2014, 2014.

Shackleton, N. J., Hall, M. A., Line, J., & Shuxi, C. (1983). Carbon isotope data in core V19-30 confirm reduced carbon dioxide concentration in the ice age atmosphere. Nature, 306, 319. doi:10.1038/306319a0

---

## Author Response (AR1)

Dear Professor Joos, dear referees,

Many thanks for the comments and constructive suggestions to improve this manuscript. We have tried to incorporate them into our revised version. Please, find details about the changes made in the responses below and the marked-up manuscript.

The author's response is presented as follows:
(1) **Referee #1**: comments from referee #1 with author's response to each comment and summary of author's changes in the manuscript,
(2) **Referee #2**: comments from referee #2 with author's response to each comment and summary of author's changes in the manuscript,
(3) **Marked-up manuscript**: Author's changes tracked in the new manuscript

As our manuscript went through major revisions, we could often not list the specific changes in the manuscript in our response. We therefore refer to the new version of the manuscript to read the combined result of all changes we have made.

Sincerely, Anne Morée and co-authors

**Referee #1**

**Comment 1)**

Modelling study in the context of paleoproxy data: The motivation behind the study is to better understand variations in oceanic $\delta^{13}C$ as measured in foraminiferas. This is discussed in the context of the two site-specific studies: Charles et al., (2010) and Ziegler et al., (2013), comparing mid-depth (400m and 1500m) to deep $\delta^{13}C$ in the Southern Ocean as well as the global study of Oliver et al., (2010). But all the discussion stays very vague and qualitative with "increased/decreased" vertical gradients over "glacial/interglacial" timescales and mostly "globally averaged" for the numerical experiments. This induces some relatively vague conclusions such as in the abstract L. 17-18, or p12 L. 20-25. This is also true in section 3.4. In addition, in that section results of Charles et al. (2010) and Ziegler et al., (2013) are discussed in a bit more detailed but they are compared to the simulated mean vertical $\delta^{13}C$ gradient, which is defined as $\delta^{13}C$ surface $\delta^{13}C$ deep, where $\delta^{13}C$ surface and $\delta^{13}C$ deep respectively represent mean $\delta^{13}C$ for depths above and below 250m (please note that the "deep" ocean cannot be defined as the area below 250m depth). This is however different to Ziegler et al., who compare ~400m depth to the deep ocean (~3000m), and Charles et al., (2010) who compare cores at ~1200m and ~4600m. In general, wouldn't it make sense to show vertical profiles of globally average or basin average $\Delta\delta^{13}C$ ($\delta^{13}C$ at depth compared to $\delta^{13}C$ averaged over the first 250m)? Such a figure could replace Figure 4 and add a bit more information about the processes at play.

**Author response to Comment 1)**

Referee #1 kindly made us aware of the too generalized and qualitative discussion throughout the manuscript. We addressed this issue by 1) providing and discussing basin-averaged $\delta^{13}C$ profiles (new Figure 4 and S7), 2) redefining $\Delta\delta^{13}C$ (Section 2) and 3) extending our literature study. These adjustments especially changed section 3.3, 3.4 and 4 – which are now presented on both a global and a basin scale.

**Author's changes in the manuscript in response to Comment 1)**

   - Addition of a new Figure 4 and S7 by basin-mean vertical profiles of $\delta^{13}C$ for the Southern Ocean, North Pacific, South Pacific, North Atlantic, South Atlantic and Indian ocean.
   - Redefining $\Delta\delta^{13}C$ and updating the results accordingly (Section 2, and throughout manuscript)
   - Updating discussion/abstract/conclusion to be more quantitative and specific and using the revised Figure 4 and S7 to discuss the sensitivity of $\Delta\delta^{13}C$ on a basin scale (throughout manuscript)

**Comment 2)**

Air sea gas exchange experiments: I find the results quite surprising. A $pCO_2$ increase and $\delta^{13}CO2$ decrease for fast gas exchange make sense, but a $pCO_2$ increase for a slow gas exchange is surprising. There are no graphs shown for the slow gas exchange case, so it is hard to judge

**Author response to Comment 2)**

The authors agree with Referee #1 that the discussion of $pCO_2$ sensitivity to slow gas exchange rates is not explained enough in the current manuscript. As stated in the manuscript, '$pCO_2^{atm}$ is governed by the transient change in the net air-sea gas exchange flux Fnet, which occurs until a new equilibrium is established'. In order to explain the slow gas exchange experiment, we added carbon flux figures at 100 years (new Fig. S4), when the transient response determines the new equilibrium atmospheric $pCO_2$ (new Fig. S5). Here one can see that gas exchange is reduced as compared to the model control run for the 'Gas slow' experiment, and increased relative to the control for the 'Gas fast' experiment. Integrated globally, the net air-sea C flux is into the atmosphere during this transient phase for both experiments. In addition to presenting these new SI figures, we revised section 3.3.1 to better explain our results. Last, effects of slow gas exchange on marine $\delta^{13}C$ is now presented in new Fig. S6.

**Author's changes in the manuscript in response to Comment 2)**

   - New figures on carbon fluxes during the transient phase (Fig. S4), atmospheric development during the experiments (Fig. S5) and effects of slow gas exchange on marine $\delta^{13}C$ (Fig. S6).
   - Revised section 3.3.1

**Comment 3)**

POC sinking rate: P7, L.20-21: As POC sinking rate increases, the decrease in air-sea gas exchange is most likely due to a reduced advection/mixing of carbon rich waters into the mixed layer. P7, L.28 it is stated that marine $\delta^{13}C$ increases overall when POC sinking rates are high. Since $\delta^{13}C^{atm}$ increases under high POC sinking rates, it seems surprising that marine $\delta^{13}C$ would also overall increase. . . In fact, the limited negative $\delta^{13}C$ anomalies shown in Figure 5 are surprising. Is there a strong increase in organic carbon burial? Would it then make sense to show the transient changes? I am not sure about L. 33-34 p7: the difference in between the global change in POC and SO only change in POC export could only be due to difference in the area to which the forcing is applied, but might not be specific to SO. When applied globally, there is a significant impact on global export production as well as marine and atmospheric $\delta^{13}C$ . The SO is a relatively small area of the ocean, so changes applied to that region only can be easily compensated. Results could be discussed with respect to previous experiments performed with the Bern3D and looking at the influence of the remineralization depth on atmospheric $CO_2$ and $\delta^{13}C$ (e.g. Roth et al., 2014 Earth system dynamics and Menviel et al., 2012, Quaternary Science Reviews).

**Author response to Comment 3)**

We thank Referee #1 for several detailed comments on our analysis of the POC sinking rate experiments. We revised section 3.3.2 to include a better literature review and improved explanation of the original P7, L.20-21, L.28 and L. 33-34 as requested. We also extended the model description in section 2, so that the POC sinking experiment is better described. In specific, for the original P7, L.20-21: The reduced air-sea gas exchange rate in response to high POC sinking rates is due to the almost complete export of surface ocean carbon to depth – thus not permitting escape to the atmosphere. Net upward advection/mixing of carbon and nutrients is thus reduced. For original P7, L.28: Both marine (+0.15 permil) and atmospheric $\delta^{13}C$ (+0.23 permil) increase because there is indeed a relatively higher loss of $^{12}C$ than $^{13}C$ (in POC) to the sediments in our experiment. The results presented are thus a transient response (new Fig. S5). We have added an additional 10 000 years to the POC fast experiment to show the effects of a continued experiment (new Fig. S5), but argue not to go beyond that due a.o. extremely long equilibration times (as stated in Roth (2014), $\delta^{13}C$ changes for over 200 000 years). The transient character of the experiment is now more clearly described in section 3.3.2. For original P7, L. 33-34: The relatively minor effect of the SO-only POC experiment is indeed compensated for outside of the SO, thank you for this improved explanation of our results.

**Author's changes in the manuscript in response to Comment 3)**

- Thorough revision of section 3.3.2 to include more literature and better explain the results
- Clarification of transient character of the POC sinking experiment due to sediment burial
- Extended model description in section 2

**Comment 4)**

$V_{max}$: It is quite surprising that $\delta^{13}C^{atm}$ decreases when nutrient utilization increases.
P8, L. 27: I doubt the correct reason for the surface negative $\delta^{13}C$ anomaly is put forward. Maps of changes in export production and nutrients could be added to better understand the model response. If the nutrient advection to the surface of regions outside of SO is reduced, then so should be the advection of carbon rich - $^{13}C$ depleted waters. This is also consistent with the significant atmospheric $CO_2$ reduction, but the $\delta^{13}CO2$ is more surprising. The change in nutrient utilization in the Southern Ocean should be given, as well as control and perturbed surface nutrient content.

**Author response to Comment 4)**

While re-analysing the $V_{max}$ experiment, we discovered a programming mistake in the setup of the experiment, which caused the non-SO maximum nutrient uptake rate to change as well. We therefore repeated the experiment, which changed the results. We therefore rewrote section 3.3.3 and adjusted the presented results of the $V_{max}$ experiment throughout the manuscript. The effects on phosphate and oxygen concentrations are included in the text, atmospheric fluxes are presented in a new Figure S7 and the effects on POC export production as compared to the control are presented in new Figure S8.

**Author's changes in the manuscript in response to Comment 4)**

- New SI Figure S7 (equilibrium air-sea fluxes) and S8 (POC export production)
- Revised the whole of section 3.3.3 after the discovery of a mistake in the experiment setup

**Comment 5)**

Sea-ice: Legend of Figure S4 needs additional information

**Author response to Comment 5)**

The caption of the old Figure S4 was indeed incomplete. We added units and additional text on how the figure should be understood: the new Figure is S3.

**Author's changes in the manuscript in response to Comment 5)**

- Caption of Figure S4 (now Fig. S3) adjusted to 'The $pCO_2$ difference [ppm] between the surface ocean and the atmosphere for the model control run, based on an atmospheric value of 279 ppm. Negative values indicate a potential carbon flux into the ocean.'

**Comment 6)**

Hasted conclusions: The vertical gradient of $\delta^{13}C$ is stated to vary by no more than 0.5 permil. But it should be noted that this includes the full range of anomalies obtained: from much lower to much higher than the control state. For example, the maximum changes in vertical $\delta^{13}C$ gradient are obtained for Vmax ($\sim$+0.2 permil) and a fast gas exchange ($\sim$-0.25 permil), thus leading to $\sim$0.5 permil change. It would be more appropriate to say that the maximum variation of each parameter leads to a $\sim$0.25 permil change in vertical $\delta^{13}C$ gradient, as the pre-industrial control state is an interglacial state.

Section 3.4., p10: very broad statements are made with respect to the impact of changes in ocean circulation on $\delta^{13}C$ L. 17-18 and L. 20-27. These statements do not rely on any quantitative work on the impact of changes in ocean circulation on oceanic $\delta^{13}C$ . The authors could for example consider looking at Menviel et al., 2015 (Global Biogeochemical Cycles) to have a better estimate of the impact of ocean circulation changes on $\delta^{13}C$ . L. 21 to 23 are particularly unjustified because the rate of change of $\delta^{13}C$ resulting from both biogeochemical changes and oceanic circulation are not studied here.

L. 14-15, p 12: I don't think that the results shown here indicate that the changes in $pCO_2$ and $\delta^{13}C^{atm}$ are dependent on the location of the sea-ice edge, nor that sea-ice has a strong impact on atmospheric or oceanic $\delta^{13}C$ .

**Author response to Comment 6)**

We addressed the issues raised in comment 6 by making the discussion (section 3.3 and 3.4) and conclusions (section 3.4 and 4) region-specific (new Figure 4 and S7) and by doing a more elaborate literature study throughout the manuscript. Note that the new definition of $\Delta\delta^{13}C$ causes all gradients to be stronger than in the previous version of the manuscript.

**Author's changes in the manuscript in response to Comment 6)**

- Effects of the sensitivity experiments on $\Delta\delta^{13}C$ are now described both globally and on a basin scale (sections 3.3, 3.4 and 4)
- Our literature study is extended to include more specific statements on the effects of ocean circulation on $\delta^{13}C$ and compare the basin-specific results to previous studies.

**Comment minor points and typos)**

Throughout the text, please write "biogeochemical" without parentheses. P2, L. 3: "Air-sea" P6- L.2, please rephrase P6, L. 29: Please remove "In the ocean," Figure 8: y axis of second plot should read "$p$CO$_2$ (ppm)"

**Author response to minor points and typos)**

Our apologies for these mistakes, and thank you for pointing them out.

**Author's changes in the manuscript in response to minor points and typos)**

  - We replaced (bio)geochemical with biogeochemical throughout the manuscript
  - We replaced air-se with air-sea on original P2, L. 3
  - We removed P6, L. 29 'In the ocean'
  - We replaced the Figure 8 y-axis units with [ppm]

**Referee #2**

**Comment 1 General comments**

Anne L. Morée and co-authors use the HAMburg Ocean Carbon Cycle Model in its configuration for long-term simulations, HAMOCC2s (Heinze and Maier-Reimer, 1999). The authors report the results of four sensitivity experiments (actually four plus two, as two out of the four are run in duplicate, once for the global ocean and once for the Southern Ocean) to analyse (1) the effect of variations of the air-sea exchange parameters, (2) the sequestration efficiency of the organic pump via changed particulate organic carbon (POC) sinking rates, (3) the sequestration efficiency of the organic pump via increased nutrient utilisation efficiency, (4) changing sea-ice cover on atmospheric $p$CO$_2$, $\delta^{13}$C of atmospheric CO$_2$ and $\delta^{13}$C in the ocean, and more specifically on the global mean vertical gradient of $\delta^{13}$C in dissolved inorganic carbon (DIC), quantified as the difference between DIC $\delta^{13}$C in the surface and the deep ocean, denoted $\Delta\delta^{13}$C . Upon reading the abstract of this paper I got really excited. Carbon isotopes are a particularly useful tool for studying carbon cycling between the different spheres that make it up. They have been used for a long time for this purpose, but over the past years a wealth of new data have been published and more and more comprehensive global data compilations have become available. The time is thus ready to re-assess the different mechanisms with a model that offers an excellent balance between the comprehensiveness of the processes taken into account and their complexity and execution time, so that meaningful simulation experiments can be carried out for time scales of tens to hundreds of millennia.

The manuscript itself, however, did unfortunately not meet my expectations, far from. The language used, albeit generally fluent, seriously lacks precision and is rather colloquial. As an example, we repeatedly read that $\delta^{13}$C is depleted or enriched. It is of course DIC that is depleted or enriched in $^{13}$C. A relative deviation — such as $\delta^{13}$C — cannot be enriched or depleted; it can be high(er), greater or low(er), decreased or increased. The literature review is very poor; the same holds for comparison of the results obtained here to those of previous studies. Many important previous studies that called upon carbon isotopes for the study of glacial-interglacial carbon cycle changes are not cited (see below for details). $\Delta\delta^{13}$C , the proxy that is central to the paper really ought to be introduced with a more solid background. It was probably first used by Broecker (1982), at the very beginning of the "gold rush"

time of the glacial-interglacial atmospheric $CO_2$ problem studies (1980s). It was then used as a proxy for glacial-interglacial $pCO_2$ variations, later fell out of favour, but has resurfaced over the past few years. One thing that would be important to emphasize here is, that it evolved in time: during those early stages, $\Delta\delta^{13}C$ stood for the difference between $\delta^{13}C$ in the deep and the surface ocean DIC. In the recent studies (e.g., Ziegler et al. (2013)), it now most often stands for the difference between $\delta^{13}C$ of DIC in the deep sea and intermediate-depth (typically 400 m). The model description is incomplete. The processes that are relevant for the study are not described at all, only a reference to a previous paper is given. The experimental design leaves quite a number of questions open: the duration of the simulation experiments is only 2000 years. The separation between surface and deep ocean waters is questionable and as it obviously has an important influence on the results, the side-effects of this choice should have been discussed. Not all of the figures are reader-friendly: on Figure 5, e.g., readers are expected to visually extract $\Delta\delta^{13}C$ from latitude-depth transects of $\delta^{13}C$ by first averaging the topmost 250 m, then the depths below and to subtract both averages from each other. As a consequence, I cannot recommend this manuscript for publication in Biogeosciences at this stage. It should nevertheless be possible to reconsider it after a major revision and I strongly encourage the authors to prepare a version that remedies to all the shortcomings mentioned here. Please provide us with a better description of what is done, how it is done and why it is done that way. The study deals with an interesting and timely subject. The biogeochemical model at hand perfectly fits the needs. Please take full advantage of the possibilities it offers!

**Author's response to Comment 1**

Thank you for your detailed and thorough review of our manuscript. We appreciate the effort you have made to improve it: See below for a detailed reply to your comments. Regarding the points you only make in Comment 1, we hope you will see that we improved the precision of the language in general. We also replaced the use of enriched/depleted when referring to $\delta^{13}C$ with increased/decreased or higher/lower throughout the manuscript. We also extended the introduction to include a paragraph on the development of $\Delta\delta^{13}C$ research.

**Author's changes in the manuscript in response to Comment 1**

- Adjusted the mention of 'four sensitivity experiments' to 'a set of sensitivity experiments'
- Replaced enriched/depleted when referring to $\delta^{13}C$ with increased/decreased or higher/lower throughout the manuscript.
- Language is improved throughout the manuscript to be more specific and quantitative.
- Add information on the development of $\Delta\delta^{13}C$ research in the introduction, based on a selection of the papers mentioned in Comment 2.1.

**Comment 2.1 Literature**

Since the $\Delta\delta^{13}C$ proxy has been in usage for more than 35 years, there is a wealth of studies that are available. They range from data-oriented studies to model-based studies, covering very similar approaches as done here. Only very few of them are cited in the manuscript and it is not entirely clear for what reasons they are included and others are excluded. There are more than 20 papers that come to my mind right away in this framework and that have not been considered in the literature review and the discussion of this paper
[... literature list provided by Referee #2 ...]

Please do not get me wrong: I do not expect all of these papers to be cited. However, even this "out-of-the-mind" list is simply so long (and still far from exhaustive) that it is incomprehensible that none of these studies has been cited or taken into account for the purpose of discussing the results.

**Author's response to Comment 2.1**

We have included most of the references listed by Referee #2, as well as some others, in our revised manuscript (mainly affection Sections 1, 3.3, 3.4, 3.5 and 4). The reference list is updated accordingly at the end of the revised manuscript. We feel that this greatly improved our manuscript as we could now provide the reader with both a better comparison to previous studies and with an improved explanation of our results. We thus thank Referee #2 for pointing us to these studies.

**Author's changes in the manuscript in response to Comment 2.1**

- Extension of the literature review by about 20 previous studies on this or related topics. These references are mainly included in the discussion of our results (Sections 3.3-3.5).

**Comment 2.2.1 Model description is insufficient**

The model description given in the paper neither allows to reproduce the results reported here without a lot of guesswork, nor does it allow to fully understand the results. The provided description is in some instances even confusing: on page 3 (lines 13–14), we read that "POC is carried as a tracer as well as transported downwards according to an exponential penetration depth and constant settling velocity, [. . . ]". The exponential penetration profile and the constant settling velocity are of course not independent of each other. In the original technical reference to HAMOCC2s (Heinze and MaierReimer, 1999) – not cited in the manuscript – we read that "The vertical flux of biogenic particulate matter is parametrised through exponential redistribution profiles which implicitly include both sinking velocity and re-dissolution rate." This is not the same! It is quite easy to establish that the characteristic length scale in the exponential profile is equal to $\omega/k$, if the (constant) settling velocity is denoted $\omega$ and POC respiration is assumed to follow first order kinetics with a rate constant k. Since one of the experiments involves changes of the settling velocity, the adopted parametrisations must be correctly described.

**Author's response to Comment 2.2.1**

We corrected the model description in Section 2 to include a better explanation of the POC sinking parameterisation, as well as the setup of all sensitivity experiments. Next to the description in the main text, we provide the reader with the model details on POC production and sinking in SI 1A.

**Author's changes in the manuscript in response to Comment 2.2.1**

- Model description improved in Section 2 as well as details added in SI 1A
- Sensitivity experiment description extended in Section 2

**Comment 2.2.2 Model spin-up procedure**

The description of the model spin-up procedure lacks important details. We only read that "[. . . ] a fixed weathering input is used to tune the ocean inventories to values comparable to the observations." (page 3, line 24). On the basis of what quantitative constraints is this weathering flux determined? Are there separate fluxes

• for nutrients (phosphate)? – which would be necessary if organic matter is buried in the sediment together with the nutrients they lock up

• for DIC and alkalinity? – which would have to be separated if organic matter and carbonate are buried in the sediment

• for dissolved silica? – opal is also included in the model

• for $^{13}$C? – what is the $\delta^{13}$C signature of the DIC flux?

A decent model description would have answered half of the questions already. . . To what extent are the mismatches in the deep-ocean $\delta^{13}$C and PO4 concentration resulting from this spin-up? I would expect that they go together with global $^{13}$C and PO4 inventory mismatches as well, which, according to the description given here, are constraints.

**Author's response to Comment 2.2.2**

We improved the model description to include more detail, including the equilibration of the model. As now described in the revised Section 2, the 'best-fit' weathering value for $^{13}$C was found by running the model with a restored (to a value of -6.5 ‰) atmospheric $\delta^{13}$C until the burial rate reached equilibrium with weathering (after ~110000 model years). After that, we permitted free development of atmospheric $\delta^{13}$C . In this way, the ocean inventories remained close to observed, while permitting free atmospheric change of $\delta^{13}$C and $p$CO$_2$$^{atm}$. Weathering fluxes are added homogeneously over the first ocean layer as dissolved matter in a fixed stoichiometric ratio for C, O$_2$, Alkalinity, PO$^4$$_{3-}$ and Si. The $^{13}$C/$^{12}$C ratio in the weathering flux would be equivalent to a $\delta^{13}$C of DIC of 14 ‰.

**Author's changes in the manuscript in response to Comment 2.2.2**

-   We extended the explanation of the spinup procedure in Section 2 regarding burial/weathering to include tracers, stoichiometry, and more details.

**Comment 2.2.3 Sensitivity experiment duration**

The quality of the spin-up experiment is well quantified (residual drifts etc.). Unfortunately, nothing similar is reported for the sensitivity experiments. Readers are only told that these have been run for 2000 yr with the steady-state control run as initial condition. The strength of the model design for allowing long-term simulation experiments is initially emphasized (page 3, lines 10–11), a 110 000 yr spin-up run is carried out, and then the core experiments for the paper are run over a comparatively short duration of 2000 years only. For some of the perturbations (e.g., POC penetration depth changes. . . ), the model carbon cycle is still in the transient phase 2000 years after the onset of the perturbation. The choice of such short simulation experiments is thus rather incomprehensible.

**Author's response to Comment 2.2.3**

The length of the sensitivity experiments is chosen to be 2000 years, as we observe all experiments except for those on the POC sinking rate to show very little change in atmospheric carbon signature after this time (now also presented to the readers in new Fig. S5): Equilibrium is often already reached within the first ~800 years. We agree that the effects of changing the biological pump (i.e., the POC and Vmax experiments) are still ongoing after 2000 years. To reach full isotopic equilibrium in the ocean however, over 200 000 model years of runtime could be needed (Roth (2014), see adjustments made to Page 7, lines 16-17). In an open system, the sediment loss of nutrients and carbon over time will empty the whole ocean of nutrients, which would not give very meaningful results. Besides that, over 200 000 years other feedback processes would happen as well in reality. To show the continued

effect of a change in the biological pump efficiency, we provided the reader with atmospheric development results of an extra 10 000 years for the fast POC sinking rate experiment in new Figure S5. From this, we observe atmospheric development of $\delta^{13}C$ indeed going beyond 12000 years, after which we stopped the experiment (Fig. S5). We summarize this result by stressing in the manuscript that the results of the POC sinking rate experiments are transient results (Sections 2 and 3.3.2).

**Author's changes in the manuscript in response to Comment 2.2.3**

- Provide the reader with the atmospheric development per sensitivity experiment of $pCO_2$ and $\delta^{13}C$ in new Fig. S5, with an additional 10000 years for the 'POC fast' experiment
- Clarify in sections 2 and 3.3.2 that there are still ongoing changes in the model

**Comment 2.3.1 Up- and downward fluxes, equilibrium $\delta^{13}C$**

Up- and downward fluxes, equilibrium $\delta^{13}C$ Analysis of the results involves up- and downward fluxes Fup and Fdown: how are these obtained? To my best knowledge, it is only the net exchange flux Fnet which is proportional to the $pCO_2$ difference between the surface water and the overlying atmosphere that can be calculated. The equilibrium $\delta^{13}C$ ($\delta$ 13Ceq, first mentioned on page 7 at line 1) is not defined and an explanation how this is calculated is missing as well.

**Author's response to Comment 2.3.1**

In the model, separate fluxes Fup and Fdown are calculated by splitting the gas transfer formulation into two parts. A paragraph on this is added to the SI (Section SI 1B) in order to explain this in detail. For clarification of the use of $\delta^{13}C$ diseq and $\delta^{13}C$ eq, we add definitions in Section 3.3.1. $\delta^{13}C$ diseq = $\delta^{13}C$ - $\delta^{13}C$ eq, where $\delta^{13}C$ eq represents the $\delta^{13}C$ signature a water parcel would have had if it would have fully equilibrated with the atmosphere. Not that $\delta^{13}C$ eq is not calculated, but the Gas Fast experiment provides insight into where the surface ocean is over or undersaturated with respect to $\delta^{13}C^{atm}$. We also included several new references to improve our explanation and reasoning.

**Author's changes in the manuscript in response to Comment 2.3.1**

- SI 1B describes Fup and Fdown
- Definition provided of $\delta^{13}C$ diseq and $\delta^{13}C$ eq in section 3.3.1

**Comment 2.3.2 Separation between surface and deep realms**

In this study, the ocean is simply partitioned into a surface part, which encompasses the water masses above a 250 m depth horizon, and a deep part for the rest. No justification or explanation regarding this choice are given. First of all, it is a choice that leads to complications. Information gathered from previous publications based up HAMOCC2s (Heinze et al., 1999, 2016) indicates that the eleven-layer configuration has no layer interface at 250 m depth, but a layer centred on 250 m depth. A more natural separation would be located at layer boundaries. Secondly, this choice is critical as it controls the results of the study to a large extent extent. At 250 m depth, the depth profile of DIC $\delta^{13}C$ is generally rapidly decreasing (see e.g., Kroopnick (1985), but this should also be visible from the model results). Accordingly, the average surface ocean $\delta^{13}C$ will be strongly biased towards lower values and the deep ocean slightly towards higher values. As a consequence, the amplitude of the vertical gradient, $|\Delta\delta^{13}C|$, is thus systematically underestimated. I think that surface ocean $\delta^{13}C$ would more conveniently be

taken from the surface layer down to 50 or 112.5 m depth (these are layer boundary depths in the 11-layer HAMOCC2s configuration, or even regionally variable in case information on the local mixed-layer depth would be available), and the deep ocean from the 1500 or the 2500 m depth horizons down to the sea floor. In any case care must be taken in the model-data comparison to make sure that surface-to-deep model gradients are compared to surface-to-deep data gradients and not to intermediate-to-deep data gradients.

**Author's response to Comment 2.3.2**

We see the potential problem with choosing the surface ocean as above 250 m depth and the deep ocean as below 250 m depth. We adjusted the definitions to better fit the model design and to prevent the strong surface ocean $\delta^{13}C$ gradient to influence the averaging too much. In order to do so, we define the model photic layer (top 50 meter) as the 'surface ocean', because this is where biological production and fractionation during air-sea gas will mostly increase $\delta^{13}C$ . We define 'the deep ocean' as the lowest model layer above the sea floor (if this over 3 km depth), as this is were benthic foraminifera will dwell and this ocean volume will be least influenced by the strong gradient in the vertical $\delta^{13}C$ profiles (which could influence $\Delta\delta^{13}C$ ). As Referee #2 mentions, this increases our estimate of $\Delta\delta^{13}C$ everywhere; however, it did not change our conclusions. We realise that due to the different definitions used for $\Delta\delta^{13}C$ over the past decades/in different studies, no definition chosen by us will make direct comparison with a previous study possible. We feel however that by providing basin-averaged vertical gradients of $\delta^{13}C$ , the reader could deduce their gradient of interest, or directly use the $\Delta\delta^{13}C$ we report.

**Author's changes in the manuscript in response to Comment 2.3.2**

- Adjust $\Delta\delta^{13}C$ definition to include the surface ocean as the ocean above 50 m depth/the photic zone and the deep ocean to be the lowermost wet layer in the ocean, if above 3 km depth.
- All reported $\Delta\delta^{13}C$ values changed to fit the new definition of $\Delta\delta^{13}C$

**Comment 2.3.3 Regionalization**

$\Delta\delta^{13}C$ results are only shown in the global mean. The three-dimensional HAMOCC2s should allow for a finer analysis than that. In the text, regional $\Delta\delta^{13}C$ outcomes are sometimes mentioned, but it would be useful to have these results reported graphically as well, at least for basins or sub-basins (e.g., North Atlantic, South Atlantic, North Pacific, South Pacific, Southern Ocean). Figure 4 could be easily adapted to show such more regionalized values in a useful and expressive way.

**Author's response to Comment 2.3.3**

A less generalized and more basin-specific discussion of the results is made throughout the manuscript in response to comment 2.3.3. The main improvement lies in the presentation and discussion of the new Figure 4 and S7, which presents basin-average $\delta^{13}C$ gradients in response to the sensitivity experiments. In combination with the extended literature review, this made the whole manuscript more region-specific as also requested in the general comment 1 of Referee #2.

**Author's changes in the manuscript in response to Comment 2.3.3**

- New Figure 4 and S7 show basin-mean $\delta^{13}C$ profiles per sensitivity experiment for the North Atlantic, South Atlantic, North Pacific, South Pacific, Southern Ocean and Indian Ocean, with the value for $\Delta\delta^{13}C$ stated besides the profile. Part of this plot is put in the SI (new Figure S9),

in order to not overwhelm the reader and focus on the most significant results. The $\Delta\delta^{13}C$ value reported here is based on the new $\Delta\delta^{13}C$ definition.

- Adjust the results, discussion and conclusion sections to use and describe the new Figure 4 and S7.

**Comment 2.4 Discussion shortcomings**

Parts of the discussion are rather confusing. Section 3.4 is one of them. On one hand, we read that "The idealised and large perturbations [. . . ] show that mean $\Delta\delta^{13}C$ varies no more than 0.5‰" on the other hand that "[the] reconstructed intra-millennial variability in $\Delta\delta^{13}C$ could be driven more by changes in the biogeochemical state than by changes in ocean circulation because (bio)geochemical changes might occur more rapidly than whole-ocean circulation changes." Are large and whole-ocean changes in the biogeochemical state of the ocean really that more realistic on the time scales of a few millennia than circulation changes? At the latest from page 10, lines 29–30 on it is not clear any more which conclusions to draw from this study. Readers that have come this far will have seen the discussion revolve around SO $\Delta\delta^{13}C$ in several instances, to learn now that, except for the North Atlantic, "data are too sparse to get a coherent picture of $\Delta\delta^{13}C$ variations". Previously we have been shown that in the North Atlantic the deep-sea $\delta^{13}C$ is mainly controlled by the air-sea exchange $\delta^{13}C$ .

**Author's response to Comment 2.4**

We clarified the discussion by putting it in a broader context (extended literature study, see comment 2.1) and by discussing results on a basin scale as described in our response to the comments above.

**Author's changes in the manuscript in response to Comment 2.4**

- Incorporation in the discussion of additional literature (see comment 2.1)
- Discussing the results on a basin scale (new Figure 4 and S7, and Sections 3.3-4)

**Comment 3 Technical comments**

Page 1, lines 3 and 27–28: "The standardised $^{13}C$ isotope, $\delta\ ^{13}C$, [. . . ]": I have never seen this denomination in the peer-reviewed literature before. $\delta^{13}C$ expresses the molar $^{13}C/^{12}C$ ratio of a sample in terms of its relative deviation from the ratio in a standard (initially PDB, now VPDB), generally expressed in permille. The references provided for this "standardised $^{13}C$ isotope" are incomprehensible: Stenström et al. (2011) is a non peer-reviewed internal university report, Stuiver and Polach (1977) deals with radiocarbon. It should be straightforward to find an appropriate textbook reference for $\delta\ ^{13}C$.

Page 2, line 3: "air-se" should read "air-sea" Page 2, line 6: "10s" should be written out as "tens"

Page 2, line 24: the study by Shackleton and Pisias (1985) absolutely needs to be cited here alongside Charles et al. (2010) and Oliver et al. (2010).

Page 3, line 17: should "HAMOCC2" not read "HAMOCC2s"?

Page 3, line 23: should "HAMOCC2" not read "HAMOCC2s" again?

Page 4, line 23: "Eide (2017)": 2017a or 2017b?

Page 4, line 23: on the basis of the provided mean values, the intercept of the $\delta\ ^{13}C$:PO4 relationship is 3.27733, which would normally be rounded to 3.3, not to 3.4. Please check the numbers.

Page 5, line 12: "The modelled global POC production is [. . . ]": I guess this is the new or the export production – please clarify!

Page 5, lines 26–27: "[. . . ] with the exception of the Arctic Ocean where no POC production is modelled due to the sea ice cover [. . . ]": elsewhere in the paper we read that the sea-ice cover also isolates the surface ocean with respect to air-sea exchange. Does the partitioning into $\delta^{13}C$ perc bio and $\delta^{13}C$ percatm make sense in ice-covered regions?

Page 6, line 4: "change more than" should read "change by more than"

Page 6, line 9: "[. . . ] due to the fact that $^{12}C$ needs to speciate [. . . ]": this does not make sense. $^{12}C$ can only equilibrate at the same time as $^{13}C$ – there are only the two of them. $^{12}C$ should probably be corrected to DIC or $CO_2$ (aq).

Page 6, line 19: "[. . . ] 22% of the global ocean area [. . . ]": does this include the ice-covered parts of the SO? – please specify

Page 6, line 21: "Fu" should read "Fup"

Page 6, line 22: "Fd" should read "Fdown"

Page 6, line 30: "[. . . ] lowers the surface ocean $\delta^{13}C$ −0.2 to −0.9 ‰ in the lower latitudes [. . . ]" should read "[. . . ] lowers the surface ocean $\delta^{13}C$ by −0.2 to −0.9 ‰ at the lower latitudes [. . . ]"

Page 6, line 31: "in high latitudes" should read "at high latitudes"

Page 6, line 31–32: "These results indicate the sign of the thermodynamic $\delta^{13}C$ disequilibrium between surface ocean and atmosphere." – this sentence does not make sense, please reformulate.

Page 7, line 7: please add the ‰ sign to the 0.65 and the 1.00

Page 7, lines 16–17: "A more efficient biological pump [. . . ] leads to a loss of carbon to the sediments, which dominates the effects on pCOatm 2 and $\delta^{13}C^{atm}$.": after 2000 years of simulation these effects have certainly not yet developed to their full strength.

Page 7, lines 24–25: "remineralisation horizon": a horizon depicts, in my understanding, a surface or a narrow zone, such as the calcite saturation horizon. I am not aware of the existence of a POC remineralisation horizon (and not even a carbonate remineralization horizon). Please rewrite.

Page 7, lines 29–30: "When reducing the biological pump efficiency both remineralisation and POC production are confined to the surface ocean.": as far as I know HAMOCC2s, the POC production is always confined to the surface and the remineralisation is taking place in subsurface intermediate and greater depths. Would "With a lower POC sinking rate, the remineralisation is confined to shallower depths." not be more correct?

Page 8, lines 5–9: Figure 5 which is referred to here, depicts $\delta^{13}C$ and DIC anomalies with respect to the control run. Having readers derive information about $\Delta\delta^{13}C$ from that figure is really asking too much. Why not provide the latitudinal evolution of the $\Delta\delta^{13}C$ alongside? This would be a straightforward line plot.

Page 9, section 3.3.4: I would expect that such large ice-cover changes would also lead to circulation changes. A comment on this would be of order, wouldn't it?

Page 9, line 25: $\delta^{13}C$ eq: see above

Page 10, lines 23–24: "Analysis of SO $\Delta\delta^{13}C$ reconstructions from sediment cores at 42◦S and 46◦S (Charles et al., 2010) shows that there is a strong correlation between these cores and Northern Hemisphere $\Delta\delta^{13}C$ variations." This is not correct. Charles et al. (2010) show that there is a tight correlation between SO $\Delta\delta^{13}C$ and "Northern Hemisphere climate fluctuations"; their paper does not even mention any $\Delta\delta^{13}C$ record outside the SO.

Figures: if $\Delta\delta^{13}C$ informations are to be read from a figure, this latter should then also show $\Delta\delta^{13}C$.

Page 24, Figure 8b: units for $p$CO$_2$ on the vertical axis should be ppm or µatm on the vertical axis, not ‰.

**Author response to Comment 3**

We apologise for the mistakes/lacking information at the points you have listed. We clarified and correct the manuscript accordingly. Because of the major revision of the paper, some sentences may have been totally rephrased or replaced. See below for details.

**Author's changes in the manuscript in response to Comment 3**

Page 1, lines 3 and 27–28: Replaced
'The vertical marine $\delta^{13}C$ gradient is the surface-to-deep difference in $\delta^{13}C$ , the standardised $^{13}C$ isotope (Stenström et al., 2011; Stuiver and Pollack, 1977). $^{13}C$ is slightly heavier than the $^{12}C$ isotope, which causes a fractionation effect during air-sea gas exchange and biogenic carbon uptake during photosynthesis (Laws et al., 1997; Mackenzie and Lerman, 2006; Zhang et al., 1995).'
By
'The vertical marine $\delta^{13}C$ gradient ($\Delta\delta^{13}C$ ) is the surface-to-deep difference in $\delta^{13}C$ , the standardised $^{13}C/^{12}C$ ratio expressed in permil (Eq. 1 and 2) (Zeebe and Wolf-Gladrow, 2001). $^{13}C$ is slightly heavier than the $^{12}C$ isotope which causes a fractionation effect during air-sea gas exchange and photosynthesis, thereby changing $\delta^{13}C$ and $\Delta\delta^{13}C$ (Laws et al., 1997; Mackenzie and Lerman, 2006; Zhang et al., 1995)'

$$\delta^{13}C = \left( \frac{\left(\frac{^{13}C}{^{12}C}\right)_{sample}}{\left(\frac{^{13}C}{^{12}C}\right)_{standard}} - 1 \right) * 1000\ ^o/_{oo}$$
, (Eq. 1)

where we used the PDB ($^{13}C/^{12}C$)standard (0.0112372).

Page 2, line 3: Corrected as suggested
Page 2, line 6: Corrected as suggested
Page 2, line 24: Added reference to Shackleton and Pisias (1985)
Page 3, line 17: Corrected to HAMOCC2s
Page 3, line 23: Corrected to HAMOCC2s
Page 4, line 23: This should be 2017b, corrected accordingly
Page 4, line 23: Original lines 18-19 are meant here, this should indeed be 3.3 - corrected
Page 5, line 12: Corrected to "The modelled global export POC production is [. . . ]"
Page 5, lines 26–27: Partitioning in air-sea gas exchange and biological components does mean something in ice-covered regions, as the upstream signal will be visible in such regions, and if the water mass transports POC, the biological-remineralisation signal can increase with water mass age under the ice as well.
Page 6, line 4: Corrected as suggested
Page 6, line 9: Corrected as suggested
'This difference in equilibration time is due to the fact that $^{12}C$ needs to speciate into all marine carbon species to reach equilibrium (~20x slower than $O_2$), after which 13 10 C needs to go through full isotopic exchange between all carbon species to reach equilibrium (~10x slower than $^{12}C$) (Jones et al., 2014; Galbraith et al., 2015).'
to
'This difference in equilibration time is due to the fact that DIC needs to speciate into all marine carbon species to reach equilibrium (~20x slower than $O_2$), while $^{13}C$ needs to go through full isotopic exchange between all carbon species to reach equilibrium (~10x slower than DIC) (Jones et al., 2014; Galbraith et al., 2015; Broecker and Peng, 1974).'
Page 6, line 19: "[. . . ] 22% of the global ocean area [. . . ]" corrected to "[. . . ] 22% of the global ice-free ocean area [. . . ]"
Page 6, line 21: Corrected as suggested

Page 6, line 22: Corrected as suggested

Page 6, line 30: Corrected as suggested

Page 6, line 31: Corrected as suggested

Page 6, line 31–32: "These results indicate the sign of the thermodynamic $\delta^{13}$C disequilibrium between surface ocean and atmosphere." adjusted to "These results show whether the thermodynamic $\delta^{13}$C disequilibrium $\delta^{13}$C diseq is positive or negative."

For clarification of the use of $\delta^{13}$C diseq and $\delta^{13}$C eq, we added definitions in section 3.3.1.

Page 7, line 7: Corrected as suggested

Page 7, lines 16–17: The authors agree that these effects have not yet developed to their full strength, and adjusted the sentence (first paragraph Section 3.3.2) to 'A more efficient biological pump (here, a higher POC sinking rate) leads to a loss of carbon to the sediments, which affects $p$CO$_2^{atm}$ and $\delta^{13}$C$^{atm}$ long-term. The results presented here are therefore 2000-year transient results because full equilibrium of marine $\delta^{13}$C could take over 200 000 years (Roth et al. 2014). We observe atmospheric development of $\delta^{13}$C beyond 12000 years, after which we stopped the experiment (Fig. S5). An even longer experiment duration would no longer be meaningful as the open system loses carbon and nutrients to the ocean sediments and in reality, other processes and feedbacks would occur on such timescales (Tschumi et al., 2011).'

Page 7, lines 24–25: "remineralisation horizon" replaced by 'POC remineralisation'

Page 7, lines 29–30: We rephrase this sentence to "With a lower POC sinking rate, the remineralisation is confined to the surface ocean."

Page 8, lines 5–9: We addressed the issue with the visualisation of $\Delta\delta^{13}$C by presenting basin-specific $\delta^{13}$C profiles in an new Figure 4 (see also comment 1 to Referee #1), with a basin-average $\Delta\delta^{13}$C noted next to each profile. Referral to that new figure instead of Figure 5 should provide the reader with enough information to understand the effects of the sensitivity experiment on $\Delta\delta^{13}$C .

Page 9, section 3.3.4: We added a sentence here to state that 'Ocean circulation changes that could result from a changed sea ice cover are not taken into account, as we want to study the potential isolated effect of sea ice on biological production and air-sea gas exchange.'

Page 9, line 25: See response to Page 6, line 31–32

Page 10, lines 23–24: In discussing our results in view of more literature, we also corrected the comparison and discussion with Charles et al. (2010) their results (major revision of Section 3.4).

Figures: When referring to a figure when discussing or presenting $\Delta\delta^{13}$C , we now refer to the basin-specific $\delta^{13}$C profiles that are presented in a new Figure 4, and include a value for $\Delta\delta^{13}$C for each basin. The $\Delta\delta^{13}$C value is based on the new $\Delta\delta^{13}$C definition.

Page 24, Figure 8b: Corrected to ppm

**Marked-up manuscript**

[revised manuscript text omitted]
 δ¹³C distribution, the Δδ¹³C as well as δ¹³C^atm and pCO₂^atm. pCO₂^atm is governed by thea transient change in the net air-sea gas exchange flux $F_{net}$, which occurs until a new equilibrium is established. We find an increase of pCO₂^atm by 9 ppm (fast gas exchange) and by 4 ppm (slow gas exchange), respectively. If gas exchange is only changed in the SO (i.e. for 22 % of the global ice-free ocean area), an effect of 5 ppm and 1 ppm increase is found (Table 2). The spatially variable prior pCO₂ disequilibrium in the SO (Fig. S3) plays an important role in the pCO₂^atmatmospheric pCO₂ sensitivity: The larger increase of the outgassing flux $F_{up}$ of the SO as compared to the carbon uptake flux $F_{down}$ leads to a reduced SO carbon sink and higher pCO₂^atm at increased gas exchange rates. The reduction in air-sea C flux for the slow gas exchange experiment causes $F_{net}$ to decrease during the transient phase (Fig. S4 and 5), leading to an increase in pCO₂^atm which develops during the first ~600 years. $F_{net}$ is reduced during the transient phase because the slow gas exchange rate decreases Southern Hemispheric net C uptake, while maintaining Northern Hemispheric net C outgassing, also for the global experiment. Interestingly, the δ¹³C^atm gets decoupled from the pCO₂^atm signal as δ¹³C^atm decreases (to -6.8 ‰) during fast gas exchange and increases (to -6.3 ‰) when the gas exchange rate is reduced. This is explained by the increase in the global amount of air-sea gas exchange $F_{u+d}$ in the fast gas exchange experiment. Such an increase leads to a smaller thermodynamic disequilibrium, which enriches increases the mean marine δ¹³C and depletes lowers δ¹³C^atm. The opposite occurs for the sSlow gas exchange reduces $F_{u+d}$, causing less fractionation to occurexperiment which explains the increase of δ¹³C^atm. Moreover, our SO-only experiments show that these effects on δ¹³C^atm the atmosphere are more pronounced if gas exchange only changes in the SO. This indicates that the remainder of the ocean offsets part of the atmospheric sensitivity to SO change.

In the ocean, δ¹³C shows a different response in high latitudes as compared to the lower latitudes in the surface ocean (Fig. 3a and S6): An increased air-sea gas exchange rate lowers the surface ocean δ¹³C of DIC by -0.2 to -0.9 ‰ in at the lower latitudes and increases surface ocean δ¹³C in at high latitudes by 0.2-0.5 ‰ (Fig. 3 and 4). These results indicate whether $\delta^{13}C_{diseq}$ is positive or negative, since δ¹³C is closer to equilibrium at high gas exchange rates. These results indicate the sign of the thermodynamic δ¹³C disequilibrium between surface ocean and atmosphere. In line with previous studies (Schmittner et al., 2013; Galbraith et al., 2015) the disequilibrium is negative ($\delta^{13}C < \delta^{13}C_{eq}$) at high latitudes and in low latitude upwelling regions, and positive elsewhere. This can be understood from the difference between the natural δ¹³C distribution (Fig. 1) and the potential ~2 ‰ $\delta^{13}C_{eq}$, which would require an increase in δ¹³C in cool high latitude surface waters and a decrease in warm low latitude surface waters (Murnane and Sarmiento, 2000). The net effect of a slower gas exchange rate on surface ocean δ¹³C is less pronounced than the effect of and reversed to the effects discussed for an increased gas exchange rate (not shownFig. S6, Fig. 3). The smaller effects seen for slow gas exchange indicate that the control model ocean is a 'slow ocean', i.e. closer to (very) slow gas exchange than to thermodynamic equilibrium (fast gas exchange).

The effect of the gas exchange rate on marine δ¹³C is mostly established in the top 250 to 1000 m of the water column (Fig. 3c, d, Fig. 4). Recording this air-sea gas exchange signal thus strongly depends on the reliability of planktic δ¹³C-based $\delta^{13}C_{surface}$ reconstructions and knowledge of the living depth represented by the planktic foraminifera. The signal penetrates deepest (to ~2000 m depth) into the North Atlantic (Fig. 4, Fig. S7), where δ¹³C is strongly influenced by air-sea gas exchange

(Fig. 2a). However, the interpretation of variations in North Atlantic benthic $\delta^{13}$C as coming partly from air-sea gas exchange (Lear et al., 2016) is not supported by our experiment. Due to the limited export of the $\delta^{13}$C signal to depth, the sensitivity of $\Delta\delta^{13}$C to the gas exchange rate  mainly  comes from surface ocean $\delta^{13}$C. Globally, the $\Delta\delta^{13}$C weakens to 0.84 ‰ when the thermodynamic disequilibrium is decreased (i.e. 'Gas fast', Fig. 5) and $\Delta\delta^{13}$C strengthens to 1.00 ‰ when the thermodynamic disequilibrium is increased ('Gas slow', Fig. 5). The extent to which thermodynamic equilibrium can develop is thus an efficient way to change the biologically-induced $\Delta\delta^{13}$C (Murnane and Sarmiento, 2000), however only in lower latitudes where $\delta^{13}$C$_{diseq}$ is positive.  Importantly, The SO  $\Delta\delta^{13}$C signal has an opposite sign of the global mean and low latitude regions: When the thermodynamic disequilibrium is decreased (increases), basin-mean $\Delta\delta^{13}$C in the SO increases (decreases) and thus intensifies the biologically-induced $\Delta\delta^{13}$C changes (Fig. 3a4).

**3.3.2 The biological pump: POC sinking rate**

The net effect of a regionally changed biological pump efficiency depends on the sequestration efficiency, which depends on the interplay between the biological pump and ocean circulation (DeVries et al., 2012). A more efficient biological pump (here, a higher POC sinking rate) leads to a loss of carbon to the sediments, which affects $p$CO$_2^{atm}$ and $\delta^{13}$C$^{atm}$ on millennial timescales. Here we present results from a 2000-year simulation (as for the other experiments), which is still in a transient phase. To reach a full equilibrium of the system could take as long as 200 000 years (Roth et al. 2014). On these long timescales other processes and feedbacks would occur (Tschumi et al. 2011), which complicates the attribution of changes to a primary trigger. A fast POC sinking rate leads to a $p$CO$_2^{atm}$ decrease of  28 ppm  and  higher (-6.2 ‰) atmospheric $\delta^{13}$C  after 2000 years (Table 2, Fig. S5) as well as a shift of the mean ocean $\delta^{13}$C by ~0.15 ‰, caused by the sediment burial of low-$\delta^{13}$C POC. The imbalance between weathering and burial fluxes can thus cause profound changes in both marine and atmospheric $\delta^{13}$C, and moreover provides an important feedback for the long-term marine carbon cycle (Roth et al., 2014; Tschumi et al., 2011). In our experiment, an efficient biological pump leads to a global ~10 % decrease in the amount of air-sea gas exchange $F_{u+d}$ because  of efficient export of carbon to depth, thereby lowering the net upward advection of carbon. A mirrored but weaker response is modelled for a decrease in biological pump efficiency. Halving the POC sinking rate leads to a 13 ppm increase in $p$CO$_2^{atm}$ (of which 28 % can be explained by the SO) and a more negative atmospheric $\delta^{13}$C (-6.7 ‰) and increased $F_{u+d}$ (Table 2, Fig. S5).

Surface ocean $\delta^{13}$C is mostly influenced by the changes in productivity and the vertical displacement of the POC remineralisation depth. The deepening of the remineralisation depth has been extensively discussed in the literature (Boyle, 1988; Keir, 1991; Mulitza et al., 1998; Roth et al., 2014), and likely explains lowered mid-depth glacial $\delta^{13}$C together with

changes in ocean circulation (for example, Toggweiler, 1999) horizon. POC sinking removes nutrients and preferentially light $^{12}$C carbon from the surface ocean, while exporting them to the deep ocean. If POC sinking rates are high, Tthis leaves theincreases the surface ocean more enriched in δ$^{13}$Cδ$^{13}$C (contributing to the δ$^{13}$C$^{atm}$ increase) and lowersthe deep ocean δ$^{13}$C 
[revised manuscript text omitted]
 Locally however, larger variations in $\Delta\delta^{13}C$ can occur. $\Delta\delta^{13}C$ reconstructions based on sediment core $\delta^{13}C$ data show $\Delta\delta^{13}C$ variations of ~1 ‰ over the past 350 000 years (Boyle, 1988; Shackleton et al., 1983; Shackleton and Pisias, 1985; Ziegler et al., 2013; Charles et al., 2010; Oliver et al., 2010). Ocean circulation changes explain at least part of these variations in $\Delta\delta^{13}C$ (Charles et al., 2010; Heinze et al., 1991; Jansen, 2017; Heinze and Hasselmann, 1993; Oppo et al., 1990; Toggweiler 1999). However, the changes in the biogeochemical state of the ocean imposed in our experiments show that reconstructed variations in $\Delta\delta^{13}C$ may could be strongly influenced by (SO) (bio)biogeochemistry as well.cal change. $\
[revised manuscript text omitted]
 $\underline{\text{global mean}}$ $\Delta\delta^{13}C$ $\underline{\text{(Fig. 9a) and a negative linear}}$ $\underline{\text{relationship between } p\text{CO}_2{}^{atm} \text{ and global mean } \Delta\delta^{13}C \text{ (Fig. 9b) }}$ hold over a wide range of $\underline{\text{bio}}$geochemical states$\underline{,}$ $\underline{\text{as}}$ simulated in the sensitivity experiments. $\underline{\text{This result supports previous studies that show both local correlation between } \Delta\delta^{13}C}$ $\underline{\text{and } p\text{CO}_2{}^{atm} \text{ (such as found by for example Dickson et al. (2008)) and correlation of modified } \Delta\delta^{13}C \text{ between ocean basins}}$ $\underline{\text{with } p\text{CO}_2{}^{atm} \text{ (Lisiecki, 2010). The effects of ocean circulation on glacial-interglacial } \delta^{13}C^{atm} \text{ changes, not studied here, are}}$

10 $\underline{\text{most pronounced in response to Antarctic Bottom Water formation rate variations and are of the order of 0-0.15 ‰ (Menviel}}$ $\underline{\text{et al., 2015). Our results show that } \delta^{13}C^{atm} \text{ varies up to } \sim\pm0.5 \text{ ‰ in response to biogeochemical changes (Table 2).}}$ Figure 9a shows that changes in the POC sinking rate lie approximately along a line in $\delta^{13}C^{atm}$:$\Delta\delta^{13}C$ space$\underline{\text{, suggesting that changes in}}$ $\underline{\text{the biological pump efficiency is important for the } \delta^{13}C^{atm}\text{:}\Delta\delta^{13}C \text{ relationship. Likewise, the relationship between } p\text{CO}_2{}^{atm} \text{ and}}$ $\underline{\Delta\delta^{13}C \text{ is mostly coming from the biological pump, as air-sea gas exchange affects } \Delta\delta^{13}C \text{ much more than } p\text{CO}_2{}^{atm} \text{ (Fig. 9b).}}$

15 $\underline{C}$hanges in air-sea gas exchange (as simulated in the gas exchange and sea ice cover experiments) affect $\delta^{13}C^{atm}$ more than $\Delta\delta^{13}C$. This confirms the idea that $\Delta\delta^{13}C$ is governed by biological processes and will also set $\delta^{13}C^{atm}$, unless air-sea gas exchange gets the chance to dominate $\delta^{13}C^{atm}$. $\underline{\text{The high potential of SO air-sea gas exchange}}$ $\underline{\text{to steer } \delta^{13}C^{atm}}$ $\underline{\text{(Table 2: Sea ice and gas exchange rate experiments) complements recent studies showing that increased SO}}$

20 $\underline{\text{ventilation of deep ocean carbon is a likely candidate for glacial-interglacial } \delta^{13}C^{atm} \text{ excursions – which are of the order of 0.5}}$ $\underline{\text{‰ (Bauska et al., 2016; Eggleston et al., 2016; Lourantou et al., 2010; Menviel et al., 2015).}}$

25  **4 Summary and conclusions**

This study addressed the  sensitivity of $\underline{\text{modelled marine and atmospheric}}$ $\delta^{13}C$ and $\Delta\delta^{13}C$ $\underline{\text{to changes}}$ $\underline{\text{in the biogeochemical parameters under constant ocean circulation}}$, focusing on the contribution of the SO $\underline{\text{(the ocean south of}}$ $\underline{45° \text{ S, 22 \% of the global ice-free ocean area). Variations of } \Delta\delta^{13}C \text{ recorded in sediment records are sensitive to ocean}}$

30 $\underline{\text{circulation changes as shown in previous studies, but here we show that the biogeochemical state of the (Southern) Ocean also}}$ $\underline{\text{can have large effects on } \Delta\delta^{13}C \text{ across all ocean basins.}}$ Using the ocean biogeochemistry general circulation model HAMOCC2s, a set of  sensitivity experiments $\underline{\text{was carried out, which focuses on the biogeochemical aspects known to be}}$

important for δ13C and Δδ13C. Specifically, the experiments explore changes in air-sea gas exchange rate, sea ice extent (influencing both biological production and the air-sea gas exchange of carbon) and the efficiency of the biological pump through the POC sinking rate and nutrient uptake rate. gave insight in the effects of (bio)geochemical change on δ13C and Δδ13C.

5   The results show the important role of the SO in determining global δ13C and the Δδ13C sensitivities, as well as the strong spatial differences in these sensitivity. A new equilibrium state developed mostly within the first 100-800 years of the sensitivity experiments, except for the POC sinking experiment where an imbalance between weathering and burial causes a long-term drift. The δ13C signature is governed by different processes depending on location:- Air-sea gas exchange sets surface ocean δ13C in all ocean basins, contributing 60-100 % to the δ13C signature. At depth and with increasing water mass age, the

10   influence of biology increases to 50 % in the oldest water masses (North Pacific) due to POC remineralisation. This spatial diversity of the processes pattern behind the δ13C signature of a water parcel results in a non-uniform sensitivity of δ13C to (bio)biogeochemical change. Global mean Δδ13C varies up to ~±0.4 ‰ due to biogeochemical state changes in our experiments (at a constant ocean circulation) (Fig. 5). This amplitude is almost half of the reconstructed variation in Δδ13C on glacial-interglacial timescales (1 ‰), and could thus contribute to variations in Δδ13C together with ocean circulation changes.

15   However, Δδ13C can have a different response on a basin scale: The ocean's oldest water (North Pacific) responds most to biological changes, the young deep water (North Atlantic) responds strongly to air-sea gas exchange changes, and the vertically well-mixed water (SO) has a low or even reversed Δδ13C sensitivity as compared to the other basins. The amplitude of the Δδ13C sensitivity can be higher at decreasing scale, as seen from a maximum sensitivity of ~-0.6 ‰ on ocean basin scale (Fig. 4). Interestingly, the direction of both glacial (intensification of Δδ13C) and interglacial (weakening of Δδ13C) Δδ13C change

20   matches changes in biogeochemical processes thought to be associated with these periods. This supports the idea that biogeochemistry explains part of the reconstructed variations in Δδ13C, in addition to changes in ocean circulation. The variations in Δδ13C are caused by both biology and gas exchange processes in the surface ocean but mostly by biological processes in the deep ocean.

An increased gas exchange rate has the potential to reduce the biologically-induced Δδ13C through the reduction of surface

25   ocean and atmospheric δ13C. Increased gas exchange however only reduces Δδ13C in the low latitudes: In high latitudes, increased gas exchange will increase Δδ13C (by increasing δ13C$_{surface}$) because of the negative disequilibrium δ13C$_{diseq}$ (i.e. δ13C < δ13C$_{
[revised manuscript text omitted]

---

## Referee Report (RR1)

Review of the first revised version of

**"Southern Ocean controls of the vertical marine $\delta^{13}$C gradient – a modelling study"**

submitted for publication to *Biogeosciences*
by A. Morée and co-authors

**1   General comments**

**1.1   Appreciation of the replies to reviewers**

The authors have all in all well responded to the referees' comments, with two exceptions.

1. The questions raised by Anonymous Referee #2 about the weathering fluxes have only partly been addressed. Except for the average $\delta^{13}$C signature of the DIC input from weathering, there is still no information regarding the *quantitative* constraints upon which the adopted weathering fluxes have been chosen (it would also be good to know their values).

2. In the response to Comment 2.2.3, I read that "In an open system, the sediment loss of nutrients and carbon over time will empty the whole ocean of nutrients, which would not give very meaningful results." This cannot be correct. As explained earlier on, $PO_4^{3-}$ is replenished by the (constant) weathering input. Upon perturbation, the model system adjusts its carbonate, organic matter and opal production and burial fluxes to evolve towards a new steady state where burial fluxes again match the input fluxes. The ocean will not run empty, as the production decreases with decreasing nutrient concentrations thus reducing the burial rates.

**1.2   Appreciation of the revised manuscript**

The authors have rewritten their manuscript to a large extent. The language has gained in precision and the presentation and discussion in depth. The literature review is adequate now and the study is much better brought into context with previous studies.

The model description has been improved. The separation between surface and deep ocean waters has been revised and is more consistent now. Extra details are now provided in the strongly extended Supplementary Information, which also includes extra graphs with relevant results.

The discussion has been revised and manuscript now also includes a figure with C-13 isotopic profiles for different ocean basins.

However, in some instances, the discussion of the results remains unnecessarily speculative: it should be possible to derive far more quantitative insight from

the model results. The analysis of the results ought to go further than currently done. It is, e. g., incomprehensible that the diagnosed decoupling of the deep-sea $PO_4^{3-}$ concentration from $\delta^{13}C$ and $\Delta\delta^{13}C$ in the Southern Ocean nutrient depletion experiment is not analysed any further. I encourage the authors to better make out the reasons for this decoupling, because, as they – correctly –it offers an alternative to the usual proxy interpretation of deviations from the $\delta^{13}C:PO_4^{3-}$ relationship.

Finally, there are several troublesome errors in the paper. It is impossible that the weathering input of DIC has an average $\delta^{13}C$ signature of 14‰; it is also impossible for the ocean to degas $CO_2$ where the surface ocean $pCO_2$ is lower than $pCO_2^{atm}$; figures document $CO_2$ exchange fluxes in ice-covered regions, whereas the text states that ice cover blocks gas exchange. Please find details in the Specific Comments below.

Given these fundamental errors and inconsistencies in the paper — which unfortunately shed doubt on the validity of the rest of the paper as well — I cannot but ask for another major revision.

**2 Specific comments**

**2.1 Model calibration/spinup and control state**

**2.1.1 $\delta^{13}C$ of weathering DIC input**

In the manuscript (page 4, line 7 and also in the response to Comment 2.2.3), we read that "The $^{13}C/^{12}C$ ratio in the weathering flux would be equivalent to a $\delta^{13}C$ of DIC of 14‰." There is something wrong with this 14‰ value. It is first of all completely unrealistic. The total sedimentary carbon subject to continental weathering has an average $\delta^{13}C$ of $-5$‰; the most abundant carbonate source has a $\delta^{13}C$ of $1.8 \pm 0.2$‰ (Derry and France-Lanord, 1996). Second, at steady state, this input requires a sink (or a combination of sinks) with globally equivalent characteristics. There is however no realistic combination of carbonate and organic carbon sinks that could lead to such a high average $\delta^{13}C$.

**2.1.2 Control run: ocean-atmosphere $pCO_2$ gradient and air-sea $CO_2$ flux**

There are contradictions in the reported results for air-sea exchange of $CO_2$ in the control run: in Fig. S4 (Supplementary Information p. 4), we see that there is a tongue-shaped area extending into the Atlantic Ocean in the Northern Hemisphere where the air-sea-flux of $CO_2$ is positive, while the $p_{CO_2}^{oc} - p_{CO_2}^{atm}$ difference is negative there, as can be seen in Fig. S3. This is a big flaw. Please check this!

**2.1.3 Description and experimental design**

Although the description of the model spin-up procedure has been improved it still lacks many important details. Except for the carbon isotopic signature of

the weathering flux, it is still not specified on what quantitative basis the weathering fluxes have been determined. We now read that there are weathering fluxes for DIC, alkalinity, phosphate and silica, and that these are fed in at a fixed stoichiometric ratio, but that is all that is provided as information (why not simply report de values of the fluxes?).

**2.2 Results and discussion**

**2.2.1 Air-sea gas exchange rate experiments**

This section remains one of the weakest of the study. Some parts are formulated in a vague qualitative style and provide little quantitative information (page 4, lines 5–20); others are long-winding, difficult to read and understand (page 4, lines 21–31). All in all the presentation and discussion of the results remain superficial and give an impression of a half-done job.

First of all, the $pCO_2^{atm}$ results presented in this section are to some extent counter-intuitive: increasing the gas exchange rate makes $pCO_2^{atm}$ increase; reducing the same rate makes $pCO_2^{atm}$ increase as well. In general, common sense would expect that opposite changes of the value of a model parameter lead to opposite effects, possibly except with strongly non-linear models; but even with non-linear models, this expectation should be met for sufficiently small perturbations of parameter values, except in very peculiar situations. A factor of four change can probably not be considered a small variation, so some non-linear behaviour has to be expected. In any case, I would consider deviations from this global scheme very suspect and would proceed to an in-depth analysis, starting with smaller perturbations (e.g., 50%, a factor of two) – finally there is no compelling reason for the particular choice of a factor of four. Here, these striking results are reported without any further ado. Strange enough, $\delta^{13}C^{atm}$ appears to behave as expected: increasing the gas exchange rate reduces $\delta^{13}C^{atm}$; decreasing gas exchange rates increases $\delta^{13}C^{atm}$.

Second, the discussion and analysis of the results consider only one half of the situation. The $pCO_2^{atm}$ increase at increased gas exchange rates is ascribed to a weaker Southern Ocean carbon sink. Unfortunately, not a single quantitative flux value is given to support this claim! Let us apply the reasoning to the rest of the world, following the same logic. At steady state (or, after 2000 years of simulation, at quasi steady-state), the global net exchange of $CO_2$ between the atmosphere and the ocean must be zero. If the perturbation experiment results present a weaker carbon sink Southern Ocean than the control run, it should also present a weaker carbon *source* elsewhere, most likely at low latitude. Following the same logic as before, a weaker carbon source to the atmosphere would be responsible for a *lower* $pCO_2^{atm}$. The conclusions that can be drawn from the kind of semi-qualitative argument that this discussion is based upon thus appear to be ambiguous, i. e., useless. Or there could be a stronger sink elsewhere (not mentioned though). Looking at sources and sinks is probably not the most reliable way to make out the mechanisms at work. What is sure, though, is that the global net exchange of $CO_2$ between the ocean and the atmosphere is zero at steady state. Assuming a globally uniform $pCO_2^{atm}$, we

then have

$$\sum_{i,j} A_{ij}(k_w)_{ij}(pCO_2^{atm} - (pCO_2^{oc})_{ij}) = 0 \qquad (1)$$

where $i$ and $j$ denumber the grid elements, $A_{ij}$ is the surface area and $(k_w)_{ij}$ the specific gas exchange coefficient at grid point $(i, j)$. Accordingly,

$$pCO_2^{atm} = \frac{\sum_{i,j} A_{ij}(k_w)_{ij}(pCO_2^{oc}{}_{ij})}{\sum_{i,j} A_{ij}(k_w)_{ij}}. \qquad (2)$$

This holds as is for steady state under any of the air-sea exchange experiments with globally perturbed $(k_w)_{ij}$'s: if these are uniformly increased by a factor of 4 (or any other value), that value cancels out. The key to understanding the $pCO_2^{atm}$ changes lies thus in the distribution of $pCO_2^{oc}$, which in turn depends on the distribution of DIC and TA (assuming constant temperature and salinity). Gas exchange perturbations should have negligible impact on the TA distribution as long as the ocean-sediment exchange has not started to respond. So, it would be instructive to analyse how the surface ocean DIC distribution has changed.

Third, referring in this context to Fig. S4 to support the reduced Southern Ocean sink argument adds further confusion and is to some extent misleading. In Fig. S4 ice cover has not been taken into account! This is really a terrible shortcoming of that figure! As stated elsewhere in the text, the Southern Ocean south of 60 °S is permanently covered by ice, which completely blocks gas exchange. The green band in the Southern Ocean has actually no meaning (it should actually not be there). In the Southern Ocean north of 60 °S I am, unfortunately, not able to make out any significant differences between the exchange rates of $CO_2$ in the three panels. The total SO sink of atmospheric $CO_2$ appears to be quite stable to me—please feel free to prove me wrong, with adequate flux numbers, which could certainly be easily derived from the model results.
I do, however, see marked differences at low latitudes and they should be quantified in the discussion (e.g., integrated over a zonal band in the basin). And, by the way, it is not clear to me why Fig. S4 represents the situation after 100 instead of 2000 years.

To shed light on this confused situation, I have done some simulation experiments on my own with an ocean carbon cycle model, albeit of lower complexity than HAMOCC2s. First, I have performed a 120,000 yr control run, which was then continued by two 50,000 yr perturbation runs, mimicking fast gas and slow gas (using perturbations of the gas exchange constant by a factor of four). For what the results are worth and as food for thought, here is a summary of the results:

| | after 2,000 yr | | | after 50,000 yr | | |
|---|---|---|---|---|---|---|
| Experiment | $pCO_2^{atm}$ | $\delta^{13}C^{atm}$ | $\delta^{13}C^{avg}$ | $pCO_2^{atm}$ | $\delta^{13}C^{atm}$ | $\delta^{13}C^{avg}$ |
| Gas slow (/4) | 289.4 | −6.14 | 0.24 | 290.5 | −6.29 | 0.15 |
| Control | 282.4 | −6.53 | 0.25 | 282.4 | −6.53 | 0.25 |
| Gas fast (×4) | 278.4 | −6.77 | 0.26 | 277.7 | −6.63 | 0.39 |

$pCO_2^{atm}$ is reported in ppm and $\delta^{13}C$ in ‰; $\delta^{13}C^{avg}$ is the ocean-atmosphere average $\delta^{13}C$; control run extended at steady-state.

As expected, opposite perturbations of the rate constant produce effects in opposite directions. After 2,000 years, about 80% of the $pCO_2^{atm}$ difference to 50,000 years is reached. There is some significant change in $\delta^{13}C^{atm}$ beyond 2000 years, related to global ocean $\delta^{13}C$ adjustments towards the new steady state: in the gas-fast experiment, low-latitude surface ocean $\delta^{13}C$ is reduced by 0.23–0.26‰ after 2,000 years, leading to the burial of carbon with lower $\delta^{13}C$ than at the end of the control run, and thus gradually increasing the system average $\delta^{13}C$. In the gas-slow experiment, the opposite happens (after 2,000 years, the low latitude surface ocean $\delta^{13}C$ is increased by 0.1–0.2‰, and as result of burial changes global ocean $\delta^{13}C$ decreases).

I would really recommend to run all of the experiments far beyond 2000 years. So many interesting things happen once the sediment feedback is allowed to come into play. . .

**2.2.2 The biological pump: SO nutrient depletion**

On page 10 (lines 14–17) we read that

> This is interesting in light of glacial proxy interpretation, as deviations from the $\delta^{13}C:PO_4^{3-}$ relationship (Sect. 2) are usually interpreted as the influence of air-sea gas exchange on $\delta^{13}C$ (Eide et al., 2017b; Lear et al., 2016), but could thus also come from changes in nutrient uptake efficiency. As for a changed POC sinking rate, $\Delta\delta^{13}C$ is affected more in older waters (Fig. 4).

This is not only *interesting*. I would rate this as the *most important outcome* of this study. It is, unfortunately, not followed any further. No attempt is made to analyse this decoupling and to work out the contributing mechanisms.

**2.3 Supplementary Information**

As mentioned in section 2.1.2 above, there are problems with the model results reported in Figs. S3 and S4. Please check this.

**3 Technical comments**

**Manuscript**

**Throughout the paper**: please use the verb "to lower" (and similarly "to reduce" with a positive amount (e.g., "to lower by 1‰", not "to lower by $-1$‰").

**Page 1, line 8**: please add the reference for the standard ratio (Craig, 1957).

**Page 2, line 17**: please explain what a "free box atmosphere" is

**Page 3, line 30**: the correct units for temperature in the parametrization for $\epsilon$ are "K" not "℃" (try to apply it at 0 ℃. . . )

**Page 5, line 11**: strange sentence construction. I suggest to rephrase as "[. . . ] we express $\delta^{13}C_{bio}$ as a percentage (denoted $\delta^{13}C_{bio}^{perc}$ because [. . . ]"

**Page 5, line 16**: the absolute values are superfluous as both $F_{up}$ and $F_{down}$ are positive (cf. SI 1B, and also Heinze and Maier-Reimer (1999))

**Page 5, line 18**: $F_{net}$ should be defined as $F_{net} = F_{up} - F_{down}$ (cf. SI 1B, and also Heinze and Maier-Reimer (1999))

**Page 6, line 25**: replace "results in room" by "leaves room"

**Page 6, line 27**: replace "times slower" by "times more slowly"

**Page 7, line 11**: "(Fig. S4 and 5)" should probably read "(Fig. S4 and S5)"

**Page 7, line 17**: the sentence "Slow gas exchange reduces $F_{u+d}$ causing less fractionation to occur. . . " does not really make sense. Fractionation is dependent on temperature, which remains unchanged. The *contrast* or the *difference* between air and sea is changed, because the air-sea-exchange fluxes play a lesser role in the surface ocean $^{13}$C balance allowing a greater difference between atmosphere and ocean. Please rewrite.

**Page 8, line 25**: "can be"? is it or is it not? If it is, say "is", else please discuss!

**Page 9, line 2**: replace "is more confined to" by "is confined closer to"

**Page 9, line 27**: "probably"? Why speculate? The model results should allow to verify this.

**Page 10, line 11**: replace "(up to $-0.8$‰)" by "(by up to $0.8$‰)"

**Page 11, line 5**: "course" should read "source"

**Page 11, line 6**: replace "earlier" by "previously"

**Page 11, line 10**: Fig. S6 relates to the slow gas experiment. Not sure this is the one to refer to here.

**Page 11, line 13**: replace "with" by "by"

**Page 11, line 27**: replace "advance" by "spread" or "extend"

**Page 11, line 28**: replace "increased up to" by "increased by up to"

**Page 12, line 1**: replace "varies up to $\sim\pm0.4$‰" by "varies by up to about $\pm0.4$‰" (do not place two symbols immediately one after the other)

**Page 13, line 19**: replace "varies up to $\sim\pm0.5$‰" by "varies by up to about $\pm0.5$‰"

**Page 13, line 21**: "is important" should read "are important"

**Supplementary Information**

**Page 2**: there is much confusion and there are several errors in this paragraph. To bring it in line with the graphs, the model documentation (Heinze and Maier-Reimer, 1999) and the main paper, it needs to be corrected as follows:

- $F_{up} = k_w * [A]_{water}$

- $F_{down} = k_w * [A]_{air}$

- $F_A = k_w * ([A]_{water} - [A]_{air})$

**Page 4, caption to Fig. S3**: what exactly is meant by "Negative values indicate a potential carbon flux to the ocean?" Why "potential"?

**Page 4, caption to Fig. S4**: not sure what the integrated fluxes are meant to tell. Is there something special after 100 years? We try to understand the state after 2000 years. Why is the state after 2000 years not shown here instead?

**Page 4, Figs. S3 and S4**: unfortunately, the color scales chosen here are somewhat misleading. On Fig. S3, the rich green tone covers the first negative interval (i.e., the one next to 0) while on Fig. S4, it covers the first positive interval.

**Page 5, Fig. S5**: It would be best if all the figures had the same vertical axis extent. If this is not possible, at least graphs appearing side by side should have the same extents (both $pCO_2$ and $\delta^{13}C$ axes).

**References**

Craig, H.: Isotopic standards for carbon and oxygen and correction factors for mass-spectrometric analysis of carbon dioxide, Geochim. Cosmochim. Ac., 12, doi:10.1016/0016-7037(57)90024-8, 1957.

Derry, L. A. and France-Lanord, C.: Neogene growth of the sedimentary organic carbon reservoir, Paleoceanography, 11, 267–275, doi:10.1029/95PA03839, 1996.

Heinze, C. and Maier-Reimer, E.: The Hamburg Oceanic Carbon Cycle Circulation Model version "HAMOCC2s" for long time integrations, Technical Report 20, Deutsches Klimarechenzentrum, Hamburg (DE), available at `https://www.dkrz.de/mms/pdf/reports/ReportNo.20.pdf`, 1999.

---

## Referee Report (RR2)

Review of the second revised version of

**"Southern Ocean controls of the vertical marine $\delta^{13}$C gradient – a modelling study"**

submitted for publication to *Biogeosciences*
by A. Morée and co-authors

I have gone through the authors' responses to the my comments from the second review. All in all, the responses appear to be satisfactory. The required additional information (weathering fluxes etc.) have been provided. The $\delta^{13}$C of the weathering flux at $-11\,‰$ (from $+14‰$) is still somewhat peculiar: as mentioned in my previous review, the total sedimentary carbon subject to weathering has an average $\delta^{13}$C of $-5‰$, wherein the most abundant source are carbonate rocks with a $\delta^{13}$C of around $1.8‰$. Perhaps the amount of organically derived weathering DIC is larger than in reality (this would be the case the shelf carbonate sink is not considered). At least $-11\,‰$ can be more easily explained than $+14\,‰$. The model apparently included some errors that have now been corrected, making the results more plausible now.

I regret that the authors still have not changed their mind about the duration of the perturbation experiments.

I am, however, truly disappointed about the fact that even this second major (!) revision is not devoid of its share of confusion.

The previous version included in its Supplement Figures S3 and S4:

- Fig. S3 represented the surface ocean $\Delta p_{CO_2}$ for the control run;

- Fig. S4 represented the specific air-to-sea exchange flux of $CO_2$ for the control run, the Fast gas and the Slow gas experiments.

These two figures presented several deficiencies:

1. inconsistent colour scales

2. physically incompatible results

According to the "Author's response", the incompatibility between the results (deficiency (2)) was due to a plotting error during the production of Fig. S3. Regarding deficiency (1), we read on the 12th page of the authors' response (page numbers in the authors' response would have been the reviewers best friend . . . ) that "We corrected this in the new Fig. S4 so that both figures now have the first positive interval in green." Unfortunately

- the revised Supplement contains only one figure with these informations (which other one could possibly be the second of "both" having the "first positive interval in green"?);

- this is the new Fig. S4, which represents the surface ocean $\Delta p_{CO_2}$ for the control run (formerly shown on the former Fig. S3) and the Fast gas and the Slow gas experiments.

So, the new Fig. S4 actually includes the old Fig. S3 as its left panel and has equivalent panels added for the two perturbation experiments. The old Fig. S4 has been discarded. At least the old Fig. S4 (specific $CO_2$ exchange rates) and the new Fig. S4 ($\Delta p_{CO_2}$) are compatible (at first sight), but to check this, one has to compare graphs carrying different information in different revisions of the Supplement.

None of these comments is meant to be a showstopper and I am ready to give green light for the publication. I leave it to the editor to decide on whether the inconsistency between the figures in the Supplement and the Author's response needs correction or not.

---

## Author Response (AR2)

Dear Professor Joos, dear reviewer,

Thank you for your thorough feedback on our revised manuscript from the 5$^{th}$ of June, 2018. We very much appreciate the time and effort you have spent on our manuscript. We addressed each comment and concern, and hereby present our newly revised version. Please, find details about the changes made relative to the last version of the manuscript in the responses below and in the marked-up manuscript.

The author's response is presented as follows:
    (1) **Referee report**: comments from the reviewer report with author's response to each comment and summary of author's changes in the manuscript,
    (2) **Editor comments**: comments from the editor with author's response to each comment and summary of author's changes in the manuscript,
    (3) **Marked-up manuscript**: Author's changes tracked in the new manuscript

Yours sincerely, Anne Morée and co-authors

**Referee report**

**Comment 1.1) Appreciation of the replies to reviewers**

*The authors have all in all well responded to the referees' comments, with two exceptions.*
*1. The questions raised by Anonymous Referee #2 about the weathering fluxes have only partly been addressed. Except for the average δ 13C signature of the DIC input from weathering, there is still no information regarding the quantitative constraints upon which the adopted weathering fluxes have been chosen (it would also be good to know their values).*
*2. In the response to Comment 2.2.3, I read that "In an open system, the sediment loss of nutrients and carbon over time will empty the whole ocean of nutrients, which would not give very meaningful results." This cannot be correct. As explained earlier on, PO3− 4 is replenished by the (constant) weathering input. Upon perturbation, the model system adjusts its carbonate, organic matter and opal production and burial fluxes to evolve towards a new steady state where burial fluxes again match the input fluxes. The ocean will not run empty, as the production decreases with decreasing nutrient concentrations thus reducing the burial rates.*

**Author response to Comment 1.1)**

Thank you for providing us with a clear review of our latest manuscript. The two exceptions mentioned here in comment 1.1 are addressed by:
1. Extending the Methods (Sect 2) text on the weathering fluxes with stoichiometric ratios, absolute fluxes and a comparison to observational estimates (see also our reply to comment 2.1.1)
2. Our formulation of our reply was indeed not detailed enough and therefore misleading/incorrect. As this mistake is not made in the manuscript, no further large changes have been made there. We did add the word 'transient' (POC sinking rate experiments) to clarify that the imbalance between burial and weathering is temporary. What we meant to write in our previous response is that there will be a decrease of nutrients until a new equilibrium is established based on the weathering fluxes, such that input and burial are in mass balance. As this equilibration takes hundreds of thousands of years (Roth et al., 2014) for $\delta^{13}C$, we decided to present transient results for the POC experiments, as reasoned for in our previous reply.

**Author's changes in the manuscript in response to Comment 1.1)**

- The methods text on weathering is extended to include the missing information on that subject.
- The word 'transient' has been added to further stress that the weathering-burial imbalance of the POC experiments is temporary

**Comment 1.2) Appreciation of the revised manuscript**

*The authors have rewritten their manuscript to a large extent. The language has gained in precision and the presentation and discussion in depth. The literature review is adequate now and the study is much better brought into context with previous studies. The model description has been improved. The separation between surface and deep ocean waters has been revised and is more consistent now. Extra details are now provided in the strongly extended Supplementary Information, which also includes extra graphs with relevant results. The discussion has been revised and manuscript now also includes a figure with C-13 isotopic profiles for different ocean basins. However, in some instances, the discussion of the results remains unnecessarily speculative: it should be possible to derive far more quantitative insight from the model results. The analysis of the results ought to go further than currently done. It is, e. g., incomprehensible that the diagnosed decoupling of the deepsea PO34- concentration from δ 13C and Δδ 13C in the Southern Ocean nutrient depletion experiment is not analysed any further. I encourage the authors to better make out the reasons for this decoupling, because, as they – correctly –it offers an alternative to the usual proxy interpretation of deviations from the δ 13C:PO3− 4 relationship. Finally, there are several troublesome errors in the paper. It is impossible that the weathering input of DIC has an average δ 13C signature of 14 ‰; it is also impossible for the ocean to degas CO2 where the surface ocean pCO2 is lower than pCOatm; figures document CO2 exchange fluxes in ice-covered regions, whereas the text states that ice cover blocks gas exchange. Please find details in the Specific Comments below. Given these fundamental errors and inconsistencies in the paper — which unfortunately shed doubt on the validity of the rest of the paper as well — I cannot but ask for another major revision.*

**Author response to Comment 1.2)**

Thank you for again taking the time to thoroughly review our manuscript. We added more quantitative descriptions of our results throughout the manuscript. We extended our discussion of the Vmax experiment (the $\delta^{13}C:PO_4^{3-}$ deviation), see our reply to referee comment 2.2.2 for more details. The weathering signature of $\delta^{13}C$ is revised (details given in our reply to referee comment 2.1.1) and the air-sea $p$CO$_2$ difference figure (original Fig. S3, now Fig. S4) is corrected (details in our reply to referee comment 2.1.2) after we were made aware of the discrepancy between original Fig. S3 and S4. The text on light and gas penetration through sea ice now better describes the difference between the model treatment of light and gas inhibition through sea ice (Sect. 2 as well as 3.3.4). We also provided more quantitative information such as fluxes in section 3.3.1, see our reply to referee comment 2.2.1 for more details.

**Author's changes in the manuscript in response to Comment 1.2)**

As described in detail in our reply to the specific comments below

**Comment 2.1) Model calibration/spinup and control state**

*2.1.1 δ¹³C of weathering DIC input*

*In the manuscript (page 4, line 7 and also in the response to Comment 2.2.3), we read that "The 13C/12C ratio in the weathering flux would be equivalent to a δ 13C of DIC of 14‰." There is something wrong with this 14‰ value. It is first of all completely unrealistic. The total sedimentary carbon subject to continental weathering has an average δ 13C of −5‰; the most abundant carbonate source has a δ 13C of 1.8 ± 0.2‰ (Derry and France-Lanord, 1996). Second, at steady state, this input requires a sink (or a combination of sinks) with globally equivalent characteristics. There is however no realistic combination of carbonate and organic carbon sinks that could lead to such a high average δ 13C.*

*2.1.2 Control run: ocean-atmosphere pCO2 gradient and air-sea CO2 flux*

*There are contradictions in the reported results for air-sea exchange of CO2 in the control run: in Fig. S4 (Supplementary Information p. 4), we see that there is a tongue-shaped area extending into the Atlantic Ocean in the Northern Hemisphere where the air-sea-flux of CO2 is positive, while the pCO2oc − pCO2atm difference is negative there, as can be seen in Fig. S3. This is a big flaw. Please check this!*

*2.1.3 Description and experimental design*

*Although the description of the model spin-up procedure has been improved it still lacks many important details. Except for the carbon isotopic signature of the weathering flux, it is still not specified on what quantitative basis the weathering fluxes have been determined. We now read that there are weathering fluxes for DIC, alkalinity, phosphate and silica, and that these are fed in at a fixed stoichiometric ratio, but that is all that is provided as information (why not simply report de values of the fluxes?).*

**Author response to Comment 2.1)**

2.1.1 The 14 ‰ value was a mistake in the calculation, and we corrected this to -11 ‰, besides elaborating in general on the weathering fluxes and making a comparison with observational estimates. The 14 ‰ value was never in the model, so results are not affected. The new text describes that weathering fluxes are added homogeneously over the first ocean layer as dissolved matter and in a fixed stoichiometric ratio for $CaCO_3$, organic carbon, $PO_4^{3-}$, Alkalinity, and Si. Annual weathering fluxes (Tmol) are 27 for $CaCO_3$, 5 for organic carbon, 5/122 for $PO_4^{3-}$, $2*CaCO_3-16* PO_4^{3-}$ for Alkalinity, and 4.5 for Si. These values are within the uncertainties of observational estimates for Si (5.6 Tmol yr-1 (Tréguer, 2002)), $CaCO_3$ (~32 Tmol yr-1 (Milliman et al., 1996)), and organic carbon (4 Tmol yr-1 (Broecker and Peng, 1987)), and have been adjusted to improve the fit of the respective modelled marine tracer distributions as well as burial fluxes to observational estimates.

2.1.2 We thank the reviewer for spotting this, comparison of original figures S3 and S4 is indeed ground for confusion for any potential reader. The model results were correct, and we found that the mismatch between original Figures S3 and S4 was caused by a plotting error in making the original Fig. S3. The values reported in the text and analysis are not affected by this, so we expect that our replacement of Fig. S3 to a correct version (Fig. S4) will take away your concerns. We also added two extra subfigures to Fig. S4, for the gas exchange rate experiments, to clarify that section.

2.1.3. We extended the Methods section to include the stoichiometric ratios, the absolute fluxes in Tmol, and a comparison with observational estimates to improve the description of the weathering fluxes. See our response to comment 2.1.1.

**Author's changes in the manuscript in response to Comment 2.1)**

2.1.1 and 2.1.3: See changes in the text in Section 2
2.1.2: Fig. S3 replaced by correct version (Fig. S4), and extended to include gas exchange rate experiments

**Comment 2.2) Results and discussion**

*Comment 2.2.1 Air-sea gas exchange rate experiments*
*This section remains one of the weakest of the study. Some parts are formulated in a vague qualitative style and provide little quantitative information (page 4, lines 5–20); others are long-winding, difficult to read and understand (page 4, lines 21–31). All in all the presentation and discussion of the results remain superficial and give an impression of a half-done job. First of all, the pCO2atm results presented in this section are to some extent counter-intuitive: increasing the gas exchange rate makes pCO2atm increase; reducing the same rate makes pCO2atm increase as well. In general, common sense would expect that opposite changes of the value of a model parameter lead to opposite effects, possibly except with strongly non-linear models; but even with non-linear models, this expectation should be met for sufficiently small perturbations of parameter values, except in very peculiar situations. A factor of four change can probably not be considered a small variation, so some non-linear behaviour has to be expected. In any case, I would consider deviations from this global scheme very suspect and would proceed to an in-depth analysis, starting with smaller perturbations (e.g., 50%, a factor of two) – finally there is no compelling reason for the particular choice of a factor of four. Here, these striking results are reported without any further ado. Strange enough, δ13Catm appears to behave as expected: increasing the gas exchange rate reduces δ13Catm; decreasing gas exchange rates increases δ13C atm.*

*Second, the discussion and analysis of the results consider only one half of the situation. The pCO2atm increase at increased gas exchange rates is ascribed to a weaker Southern Ocean carbon sink. Unfortunately, not a single quantitative flux value is given to support this claim! Let us apply the reasoning to the rest of the world, following the same logic. At steady state (or, after 2000 years of simulation, at quasi steady-state), the global net exchange of CO2 between the atmosphere and the ocean must be zero. If the perturbation experiment results present a weaker carbon sink Southern Ocean than the control run, it should also present a weaker carbon source elsewhere, most likely at low latitude. Following the same logic as before, a weaker carbon source to the atmosphere would be responsible for a lower pCO2atm. The conclusions that can be drawn from the kind of semi-qualitative argument that this discussion is based upon thus appear to be ambiguous, i. e., useless. Or there could be a stronger sink elsewhere (not mentioned though). Looking at sources and sinks is probably not the most reliable way to make out the mechanisms at work. What is sure, though, is that the global net exchange of CO2 between the ocean and the atmosphere is zero at steady state. Assuming a globally uniform pCO2atm, we then have*

$$\sum_{i,j} A_{ij}(k_w)_{ij}(pCO_2^{atm} - (pCO_2^{oc})_{ij}) = 0 \qquad (1)$$

*where i and j denumber the grid elements, Aij is the surface area and (kw)ij the specific gas exchange coefficient at grid point (i, j). Accordingly,*

$$pCO_2^{atm} = \frac{\sum_{i,j} A_{ij}(k_w)_{ij}(pCO_2^{oc}{}_{ij})}{\sum_{i,j} A_{ij}(k_w)_{ij}}. \qquad (2)$$

*This holds as is for steady state under any of the air-sea exchange experiments with globally perturbed (kw)ij 's: if these are uniformly increased by a factor of 4 (or any other value), that value cancels out. The key to understanding the pCO2atm changes lies thus in the distribution of pCO2oc, which in turn depends on the distribution of DIC and TA (assuming constant temperature and salinity). Gas exchange perturbations should have negligible impact on the TA distribution as long as the ocean-sediment exchange has not started to respond. So, it would be instructive to analyse how the surface ocean DIC distribution has changed. Third, referring in this context to Fig. S4 to support the reduced Southern Ocean sink argument adds further confusion and is to some extent misleading. In Fig. S4 ice cover has not been taken into account! This is really a terrible shortcoming of that figure! As stated elsewhere in the text, the Southern Ocean south of 60 °S is permanently covered by ice, which completely blocks gas exchange. The green band in the Southern Ocean has actually no meaning (it should actually not be there). In the Southern Ocean north of 60 °S I am, unfortunately, not able to make out any significant differences between the exchange rates of CO2 in the three panels. The total SO sink of atmospheric CO2 appears to be quite stable to me—please feel free to prove me wrong, with adequate flux numbers, which could certainly be easily derived from the model results. I do, however, see marked differences at low latitudes and they should be quantified in the discussion (e.g., integrated over a zonal band in the basin). And, by the way, it is not clear to me why Fig. S4 represents the situation after 100 instead of 2000 years.*

*To shed light on this confused situation, I have done some simulation experiments on my own with an ocean carbon cycle model, albeit of lower complexity than HAMOCC2s. First, I have performed a 120,000 yr control run, which was then continued by two 50,000 yr perturbation runs, mimicking fast gas and slow gas (using perturbations of the gas exchange constant by a factor of four). For what the results are worth and as food for thought, here is a summary of the results:*

| | after 2,000 yr | | | after 50,000 yr | | |
|---|---|---|---|---|---|---|
| Experiment | $pCO_2^{atm}$ | $\delta^{13}C^{atm}$ | $\delta^{13}C^{avg}$ | $pCO_2^{atm}$ | $\delta^{13}C^{atm}$ | $\delta^{13}C^{avg}$ |
| Gas slow (/4) | 289.4 | −6.14 | 0.24 | 290.5 | −6.29 | 0.15 |
| Control | 282.4 | −6.53 | 0.25 | 282.4 | −6.53 | 0.25 |
| Gas fast (×4) | 278.4 | −6.77 | 0.26 | 277.7 | −6.63 | 0.39 |

$pCO_2^{atm}$ is reported in ppm and $\delta^{13}C$ in ‰; $\delta^{13}C^{avg}$ is the ocean-atmosphere average $\delta^{13}C$; control run extended at steady-state.

*As expected, opposite perturbations of the rate constant produce effects in opposite directions. After 2,000 years, about 80% of the pCOatm 2 difference to 50,000 years is reached. There is some significant change in δ 13C atm beyond 2000 years, related to global ocean δ 13C adjustments towards the new steady state: in the gas-fast experiment, low-latitude surface ocean δ 13C is reduced by 0.23–0.26 ‰ after 2,000 years, leading to the burial of carbon with lower δ 13C than at the end of the control run, and thus gradually increasing the system average δ 13C. In the gas-slow experiment, the opposite happens (after 2,000 years, the low latitude surface ocean δ 13C is increased by 0.1–0.2 ‰, and as result of burial changes global ocean δ 13C decreases). I would really recommend to run all of the experiments far beyond 2000 years. So many interesting things happen once the sediment feedback is allowed to come into play. . .*

**Author response to Comment 2.2.1)**

Thank you for your elaborate comment on the air-sea gas exchange section, and helping us to understand remaining problems with this section. We discovered a problem in the gas exchange routine in the model that explains the absence of a divergence of pCO2 in the gas fast vs. gas slow experiments: The gas exchange routine was corrected for giving better convergence for the computation of the first guess value of pCO2 at the new time step (Due to the time step of one year, one needs to use an average of the $p$CO$_2$ from the old time step and an extrapolation of $p$CO$_2$ at the new time step for potential full

equilibration with the atmospheric $p\text{CO}_2$ -There was a bug in the original scheme for this extrapolation). With this modification, the correct – small – atmospheric $p\text{CO}_2$ variation for lowered gas transfer velocity is now also achieved. All model runs have been repeated with this modification to be fully consistent with each other. All Figures and values have thus been updated throughout the text. As you can see, the differences are small, and have had no effect on any conclusions involving (the vertical gradient of) $\delta^{13}C$, the main topic of our study. Importantly, the results for $p\text{CO}_2$ are now diverging for the gas exchange experiments, which is to be expected from the experiments - as the reviewer argued. We rewrote the air-sea gas exchange section to clarify the points raised in reviewer comment 2.2.1 and 2.1.2. We added values for the air-sea fluxes wherever relevant, or gave the change relative to the control run as a percentage if this aids readability. Our description is supported by an extended Fig. S4 for the $p\text{CO}_2$ gradient across the air-sea interface. Our choice of a factor 4 is reasoned for in the text in the last sentence of the first paragraph of section 3.3.1. In equilibrium, global net exchange is indeed zero as described in Eq. (1) in the reviewer comment 2.2.1, but $p\text{CO}_2$ changes only in the transient phase when Eq. (1) is non-zero - which happens in approximately the first 600 years as visible from Fig. S3. This is also why we initially presented results (in for example the original Fig. S4) during the transient phase at 100 years. As this clearly led to confusion, we removed the original Fig. S4 from the SI (but still provide it here in our reply, see our response to Comment 3) and restricted the section to describing 2000 year results. Also, note that the air-sea $p\text{CO}_2$ difference figure shows the combined result of the change in surface ocean $p\text{CO}_2$ and atmospheric $p\text{CO}_2$. We clarified the presence of a small air-sea C flux through the sea ice by extending the methods description: Here, we included the difference between light (full inhibition when ice is present) and gas transfer (partial inhibition depending on sea ice thickness). The green band in the original Fig. S4 thus had a meaning, as there is a (very) small flux through the ice, because the air-sea gas exchange rate is scaled by division through the ice thickness in cm and thus never fully becomes zero.

The duration of the sensitivity experiments is kept at 2000 years, for which we give a series of reasons: The most important one is that the main response to the applied change occurs within the first few hundred years of the experiment (as visible in Fig. S3). We performed a 50 000 year run for the Gas Fast experiment and compared the results to our previous 2000 year results, and indeed see an additional drift in atmospheric $\delta^{13}C$ and $p\text{CO}_2$ over these additional 48 000 years - of similar size to what the reviewer has presented us for his or her model. Another major reason for not continuing the experiments beyond 2000 years (except for the already presented 12000 year POC fast experiment) is that the calculated $\Delta\delta^{13}C$ is already equilibrated after 2000 years (no difference between 2000 and 50000 year result), and $\Delta\delta^{13}C$ is the topic of our study. Besides that, other processes would come into play on such long timescales as already argued in the first paragraph of section 3.3.2. A final argument against doing longer experiments is that choosing an experiment duration of for example 50 000 years would still not be enough to equilibrate $\delta^{13}C$ (as mentioned in the text and shown by Roth et al. (2014) who had a continuing $\delta^{13}C$ drift beyond 200 000 years). One should thus consider any $\delta^{13}C$ distribution to be only in quasi-equilibrium, and we choose the 2000-year $\delta^{13}C$ distribution (that has incorporated its main initial, 'fast', response to our perturbations). We stressed this by adding the word 'quasi-equilibrium' in the Methods experiment description and in several other places where it is relevant, and by extending the caption of Fig. S3.

We tested smaller perturbations of the gas exchange rate as well. The effects are as expected smaller (and diverging, from the corrected gas exchange routine). We however choose to present the more extreme cases of a very high (4x) and low (division by 4) gas exchange rate in the text: We do this in order to show the reader the potential effects of bringing surface ocean $\delta^{13}C$ very far away or very close to its equilibrium with the atmosphere – stressing the contrast between low and high latitudes.

**Comment 2.2) Results and discussion**

*Comment 2.2.2 The biological pump: SO nutrient depletion*
*On page 10 (lines 14–17) we read that This is interesting in light of glacial proxy interpretation, as deviations from the $\delta 13C:PO3-4$ relationship (Sect. 2) are usually interpreted as the influence of air-sea gas exchange on $\delta 13C$ (Eide et al., 2017b; Lear et al., 2016), but could thus also come from changes in nutrient uptake efficiency. As for a changed POC sinking rate, $\Delta\delta13C$ is affected more in older waters (Fig. 4). This is not only interesting. I would rate this as the most important outcome of this study. It is, unfortunately, not followed any further. No attempt is made to analyse this decoupling and to work out the contributing mechanisms.*

**Author response to Comment 2.2.2)**

We understand the need for a more elaborate explanation on this aspect. In order to clarify our statements, we have adjusted the precision of language in this part of Sect. 3.3.3 and provided a new Figure on the relationship between $PO_4^{3-}$ and $\delta^{13}C$ in the different experiments (Fig. 8). From this new figure, it is visible that both changes in air-sea gas exchange and changes in biological uptake in the SO can change the global goodness-of-fit to the $\delta^{13}C:PO_4^{3-}$ relationship. The fit thus depends on the relative importance of biology and air-sea gas exchange (see also Fig. 2) – and should not be interpreted as the influence of air-sea gas exchange on $\delta^{13}C$ alone (as has been done before). The sentence 'As for a changed POC sinking rate, $\Delta\delta^{13}C$ is affected more in older waters (Fig. 4)' has no connection to the deviations from the $\delta^{13}C: PO_4^{3-}$ relationship (as this relationship does not consider the vertical gradient of $\delta^{13}C$), and is moved to improve readability.

**Author's changes in the manuscript in response to Comment 2.2)**

2.2.1 Section 3.3.1 is rewritten in response to the concerns raised by the referee and editor. We also updated all figures and values in the text because all model runs had to be redone after finding a problem with the gas exchange routine (see details in our reply to comment 2.2.1). Besides that, extra text on sea ice gas and light inhibition is added to the Methods in Sect. 2.

2.2.2 We elaborated in Sect. 3.3.3 on the deviation from the $\delta^{13}C:PO_4^{3-}$ relationship, by providing extra text and adding a new figure showing this relationship for a selection of experiments.

**Comment 2.3) Supplementary Information**

*As mentioned in section 2.1.2 above, there are problems with the model results reported in Figs. S3 and S4. Please check this.*

**Author response to Comment 2.3)**

See our reply to comment 2.1.2.

**Author's changes in the manuscript in response to Comment 2.3)**

See our reply to comment 2.1.2.

**Comment 3) Technical comments**

Thank you for noting these technical mistakes. We added our reply to the comments directly below each comment in italicised text in order to improve readability of our corrections.

**Manuscript**

Throughout the paper: please use the verb "to lower" (and similarly "to reduce" with a positive amount (e.g., "to lower by 1‰", not "to lower by −1‰").
> *Corrected throughout manuscript*

Page 1, line 8: please add the reference for the standard ratio (Craig, 1957).
> *Added in text and to reference list*

Page 2, line 17: please explain what a "free box atmosphere" is
> *Rephrased to 'a one-layer atmosphere component assumed to be longitudinally well-mixed' to clarify and updated SI1B to match this adjustment*

Page 3, line 30: the correct units for temperature in the parametrization for eta are "K" not "°C" (try to apply it at 0°C...)
> *Corrected to K*

Page 5, line 11: strange sentence construction. I suggest to rephrase as "[. . . ] we express δ 13Cbio as a percentage (denoted δ 13C perc bio because [. . . ]"
> *Sentence revised*

Page 5, line 16: the absolute values are superfluous as both Fup and Fdown are positive (cf. SI 1B, and also Heinze and Maier-Reimer (1999))
> *We agree that these can be removed, the manuscript has been adjusted accordingly*

Page 5, line 18: Fnet should be defined as Fnet = Fup − Fdown (cf. SI 1B, and also Heinze and Maier-Reimer (1999))
> *This is indeed clearer, our analysis had negative downward fluxes which explained the absolute sign and the Fnet=Fup + Fdown, but now the presentation is clearer and more consistent. Thank you for drawing our attention to this.*

Page 6, line 25: replace "results in room" by "leaves room"
> *Corrected as suggested*

Page 6, line 27: replace "times slower" by "times more slowly"
> *Corrected as suggested*

Page 7, line 11: "(Fig. S4 and 5)" should probably read "(Fig. S4 and S5)"
> *Corrected*

Page 7, line 17: the sentence "Slow gas exchange reduces Fu+d causing less fractionation to occur. . . " does not really make sense. Fractionation is dependent on temperature, which remains unchanged. The contrast or the difference between air and sea is changed, because the air-sea-exchange fluxes play a lesser role in the surface ocean 13C balance allowing a greater difference between atmosphere and ocean. Please rewrite.
> *We see how a reader can misinterpret this sentence as it is not precise, and we have rewritten to increase clarity.*

Page 8, line 25: "can be"? is it or is it not? If it is, say "is", else please discuss!
> *Replaced by 'is'*

Page 9, line 2: replace "is more confined to" by "is confined closer to"
> *Corrected as suggested*

Page 9, line 27: "probably"? Why speculate? The model results should allow to verify this.
> *Corrected*

Page 10, line 11: replace "(up to −0.8‰)" by "(by up to 0.8‰)"
> *Corrected as suggested*

Page 11, line 5: "course" should read "source" Page 11, line 6: replace "earlier" by "previously"
> *Corrected as suggested*

Page 11, line 10: Fig. S6 relates to the slow gas experiment. Not sure this is the one to refer to here.
> *The sign of the change in the Gas Slow experiments equals the sign of the disequilibrium, as mentioned in the caption of Fig. S6. We nevertheless removed the reference to Fig. S6 because we see how it can be confusing when referred to in this section on sea ice.*

Page 11, line 13: replace "with" by "by"
> *Corrected as suggested*

Page 11, line 27: replace "advance" by "spread" or "extend"
> *Corrected*

Page 11, line 28: replace "increased up to" by "increased by up to"
> *Corrected as suggested*

Page 12, line 1: replace "varies up to ~±0.4‰" by "varies by up to about ±0.4‰" (do not place two symbols immediately one after the other)
> *Corrected and also corrected other occurrences of double symbol use throughout the manuscript*

Page 13, line 19: replace "varies up to ~±0.5‰" by "varies by up to about ±0.5‰"
> *Corrected as suggested*

Page 13, line 21: "is important" should read "are important"
> *Corrected as suggested*

**Supplementary Information**

Page 2: there is much confusion and there are several errors in this paragraph. To bring it in line with the graphs, the model documentation (Heinze and MaierReimer, 1999) and the main paper, it needs to be corrected as follows: • Fup = kw ∗ [A]water • Fdown = kw ∗ [A]air • FA = kw ∗ ([A]water − [A]air)
> *Corrected to be in line with the rest of the manuscript, as suggested.*

Page 4, caption to Fig. S3: what exactly is meant by "Negative values indicate a potential carbon flux to the ocean?" Why "potential"?
> *Clarified in the caption.*

Page 4, caption to Fig. S4: not sure what the integrated fluxes are meant to tell. Is there something special after 100 years? We try to understand the state after 2000 years. Why is the state after 2000 years not shown here instead?
> *The globally integrated C flux through the air-sea interface will increase or decrease $pCO_2$ during the transient phase, depending on its sign. After reaching equilibrium, the globally integrated C flux is (very close to) zero, as described in equation (1) presented by reviewer in their report. Atmospheric $pCO_2$ changes only during the transient phase where eq (1) is non-zero, and this effect is most pronounced after ~100 years, which explains our previous focus on this. As the discovery and solving of a problem with the air-sea gas exchange routine now results in the expected divergence of $pCO_2$, we decided to remove the 100-year figures and text. The focus of the manuscript is on $\delta^{13}C$, and we see the reader 'sudden' 100-year results confuse when the rest of the text is on 2000-year results. However, the updated figure (after rerunning the model with the corrected air-sea gas exchange routine) is given below in case the reviewer is interested:*

[Figure]

Page 4, Figs. S3 and S4: unfortunately, the colour scales chosen here are somewhat misleading. On Fig. S3, the rich green tone covers the first negative interval (i.e., the one next to 0) while on Fig. S4, it covers the first positive interval.

> *We corrected this in the new Fig. S4 so that both figures now have the first positive interval in green.*

Page 5, Fig. S5: It would be best if all the figures had the same vertical axis extent. If this is not possible, at least graphs appearing side by side should have the same extents (both $p$CO$_2$ and δ 13C axes).

> *We adjusted the axes in new Fig. S3 a way that graphs appearing side by side now have the same $p$CO$_2$ and $\delta^{13}C$ axes.*

**Editor Comments) Summary of comments that come in addition to the above-presented referee comments**

1. *Address apparent mismatch between Fig. S3 and S4*
2. *Compare weathering fluxes to data-based estimates*
3. *Rewrite section 3.3.1*
4. *Decide on simulation duration*
5. *Make figure captions more specific in Fig. 3 and following and provide information on time of results*
6. *Refer to equilibration timescale of $\Delta\delta^{13}C$*
7. *Page 6, line 19ff (of marked-up MS): The text on the equilibration time is not very precise and the explanation for the longer equilibration time scale of 13C compared to oxygen is in my opinion not correct. Probably you would like to address how long it takes to equilibrate the atmosphere and ocean surface layer by air-sea gas exchange (not considering surface-to-deep exchange) for O2, DIC, and 13C(DIC) after a small perturbation in surface ocean O2, DIC and δ$^{13}$C(DIC), respectively. The equilibration time depends on how fast the tracer amount in the surface layer is replaced by air-sea gas exchange as further moderated by the Revelle factor for DIC.*

> *(For this you may solve the according differential equations:*
> *d/dt x(s) = F(x(s), x(a)) and d/dt x(a) = - F(x(s), x(a))*

*to extract the exponential time scale associated with the re-equilibration. F is the net air to sea flux and x(s) and x(a) the perturbation in the atm. and surface layer) )*

8. *p6, line 25: suggest to delete this sentence as unclear and not precise. (The surface ocean $\delta^{13}$C is not in equilibrium with the atmosphere in many region due to the slow equilibration time scale for $\delta^{13}$C with respect to gas exchange with the atmosphere.)*

9. *p6, line 29: It is highly misleading to state that air-sea equilibrium would be 2 ‰ as the equilibrium depends strongly on temperature and varies greatly with latitude*

Author response and changes in the manuscript in response to Editor Comments)

1. The mismatch between Fig. S3 and S4 was caused by a plotting error in making Fig. S3, we replaced this figure (new Fig. S4) as described in our response to referee comment 2.1.2. The model results were correct, and the text and conclusions are therefore unaffected by this change.

2. The text on the weathering fluxes has been extended as described in our reply to referee comment 2.1.1

3. We thoroughly revised the text on the air-sea gas exchange experiment, making it more precise, adding more quantitative information and providing more support for our results (extended Fig. S4 as compared to original Fig. S3). While addressing the issue with the air-sea gas exchange experiments, we found a problem in the air-sea gas exchange routine of the model. We solved this problem and re-ran all model experiments with the correct air-sea gas exchange routine – See our reply to reviewer comment 2.2.1 for details.

4. We decided to keep our experiment duration at 2000 years, and give a lists of arguments in our reply to referee comment 2.2.1

5. Captions adjusted to include that all figures are 2000-year differences. The captions of the difference plots now also have a more uniform format

6. This is now referred to in Sect. 2

7. We included a revision of this paragraph in our general revision of section 3.3.1. We provide the reader with quantitative estimates of the equilibration times (from Jones et al., 2014) and stress that we discuss the equilibration timescale through the air-sea interface. We now describe the contrast between $CO_2$ (for which equilibration depends on the Revelle Buffer factor) and the $^{13}CO_2/^{12}CO_2$ ratio (which is not facilitated by the buffering reaction of $CO_2$ with $H_2O$ but scales by the $DIC:CO_2$ ratio, see the text and Galbraith et al., 2015).

8. We agree that the use of the word equilibrium is misleading when discussing $\delta^{13}$C. We therefore removed the second part of sentence in order to keep the information of small carbon outgassing in the manuscript.

9. The original text was meant to describe average equilibrated $\delta^{13}C_{surface}$, which is approximately 2 ‰ (Murnane and Sarmiento, 2000), since $\delta^{13}C^{atm}$ is about -6.5 ‰ and mean air-sea fractionation about 8.5 ‰ (Mook et al., 1986). $\delta^{13}C_{surface}$ is however indeed strongly dependent on temperature, and we therefore adjusted the text to clarify the latitudinal contrasts. We also added information on the model-specific temperature effects on air-sea fractionation.

**Marked-up manuscript**

See remainder of this document.

[revised manuscript text omitted]

---

## Author Response (AR3)

**Author's response**

Dear Professor Joos, dear reviewer(s),

Thank you for your final feedback on our revised manuscript. As we made no further changes to the manuscript, this author's response is limited to presenting the reviewer comment and our reply.

Yours sincerely, Anne Morée and co-authors

**Reviewer comment**

I have gone through the authors' responses to the my comments from the second review. All in all, the responses appear to be satisfactory. The required additional information (weathering fluxes etc.) have been provided. The $\delta^{13}C$ of the weathering flux at −11 ‰ (from +14‰) is still somewhat peculiar: as mentioned in my previous review, the total sedimentary carbon subject to weathering has an average $\delta^{13}C$ of −5 ‰, wherein the most abundant source are carbonate rocks with a $\delta^{13}C$ of around 1.8 ‰. Perhaps the amount of organically derived weathering DIC is larger than in reality (this would be the case the shelf carbonate sink is not considered). At least −11 ‰ can be more easily explained than +14 ‰. The model apparently included some errors that have now been corrected, making the results more plausible now. I regret that the authors still have not changed their mind about the duration of the perturbation experiments. I am, however, truly disappointed about the fact that even this second major (!) revision is not devoid of its share of confusion. The previous version included in its Supplement Figures S3 and S4:

• Fig. S3 represented the surface ocean ΔpCO2 for the control run;

• Fig. S4 represented the specific air-to-sea exchange flux of CO2 for the control run, the Fast gas and the Slow gas experiments.

These two figures presented several deficiencies:

1. inconsistent colour scales

2. physically incompatible results

According to the "Author's response", the incompatibility between the results (deficiency (2)) was due to a plotting error during the production of Fig. S3. Regarding deficiency (1), we read on the 12th page of the authors' response (page numbers in the authors' response would have been the reviewers best friend . . . ) that "We corrected this in the new Fig. S4 so that both figures now have the first positive interval in green." Unfortunately

• the revised Supplement contains only one figure with these informations (which other one could possibly be the second of "both" having the "first positive interval in green"?);

• this is the new Fig. S4, which represents the surface ocean ΔpCO2 for the control run (formerly shown on the former Fig. S3) and the Fast gas and the Slow gas experiments.

So, the new Fig. S4 actually includes the old Fig. S3 as its left panel and has equivalent panels added for the two perturbation experiments. The old Fig. S4 has been discarded. At least the old Fig. S4 (specific CO2 exchange rates) and the new Fig. S4 ($\Delta$pCO2 ) are compatible (at first sight), but to check this, one has to compare graphs carrying different information in different revisions of the Supplement. None of these comments is meant to be a showstopper and I am ready to give green light for the publication. I leave it to the editor to decide on whether the inconsistency between the figures in the Supplement and the Author's response needs correction or not.

**Author's reply**

Thank you again for your time and effort spent on our manuscript.

Regarding the $\delta^{13}$C of the weathering flux, we do not consider fractionation during $CaCO_3$ formation, so a direct comparison with natural riverine $\delta^{13}$C of DIC is not possible. As $\delta^{13}$C of $CaCO_3$ is less depleted than $\delta^{13}$C of organic matter, one would expect a more depleted signature (−11 ‰) in our model setup as compared to actual total sedimentary carbon (which the reviewer reports a typical $\delta^{13}$C signature for of −5 ‰).

We apologise for the remaining confusion about Figures S3 and S4. About the colour scales, 'both figures with a first positive interval in green' refers to the new Figure S4 (old Fig. S3, the pCO2 difference), which indeed has its first interval in green, as well as the equivalent of the old Fig. S4 (the air-sea fluxes), which we only presented in our 'Author's Response' (on page 12). The Figure provided in our last response on page 12 was provided for exactly that reason of comparison of the old Fig S3 and S4 with the new Fig. S3 and S4, but we decided it could be left out of the final text. As the final SI and manuscript texts and figures are not affected by this, we have chosen to make no further changes to the manuscript.

**Manuscript changes**

None